# Quantifying river water contributions to the transpiration of riparian trees along a losing river: Lessons from stable isotopes and iteration method

Yue Li[1, 2], Ying Ma[1, 2], Xianfang Song[1, 2], Qian Zhang[3], Lixin Wang[4]

[1]Key Laboratory of Water Cycle and Related Land Surface Processes, Institute of Geographic Sciences and Natural Resources Research, Chinese Academy of Sciences, Beijing 100101, China

[2]University of Chinese Academy of Sciences, Beijing 100049, China

[3]Institute of Geographic Sciences and Natural Resources Research, Chinese Academy of Sciences, Beijing 100101, China

[4]Department of Earth Sciences, Indiana University-Purdue University Indianapolis (IUPUI), Indianapolis, IN 46202, United States

*Correspondence to*: Ying Ma (maying@igsnrr.ac.cn)

**Abstract.** River water plays a critical role in riparian plant water use and riparian ecosystem restoration along losing rivers (i.e., river water recharging underlying groundwater). How to quantify the contributions of river water to the transpiration of riparian plants under different groundwater levels and the related responses of plant water use efficiency is a great challenge. In this study, observations of water stable isotopes ($\delta^2$H and $\delta^{18}$O), $^{222}$Rn, and leaf $\delta^{13}$C were conducted for the deep-rooted riparian weeping willow (*Salix babylonica* L.) in 2019 (dry year) and 2021 (wet year) along the Chaobai River in Beijing, China. We proposed an iteration method in combination with the MixSIAR model to quantify the river water contribution to the transpiration of riparian *S. babylonica* and its correlations with the water table depth and leaf $\delta^{13}$C. Our results demonstrated that riparian *S. babylonica* took up deep water (in the 80−170 cm soil layer and groundwater) by $56.5 \pm 10.8\%$. River water recharging riparian deep water was an indirect water source and contributed 20.3% of water to the transpiration of riparian trees near the losing river. Significantly increasing river water uptake (by 7.0%) and decreasing leaf $\delta^{13}$C (by −2.0‰) of riparian trees were observed as the water table depth changed from 2.7 m in the dry year of 2019 to 1.7 m in the wet year of 2021 ($p < 0.05$). The higher water availability probably promoted stomatal opening and thus increased transpiration water loss, leading to the decreasing leaf $\delta^{13}$C in the wet year compared to the dry year. The river water contribution to the transpiration of riparian *S. babylonica* was found to be negatively linearly correlated with the water table depth and leaf $\delta^{13}$C ($p < 0.01$). The rising groundwater level may increase the water extraction from groundwater/river and produce a consumptive river-water-use pattern of riparian trees, which can have an

adverse impact on the conservation of both river flow and riparian vegetation. This study provides new insights into understanding the mechanisms of the water cycle in a groundwater-soil-plant-atmosphere continuum, and managing water resources and riparian afforestation along losing rivers.

## 1 Introduction

Ongoing climate warming and groundwater overexploitation have altered river runoff and bank storage globally,

which have further resulted in widespread risks such as the flow of river water into underlying groundwater (i.e., "losing" river) and even drying up (Winter et al., 1998; Schindler and Donahue, 2006; Allen et al., 2015; Jasechko et al., 2021). Water replenishment for losing rivers and riparian revegetation has been applied worldwide to restore the river ecosystems (Smith et al., 2018; Long et al., 2020). The water replenishment to losing rivers contributed to bank storage and groundwater storage recovery by 40% (Long et al., 2020). However, large-scale riparian

revegetation increased plant transpiration substantially, which in turn led to a great loss of riparian bank storage and even river runoff (Moore and Owens, 2012; Dzikiti et al., 2013; Missik et al., 2019; Mkunyana et al., 2019). Therefore, it is critical to determine what water sources and how much river water are taken up by riparian trees and the responses of tree water use characteristics to groundwater level variations (Wang et al., 2022). This can help us to regulate river runoff and vegetation water requirements in the revegetated riparian zones.

The potential water sources of riparian trees along a losing river are generally considered a mix of soil water at different depths, groundwater, and river water (Alstad et al., 1999; White and Smith, 2020). However, there is a debate on whether river water is a potential water source for riparian trees and how it becomes available to plants. Most previous studies considered river water as a direct water source to evaluate the river water contribution (RWC) to the transpiration of riparian trees (Alstad et al., 1999; Zhou et al., 2017; White and Smith, 2020). Based

on the stable isotopic signatures of different water sources and plant stem water, these studies found that river water directly contributed up to 80% to riparian plant transpiration (Dawson and Ehleringer, 1991; Busch et al., 1992; Alstad et al., 1999; Zhou et al., 2017; White and Smith, 2020). Nevertheless, some studies argued that river water was not a potential water source and rarely contributed to the transpiration of riparian trees (Dawson and Ehleringer, 1991; Bowling et al., 2017; Wang et al., 2019a). Dawson and Ehleringer (1991) first discovered that

the mature streamside trees growing in or next to a perennial river did not use river water but depended on water from deeper strata. Similar findings have also been reported regarding riparian phreatophyte trees (*Populus*

*fremontii* and *Salix gooddingii*) and riparian deep-rooted trees (Busch et al., 1992; Bowling et al., 2017; Wang et al., 2019a). Even under shallow groundwater with high salinity, no river water was directly taken up by riparian *Eucalyptus coolabah* alongside an ephemeral arid zone river in Australia (Costelloe et al., 2008). Growing evidence suggested that riparian trees rarely took up river water directly at a certain distance from the riverbank because their lateral roots could not reach the river (Mensforth et al., 1994; Thorburn and Walker, 1994). Nevertheless, riparian trees can indirectly utilize river water that recharges deep zone (e.g., deep soil water and groundwater) when their roots tap into the groundwater level (Mensforth et al., 1994; Wang et al., 2019a). The RWC to the transpiration of riparian trees may be overestimated if the river water is considered a direct water source. How to separate and quantify the contributions of the indirect river water source to the transpiration of riparian trees near losing rivers is a great challenge.

Several approaches have been well developed in recent years to determine plant root water uptake patterns. For example, the graphical inference and direct comparison of isotopic values between plant stem water and different water sources (Dawson and Ehleringer, 1991; Busch et al., 1992; Costelloe et al., 2008; Zhao et al., 2016), statistical two- or multi-source linear mixing models (Alstad et al., 1999; Zhou et al., 2017), and the MixSIAR Bayesian mixing model (Wang et al., 2019a; Wang et al., 2020; White and Smith, 2020; Li et al., 2021) that are integrated with water stable isotopes ($\delta^2$H and $\delta^{18}$O) have been extensively employed to identify the potential water sources taken up by riparian trees. The MixSIAR model has more advantages in quantifying water source contributions and accounting for uncertainties in the isotopic values (Stock and Semmens, 2013; Ma et al., 2016). The indirect RWC to the transpiration of riparian trees can be estimated by quantifying both the direct water source contributions to the transpiration of riparian trees and the RWC to riparian deep water. A multi-iteration method (Marek et al., 1990; Zaid, 2010) is key to calculating the proportional contributions of total (old and current) river water to riparian deep water, which enhances the estimation accuracy of the indirect RWC to the transpiration of riparian trees. The radioactive Radon ($^{222}$Rn) has been broadly utilized for tracing groundwater origins and corresponding pathways in riparian zones (Close et al., 2014; Zhao et al., 2018). Based on $^{222}$Rn concentration, Stellato et al. (2013) estimated the river infiltration velocities into the riparian groundwater system in the Petrignano d'Assisi plain in central Italy, which varied from 1 to 39 m/day. It is helpful to estimate the residence time of recharged groundwater from river water and its effects on the RWC to the transpiration of riparian trees. A combination of these methods can give a more reliable quantification of the indirect RWC to the transpiration

of riparian trees.

As far as we know, the trade-off between the RWC to the transpiration of riparian trees and plant eco-physiological characteristics is unclear, which is critical to understanding the relationships between river runoff and vegetation water requirements in the revegetated riparian zones. The RWC to the transpiration of riparian trees can substantially affect the leaf-level water use efficiency (WUE) and the growth of riparian trees. Tree WUE

is a key characteristic of plant water use, which can be defined as the ratio of photosynthetic rate to transpiration rate. Since leaf $\delta^{13}C$ values are positively related to tree WUE, leaf $\delta^{13}C$ has been widely employed as an indicator of tree WUE for $C_3$ photosynthesis plants (Farquhar et al., 1989). For instance, based on the leaf $\delta^{13}C$ measurements, Thorburn and Walker (1994) found that the riparian *Eucalyptus camaldulensis* with more frequent access to river water had a higher tree WUE compared to those far away from the riverbank. Furthermore, the

variations of the water table depth (WTD) at different distances from the riverbank in the riparian zones play a critical role in both the RWC to the transpiration of riparian trees and tree WUE (Horton and Clark, 2001; Liu et al., 2017; Xia et al., 2018). Li et al. (2022) elucidated that the water table decline led to an increase in deep-water contribution to riparian *Salix babylonica* L. and tree WUE along the distance from the riverbank. Qian et al. (2017) reported a higher RWC to the transpiration of riparian *Ginkgo biloba L.* at the shallower WTD plot closer to the

riverbank compared to the other two plots away from the riverbank along a losing river. However, little attention has been paid to quantifying the relationships between the RWC to the transpiration of riparian trees and tree WUE or WTD near a losing river.

The overall goal of this study was to clarify the impacts of river water on the water use of riparian trees along a gradient of WTD. Focusing on a losing river in Beijing, China, the specific objectives of this study were as

follows: (1) proposing an iteration method in combination with the MixSIAR model and water stable isotopes ($\delta^2H$ and $\delta^{18}O$) to quantify the RWC; (2) comparing the contributions of river water to the transpiration of riparian trees along a gradient of WTD at different distances from the riverbank in dry and wet years; (3) identifying the relationships between the RWC to the transpiration of riparian trees and tree WUE (indicated by leaf $\delta^{13}C$ values) as well as WTD. Our results will provide critical insights into plantation management, bank storage conservation,

and ecosystem health maintenance for losing rivers.

## 2 Materials and methods

### 2.1 Study area

The study area was in the reaches of the Chaobai River, located in Shunyi district, Beijing, China (40°07′30″N, 116°40′37″E) (Fig. 1). A temperate continental sub-humid monsoon climate prevails in this area, with the annual mean temperature and evaporation of 11.5°C and 1175 mm, respectively. The average total precipitation from April to November between 1961 and 2021 was 532.8 mm, with 84.5% of which occurring in the rainy season (from June to September) (Fig. 2a). Owing to continuous drought and groundwater overexploitation, the Chaobai River dried up from 1999 to 2007 and the riparian ecosystem seriously degraded. The "ecological water" (including reclaimed water, reservoir water, and diverted water by the South-to-North Water Transfer Project) has been supplied through a systematic water release by dams to restore this dry river since 2007. A total of 51.1 million and 380 million cubic meters of ecological water sources were released to the Chaobai River in 2019 and 2021, respectively. More than 33 km$^2$ of the riparian zone was revegetated until 2020. The deep-rooted riparian weeping willow (*Salix babylonica* L.) was one of the most widely planted species alongside the Chaobai River because the *S. babylonica* trees could adapt well to dramatic fluctuations in the WTD. Hence, this research selected *S. babylonica* trees as representative of riparian species. Three plots at different distances of 5 m (D05), 20 m (D20), and 45 m (D45) from the riverbank (one plot per distance) were also selected for field measurements and sample collection (Fig. 1).

### 2.2 Field measurements and data collection

The field measurements were carried out from April to November in both 2019 and 2021, with no field observation in 2020 due to COVID-19. The daily precipitation data from 1961 to 2021 and the daily mean temperature (T), relative air humidity (RH), solar radiation, and reference evapotranspiration (ET$_0$) data during the observation period in the Shunyi district were collected from the China Meteorological Data Service Centre (http://data.cma.cn/en). Daily mean vapor pressure deficit (VPD) was calculated using the RH and T data (Wang et al., 2014; Schoppach et al., 2019).

The groundwater level in each plot was recorded monthly in 2019 and in 2021 via a pressure gauge (HOH-S-Y, King Water Co Ltd., Beijing, China) installed in the groundwater monitoring well. The river water level was recorded using a water level gauge at the same time as the observed groundwater levels.

## 2.3 Sample collection and isotopic analyses

Twelve sampling campaigns on May 5, June 14, July 26, August 15, September 26, and November 5 in 2019 and April 24, May 25, June 26, July 15, September 1, and November 5 in 2021 were conducted to collect groundwater, river water, soil, stem, and leaf samples. Groundwater in each plot was sampled by a sucking pump from the monitoring well, and a plexiglass hydrophore water sample collector with a capacity of 1 L was utilized to collect the nearby river water. Precipitation was sampled after each precipitation event via a device consisting of a funnel, a polyethylene bottle, and a ping-pong ball. A total of 135 precipitation samples were collected throughout the whole years of 2019 (53 samples) and 2021 (82 samples). All precipitation, groundwater, and river water samples were stored in a refrigeration box with several ice bags to minimize evaporation in the field, then they were delivered to the laboratory and kept at 4°C in the refrigerator until water stable isotope ($\delta^2$H and $\delta^{18}$O) analysis. The groundwater and river water were also collected with 100-ml brown glass vials to measure $^{222}$Rn concentration in the field.

One riparian *S. babylonica* tree was selected in each plot (three trees in total) for $\delta^2$H and $\delta^{18}$O measurements in xylem water as well as $\delta^{13}$C analysis in plant leaves. The mean breast-height diameter of three sampled trees at different distances of 5 m, 20 m, and 45 m from the riverbank was $28.6 \pm 4.4$ cm. Five mature and suberized stem samples were taken from the same riparian *S. babylonica* tree in each plot using an averruncator with a length of 5 m. We removed the bark and phloem of the sampled stems, and then put the remaining xylem samples into three reduplicative 12-ml brown glass vials sealed with parafilm. These three reduplicative xylem samples were extracted and water stable isotopes were measured. Meanwhile, more than 50 mature leaves without petioles were sampled from the collected stems using pruning shears and mixed into one leaf sample for $\delta^{13}$C analysis. The xylem and mature leaf samples were stored in a refrigeration box with several ice bags in the field. Then the xylem samples were transported to the laboratory and kept in a refrigerator at $-10$°C before water extraction and isotope analysis. The mature leaves were oven-dried at 65°C for 72 h on the day of sampling, then they were ground and passed through a 0.15 mm sieve to analyze leaf $\delta^{13}$C (Wang et al., 2019b; Cao et al., 2020).

Soils at depths of 0−5, 5−10, 10−20, 20−30, 40−60, 60−80, 90−110, 150−170, 190−210, 250−270, and 280−300 cm in one soil profile near the selected *S. babylonica* trees were sampled by a power auger (CHPD78, Christie Engineering Company, Sydney, Australia). One portion of each soil sample was put into a 12-ml brown glass vial and stored at $-10$°C before water stable isotope analysis, and the other portion was packed into an

aluminum box for gravimetric soil water content (SWC) measurement via the oven-drying method (Wang et al., 2019b; Li et al., 2021).

The automatic cryogenic vacuum distillation system (LI-2100, LICA, Beijing, China) was employed to extract water from xylem and soil samples, which generally ran for at least 2.5 h. All the extracted water from the xylem and soil samples was filtered to remove impurities. We weighed all the xylem and soil samples before and after extraction as well as oven-dried samples. Subsequently, to ensure the water extraction efficiency above 99% and to avoid isotopic fractionation during water extraction, the efficiency of water extraction was calculated as follows:

$$E_{WE} = \frac{W_{BE} - W_{AE}}{W_{BE} - W_{OD}} \times 100\% \tag{1}$$

whereas $E_{WE}$ represents the efficiency of water extraction; $W_{BE}$ and $W_{AE}$ represent the weights of xylem/soil samples before and after extraction, respectively; $W_{OD}$ represents the weights of oven-dried xylem or soil samples.

The $\delta^2H$ and $\delta^{18}O$ values of soil water, river water, groundwater, and precipitation were analyzed using an isotopic ratio infrared spectroscopy system (IRIS) (DLT-100, Los Gatos Research, Mountain View, USA) (Li et al., 2021). The isotope ratio mass spectrometry system (IRMS) (MAT253, Thermo Fisher Scientific, Bremen, Germany) which could prevent organic pollution of plants was used to measure $\delta^2H$ and $\delta^{18}O$ values of xylem water as well as leaf $\delta^{13}C$ value. There was the same measurement accuracy for both the IRIS and IRMS systems ($\pm1‰$ for $\delta^2H$ and $\pm 0.1‰$ for $\delta^{18}O$). The Vienna Standard Mean Ocean Water (VSMOW) was utilized to calibrate and normalize the $\delta^2H$ and $\delta^{18}O$ measurements in different waters, while the Vienna Pee Dee Belemnite (V-PDB) was used for calibrating leaf $\delta^{13}C$ values.

The $^{222}Rn$ concentration in the groundwater and river water samples ($C_{Water}$, Bq/l) was determined based on the air $^{222}Rn$ concentration values ($C_{Air}$, Bq/m³) measured by a $^{222}Rn$ monitor (Alpha GUARD PQ2000 PRO, Bertin Instruments, Germany). 100 ml of the water sample was slowly poured into the air-tight glass bottles and then purged with air in a closed gas cycling system. The $C_{Air}$ in the $^{222}Rn$ monitor was recorded at 10-minute intervals. The air inside the measurement set-up maintained a certain $^{222}Rn$ concentration right before the water sample injection ($C_{System}$, Bq/m³). It is generally assumed that when $C_{System}$ is around or lower than 80 Bq/m³, the existing $C_{System}$ can be ignored accordingly (Saphymo, 2017). We conducted more than four intervals to ensure that the $C_{System}$ was smaller than 80 Bq/m³. The measurement range of $C_{Air}$ was 2–2,000,000 Bq/m³ with a

measurement precision of 3% (Saphymo, 2017). The $C_{water}$ was calculated as follows:

$$C_{Water}=\frac{C_{Air}\times\left(\frac{V_{System}-V_{Sample}}{V_{Sample}}+k\right)-C_{System}}{1000} \tag{2}$$

where $V_{System}$ stands for the interior volume of the measuring set-up (ml), which is 1122 ml in this study; $V_{Sample}$ symbolizes the volume of the water sample (ml); k denotes the $^{222}$Rn distribution coefficient of water/air (–), which can be set as 0.26 within the specified temperature range around the mean room temperature of 20°C (Clever, 1985).

We identified the average residence time ($T_{res}$, day) of recharged groundwater from river water based on the $^{222}$Rn isotopes (Hoehn and Von Gunten, 1989), which was described as follows:

$$T_{res}=\frac{1}{\lambda}\times\ln\left(\frac{C_e-C_r}{C_e-C_g}\right) \tag{3}$$

where $\lambda$ represents the decay coefficient (0.181 day$^{-1}$) (Hoehn and Von Gunten, 1989); $C_e$ signifies the $^{222}$Rn concentration of background groundwater when the equilibrium between radon production and decay is reached; The measuring $^{222}$Rn concentration of groundwater in aquifers more than 100 m away from the riverbank remained constant in this study (with an average value of 7400.0 $\pm$ 35.4 Bq/m$^3$), suggesting that $C_e$ can be defined as 7400.0 Bq/m$^3$; $C_r$ indicates the $^{222}$Rn concentration of river water (Bq/m$^3$); $C_g$ stands for the $^{222}$Rn concentration of riparian groundwater (Bq/m$^3$).

## 2.4 Determination of RWC to the transpiration of riparian trees

In this study, water stable isotopes ($\delta^2$H and $\delta^{18}$O) were integrated within the MixSIAR model and an iteration method was proposed to identify the contributions of the indirect river water that recharged riparian deep water to the transpiration of riparian *S. babylonica* trees (Figs. 4-5). First, the direct water source (including soil water in different layers and groundwater) contributions to the transpiration of riparian trees were determined via $\delta^2$H and $\delta^{18}$O values of different waters and the MixSIAR model. Second, the proportional contributions of river water to riparian deep water (i.e., riparian groundwater and deep soil water in the 80−170 cm layer) were determined by the MixSIAR model and water stable isotopes. Finally, the proposed iteration method was applied to quantify the proportions of the indirect river water source taken up by riparian trees (Figs. 4-5).

The MixSIAR model is a Bayesian mixing model which can be integrated with water stable isotopes to quantify the proportions of source contributions to a mixture (Stock and Semmens, 2013). The input data of the MixSIAR model include mixture data, source data, and discrimination data. In this study, the mean and standard

deviation (SD) of the isotopic values of each water source for riparian trees/riparian deep water were inputted as source data into the MixSIAR, while the measured isotopic values of xylem water/riparian deep water were input as raw mixture data into the MixSIAR. The discrimination data for both $\delta^2H$ and $\delta^{18}O$ were set to zero because the input $\delta^2H$ and $\delta^{18}O$ values in the MixSIAR were non-fractionated or $\delta^2H$-corrected. The Markov Chain Monte

Carlo parameter was set to the run length of "very long". The trace plots and three diagnostic tests (i.e., Gelman–Rubin, Heidelberger–Welch, and Geweke) were adopted to determine whether the MixSIAR model converged (Stock and Semmens, 2013). Then, the mean and SD values of different water source contributions could be estimated using the MixSIAR model.

## 2.4.1 Quantifying proportional contributions of direct water sources to riparian trees

Soil water was an important direct water source for the transpiration of riparian *S. babylonica* trees. We measured soil water isotopes at 11 depths in each plot at a distance of 5 m, 20 m, and 45 m from the riverbank. To reduce errors in the analytical procedure, four soil layers (0−30 cm, 30−80 cm, 80−170 cm, and 170−300 cm) were determined to identify the main root water uptake depth of riparian trees according to seasonal variations in the SWC, water isotopes, and WTD. The average soil water isotope values for the 0−30 cm soil layer were determined

as the average of the soil water isotope values of 0−5 cm, 5−10 cm, 10−20 cm, and 20−30 cm soil layers because the water isotopes underwent strong evaporation and SWC changed considerably seasonally. We determined the average soil water isotope values for the 30-80 cm (average of 40-60 cm and 60-80 cm soil layers) and 80-170 cm (average of 90-110 cm and 150-170 cm soil layers) soil layers because the water isotopes and SWC were almost stable. The average soil water isotope values for the 170-300 cm soil layer were determined as the average

of the soil water isotope values of 190−210 cm, 250−270 cm, and 280−300 cm soil layers, which varied with the fluctuations of groundwater levels. Groundwater could also be considered a direct water source for phreatophyte riparian trees (Dawson and Ehleringer, 1991; Busch et al., 1992). As the isotopic composition of soil water in the 170−300 cm layer (−57.6‰ ± 2.0‰ for $\delta^2H$ and −7.3‰ ± 0.1‰ for $\delta^{18}O$) was similar to that of groundwater (−57.7‰ ± 1.4‰ for $\delta^2H$ and −7.4‰ ± 0.1‰ for $\delta^{18}O$), they were considered to be one water source (groundwater).

Mensforth et al. (1994) and Thorburn and Walker (1994) characterized the projected edge of the canopy as the extension range of lateral roots. In this way, it is possible to determine whether or not riparian trees take up river water directly. The projected edge of the canopy in our study was less than 5 m for the riparian *S. babylonica* trees

which were closest to the river (5 m away from the riverbank). This indicated that the lateral roots of *S. babylonica* trees could not tap into the river. Therefore, river water was not regarded as a direct potential water source for tree water uptake, while groundwater and soil water in the 0−30, 30−80, and 80−170 cm layers were used as direct potential water sources for riparian *S. babylonica*.

The $\delta^2H$ offsets between the xylem water in riparian trees and its corresponding potential source waters were observed in this study, which possibly resulted from $\delta^2H$ fractionation in the plant water use processes (Li et al., 2021; Cernusak et al., 2022). These $\delta^2H$ offsets could lead to large errors in estimating the water source contributions using the MixSAIR model. To eliminate the $\delta^2H$ offsets of xylem water from its potential water sources, the measured xylem water $\delta^2H$ values were corrected by the potential water source line (PWL) proposed by Li et al. (2021). The PW-excess (PW-excess = $\delta^2H - a_p\delta^{18}O - b_p$; $a_p$ and $b_p$ are slope and intercept of the PWL, respectively) was calculated to determine the $\delta^2H$ deviation from the PWL, which was subsequently subtracted from the measured xylem water $\delta^2H$ values. To quantify the contributions of direct water sources to the transpiration of riparian *S. babylonica*, the corrected $\delta^2H$ and raw $\delta^{18}O$ in xylem water were set as the mixture data in the MixSIAR model.

## 2.4.2 Quantifying water source contributions to deep soil water and groundwater

The MixSIAR model in conjunction with water stable isotopes ($\delta^2H$ and $\delta^{18}O$) was applied to quantify the proportional contributions of current (between previous sampling time t-1 and current sampling time t) river water to riparian deep water (i.e., deep soil water in the 80−170 cm layer or groundwater). The potential water sources of riparian deep soil water in the 80−170 cm layer at t included the in-situ (i.e., water that was already in the deep soil layer or groundwater) soil water in this layer at t-1, soil water in the 0−80 cm layer at t-1, river water between t-1 and t, precipitation between t-1 and t, and groundwater between t-1 and t (Fig. 4a). We considered the in-situ groundwater at t-1, soil water in the 0−170 cm layer at t-1, river water between t-1 and t, and precipitation between t-1 and t as the potential water sources for riparian groundwater at t (Fig. 4b). The isotopic changes from t-1 to t (such as fractionation during this period) were negligible when calculating the contribution of upper soil water (i.e., in the 0-80 cm or 0-170 cm layers) at t-1 to deep moisture (i.e., soil water in the 80-170 cm layer or groundwater). The $\delta^2H$ and $\delta^{18}O$ values of riparian deep water at t were set as the mixture data in the MixSIAR model, while the water isotopes of their potential water sources were regarded as the source data.

### 2.4.3 An iteration method to determine RWC to the transpiration of riparian trees

After determining both riparian deep-water contributions to the transpiration of trees and the RWC to riparian deep water, the proportional contributions of the river water between t-1 and t to the transpiration of riparian trees were quantified. It is worth noting that riparian deep soil water (80-170 cm) and groundwater could be recharged by river water continuously when the groundwater levels lay below the riverbeds (i.e., losing rivers). Therefore, the proportional contribution of the old river water (before t-1) to riparian deep water should not be ignored. The total RWC to riparian deep water should be quantified explicitly during the entire period of the river flow into the riparian deep zone since 2007. We suppose that the contributions of old river water to riparian in-situ deep water are identical to those of current river water (between t-1 and t) to riparian in-situ deep water. We proposed an iteration method using the following expression to quantify the total RWC to the transpiration of riparian *S. babylonica* trees near the losing rivers:

$$RWC = P_s * S_r + P_g * G_r$$

$$= P_s*(s_r^t + s_r^{t-1}) + P_g*(g_r^t + g_r^{t-1})$$

$$= P_s*(s_r^t + s_r^t*s_s^{t-1} + s_r^t*(s_s^{t-1})^2 + s_g^t*g_r^t + s_g^t*g_r^t*g_g^{t-1} + s_g^t*g_r^t*(g_g^{t-1})^2) + P_g*(g_r^t + g_r^t*g_g^{t-1} + g_r^t*(g_g^{t-1})^2)$$

$$= (P_s*s_r^t + P_g*g_r^t + P_s*s_g^t*g_r^t) + (P_s*s_r^t*s_s^{t-1} + P_g*g_r^t*g_g^{t-1} + P_s*g_r^t*s_g^t*g_g^{t-1}) + (P_s*s_r^t*(s_s^{t-1})^2 + P_g*g_r^t*(g_g^{t-1})^2 +$$

$$P_s*s_g^t*g_r^t*(g_g^{t-1})^2) \tag{4}$$

where $S_r$ and $G_r$ represent total RWC to riparian deep soil water in the 80−170 cm layer and groundwater, respectively; $P_s$ and $P_g$ represent the contributions of riparian deep soil water in the 80−170 cm layer and groundwater to the transpiration of riparian trees, respectively; $s_r^{t-1}$ and $g_r^{t-1}$ denote the proportional contributions of the old river water (before t-1) to riparian deep soil water in the 80−170 cm layer and groundwater, respectively; $s_s^{t-1}$, $s_r^t$, and $s_g^t$ signify the proportional contributions of in-situ soil water in the 80−170 cm layer at t-1, river water during t-1 to t, and groundwater during t-1 to t to riparian deep soil water in the 80−170 cm layer at t, respectively; $g_g^{t-1}$ and $g_r^t$ symbolize the proportional contributions of in-situ groundwater at t-1 and river water from t-1 to t to riparian groundwater at t, respectively.

The expression of "$P_s*s_r^t + P_g*g_r^t + P_s*s_g^t*g_r^t$" in Equation (4) was proposed to determine the current river water (between t-1 and t) contributions to the transpiration of riparian trees. The second iteration ($P_s*s_r^t*s_s^{t-1} + P_g*g_r^t*g_g^{t-1} + P_s*g_r^t*s_g^t*g_g^{t-1}$) and the third iteration ($P_s*s_r^t*(s_s^{t-1})^2 + P_g*g_r^t*(g_g^{t-1})^2 + P_s*s_g^t*g_r^t*(g_g^{t-1})^2$) were applied to quantify the proportional contributions of old river water that recharged riparian in-situ deep water to trees (Fig. 5). We only

applied three iterations because the differences between the RWCs in the third iteration and the next iteration were smaller than 0.1%. Using this proposed iteration method, we accurately estimated the total proportions of old and current river waters to the transpiration of riparian trees.

## 2.5 Statistical analysis

For each variable, we tested the homogeneity of variance between the two studied years and between the three plots using Levene's test. The one-way analysis of variance (ANOVA) was applied to examine differences in each variable among three plots in 2019 and 2021 ($p < 0.05$). The variables included the WTD, SWC, $\delta^2H$ values, and $\delta^{18}O$ values of different water sources and xylem water, $^{222}Rn$ concentration of river water and groundwater, contributions of different water sources to riparian deep water or trees, and leaf $\delta^{13}C$ values. The linear regression model was fitted to the whole dataset in both years to obtain the general relationships between the WTD, leaf $\delta^{13}C$ values, and the RWC to the transpiration of riparian trees. The statistical analysis was carried out in Microsoft Excel (v2016) and SPSS (24.0, Inc., Chicago, IL, USA).

## 3 Results

### 3.1 Hydro-meteorological conditions

The observation period (from April to November) in 2021 was wet with total precipitation of 802.5 mm, which was 1.8 times greater than for the drier year 2019 (445.6 mm) (Fig. 2a). The precipitation amount during the rainy season accounted for 75.4% and 97.0% of the whole precipitation in 2019 and 2021, respectively. The annual mean temperature during the observation period in 2019 and 2021 was 22.4°C and 21.8°C, respectively. The average daily VPD during the observation period was significantly greater in the dry year of 2019 (1.1 kPa) than in the wet year of 2021 (0.9 kPa) ($p < 0.05$) (Fig. S1a and b). There was a significant difference in the average daily $ET_0$ from June to September between the dry year of 2019 (5.0 mm/day) and the wet year of 2021 (4.3 mm/day) ($p < 0.05$), but no significant difference was observed during the rest of observation period (i.e., April, May, October, and November) between the two years ($p > 0.05$) (Fig. S1c and d). No significant difference was found in the daily mean net radiation during the observation period between the dry year of 2019 and the wet year of 2021 ($p > 0.05$) (Fig. S1 c and d).

The river water level fluctuated between 27.9 m and 28.9 m in 2019 and between 27.7 m and 29.3 m in 2021

(Fig. 3). The mean WTD across the three plots was significantly ($p < 0.05$) deeper in 2019 ($2.7 \pm 0.3$ m) than in

2021 ($1.7 \pm 0.5$ m). The WTD increased with increasing distances from the riverbank in both 2019 and 2021 (Fig.

3). The river water continuously recharged the groundwater system ("losing" river) during the observation periods

in 2019 and 2021, which was indicated by a lower groundwater level than the river water level (Fig. 3).

Significantly higher SWC was observed in 2021 compared to 2019 ($p < 0.05$) (Fig. S2). The SWC of each soil

layer at D45 was significantly lower than that at D05 and D20 in 2021 ($p < 0.05$), while no pronounced difference

was observed in the SWC in the 0−30 cm layer among the three plots in 2019 ($p > 0.05$) (Fig. S2).

## 3.2 Direct water source contributions to the transpiration of riparian trees

Precipitation was significantly more depleted in $\delta^2H$ and $\delta^{18}O$ in 2021 (−52.9‰ $\pm$ 30.2‰ for $\delta^2H$ and −8.1‰ $\pm$

3.8‰ for $\delta^{18}O$) than in 2019 (−29.2‰ $\pm$ 18.8‰ for $\delta^2H$ and −4.1‰ $\pm$ 3.0‰ for $\delta^{18}O$) ($p < 0.05$) (Fig. 6). The

slope of the local meteoric water line in 2021 (7.8) was significantly higher than in 2019 (5.5) ($p < 0.05$),

suggesting that the falling raindrops underwent stronger sub-cloud evaporation in 2019 (Zhao et al., 2019). The

$\delta^2H$ and $\delta^{18}O$ values of the surface soil water (above 30 cm depth) were significantly lower and more variable in

2021 than in 2019 ($p < 0.05$) (Fig. 6). In contrast, there were slightly higher water isotopic compositions in the

30−170 cm soil layer in 2021 compared to 2019. No significant difference was observed in the isotopic

compositions of the soil water below 170 cm depth and groundwater between 2019 and 2021 ($p > 0.05$). The $\delta^2H$

and $\delta^{18}O$ values of soil water in the 80−170 cm layer were significantly lower than those of groundwater in 2019

($p < 0.05$), while no significant difference was observed between soil water isotopes in the 80−170 cm layer and

groundwater isotopes in 2021 ($p > 0.05$). Groundwater was significantly more depleted in $\delta^2H$ and $\delta^{18}O$ compared

to river water in both years ($p < 0.05$) (Fig. 6). The $\delta^2H$ and $\delta^{18}O$ values of xylem water during the observation

periods in 2019 and 2021 were not significantly different ($p > 0.05$), but they were gradually lower with the

increasing distance from the riverbank.

The contributions of the surface soil water to the transpiration of riparian trees in 2019 ($20.1\% \pm 9.7\%$) were

similar to those of 2021 ($19.0\% \pm 10.5\%$). No significant difference was also observed in the soil water

contributions to the transpiration of riparian *S. babylonica* in the 30−80 cm layer between the two years ($p>0.05$)

(Fig. 7). The *S. babylonica* tree13hinese principally relied on riparian deep water below the 80 cm depth in both

2019 (55.9%) and 2021 (57.1%). There was no significant difference in the riparian deep-water contributions to

the transpiration of *S. babylonica* trees between the three distances from the riverbank ($p > 0.05$) (Fig. 7).

Nevertheless, the soil water contributions in the 80−170 cm layer to the transpiration of riparian trees decreased with increasing distance from the riverbank in both years, whereas the proportions of groundwater taken up by riparian trees increased from D05 to D45 in both 2019 (from 27.6% to 32.1%) and 2021 (from 17.0% to 32.2%)

(Fig. 7). café groundwater contributions to the transpiration of riparian *S. babylonica* trees increased significantly ($p < 0.05$) from April to July in both years. They plummeted significantly ($p < 0.05$) and reached a minimum in September 2021.

## 3.3 Water source contributions to riparian deep soil water and groundwater

The primary water sources of riparian deep soil water in the 80-170 cm layer were the in-situ soil water in this

layer (with a mean value of 33.1%) and groundwater capillary rise (with a mean value of 25.3%) in 2019 (Fig. 8). However, the in-situ soil water in the 80-170 cm layer (with a mean value of 23.9%), groundwater capillary rise (with a mean value of 24.6%), and river water (with a mean value of 24.4%) contributed almost equally to riparian deep soil water in 2021. The in-situ soil water contribution to riparian deep soil water was significantly higher in 2019 than in 2021 ($p < 0.05$). However, the river water contributed less to riparian deep soil water in 2019 (with

a mean value of 15.7%) compared to 2021 ($p < 0.05$). The RWC to riparian deep soil water was the lowest in August 2019 ($11.3\% \pm 4.5\%$) and in June 2021 ($13.6 \pm 3.8\%$), respectively. The in-situ soil water contributions to riparian deep soil water showed a significant increase with increasing distance from the riverbank, while the RWC to riparian deep soil water decreased from D05 to D45 in both years ($p < 0.05$) (Fig. 8).

The in-situ groundwater contribution was significantly higher in 2019 (with a mean value of $56.0\% \pm 11.2\%$)

than in 2021 ($37.1\% \pm 16.7\%$) ($p < 0.05$) (Fig. 9). The average contribution of the river water to riparian groundwater was $28.1\% \pm 12.1\%$ during the observation period. There was a significantly higher RWC to riparian groundwater in 2021 (with a mean value of $35.1\% \pm 11.9\%$) than in 2019 (with a mean value of $21.1\% \pm 7.2\%$) ($p < 0.05$). The lowest RWC ($13.0\% \pm 1.2\%$) occurred in August with the lowest groundwater level of 3.1 m in 2019, whereas the contribution of river water to riparian groundwater ($47.1\% \pm 13.2\%$) was the highest in July

with a higher groundwater level of 1.8 m in 2021 (Figs. 3 and 9). The proportional contribution of the in-situ groundwater to riparian groundwater increased with the increasing distance from the riverbank during the observation periods, while the RWC to riparian groundwater decreased significantly from D05 to D45 ($p < 0.05$) (Fig. 9). There was a significant increase of $^{222}$Rn activity in groundwater from D05 ($494.5 \pm 107.5$ Bq/m$^3$) to D45 ($787.4 \pm 153.2$ Bq/m$^3$) ($p < 0.05$) (Table 1). The $T_{res}$ of recharged groundwater from river water increased

from D05 (0 days) to D45 ($0.15 \pm 0.13$ days) (Table 1). This also indicated that the river recharged riparian deep

strata rapidly and frequently, particularly more significant in the plots closer to the riverbank.

## 3.4 Seasonal variations in RWC to the transpiration of riparian trees

The proportional contributions of river water to the transpiration of riparian *S. babylonica* trees were significantly

higher in 2021 (with a mean value of $23.8\% \pm 7.8\%$) than in 2019 (with a mean value of $16.8\% \pm 4.7\%$) ($p < 0.05$).

Specifically, the most significant monthly difference in the RWC to the transpiration of riparian *S. babylonica*

trees between the dry year of 2019 and the wet year of 2021 was up to $19.8\%$ ($p < 0.01$). The monthly maximum

RWC to the transpiration of *S. babylonica* trees was significantly higher in the wet year of 2021 ($35.2\% \pm 7.0\%$)

compared to the dry year of 2019 ($24.2\% \pm 3.0\%$) ($p < 0.05$).

The riparian *S. babylonica* took up the most river water in July 2021 ($35.2 \pm 7.0\%$), whereas the highest

RWC to the transpiration of riparian trees occurred in June 2019 ($24.2\% \pm 1.6\%$). The minimum river water

uptake for riparian *S. babylonica* in 2021 was in September ($17.7\% \pm 2.7\%$), while trees took up the least river

water in August 2019 ($13.2\% \pm 1.9\%$). Although the precipitation amount in the rainy season was much higher

than in the drought season ($p < 0.01$), no significant difference in the RWC to the transpiration of riparian *S.*

*babylonica* trees was observed between the rainy and drought seasons in the same year ($p > 0.05$) (Figs. 2 and 9).

The difference values of the RWC to the transpiration of riparian trees between the rainy and dry seasons were

not significantly different ($p > 0.05$) in both 2019 ($-4.0\%$) and 2021 ($-4.4\%$) (Fig. 9). This suggested that there

were no significant seasonal variations in the RWC to the transpiration of riparian trees within a year ($p > 0.05$).

The water uptake of river water by riparian *S. babylonica* was significantly different between the three plots

in 2019 ($p < 0.05$), while no difference was observed between the three plots in 2021 ($p > 0.05$) (Fig. 10). In

particular, the RWC to the transpiration of riparian trees decreased significantly by $6.9\%$ from D05 ($20.0\%$) to

D45 ($13.1\%$) in 2019 ($p < 0.05$), whereas there was no significant difference in 2021 ($p > 0.05$) (Fig. 10).

## 3.5 Relationships between leaf $\delta^{13}C$, RWC to the transpiration and WTD

The leaf $\delta^{13}C$ of riparian *S. babylonica* trees was significantly higher in 2019 ($-27.7‰ \pm 1.0$ ‰) than in 2021

($-29.7‰ \pm 0.7$ ‰) ($p < 0.05$) (Table 2). There was a significant increase of the leaf $\delta^{13}C$ from D05 ($-28.8‰$) to

D45 ($-27.0‰$) in 2019 ($p < 0.05$), while no significant difference in the leaf $\delta^{13}C$ was observed between three

plots in 2021 ($p > 0.05$). The lowest leaf $\delta^{13}C$ value of riparian trees occurred on August 15 in 2019 and July 14

in 2021. These minimum values of leaf $\delta^{13}C$ occurred when intense rainfall had not occurred in both years.

There was a significantly negative relationship between the RWC to the transpiration of riparian trees and WTD ($R^2 = 0.57$; $p < 0.01$) (Fig. 11a). The leaf $\delta^{13}C$ of riparian *S. babylonica* was found to be negatively correlated with the RWC to the transpiration of riparian trees ($R^2 = 0.61$; $p < 0.01$) but positively linearly related to WTD ($R^2 = 0.37$; $p < 0.01$) (Fig. 11b and c). This demonstrated that deeper WTD ($2.7 \pm 0.3$ m) resulted in lower RWC to the transpiration of riparian *S. babylonica* and higher leaf-level WUE in the drier year of 2019. In contrast, the riparian *S. babylonica* under shallower WTD ($1.7 \pm 0.5$ m) gave rise to higher RWC but lower leaf-level WUE in the wetter year of 2021.

## 4 Discussion

### 4.1 RWC to the transpiration and effects of the distance from the river on RWC

We identified deep-rooted riparian trees near the losing river to use a small proportion of river water (less than 25%) for transpiration (Fig. 10). The small RWC to the transpiration of riparian trees may originate from three non-exclusive processes: first, the lateral roots of riparian trees further than 5 m away from the riverbank rarely took up river water directly when their projected edges of the canopy (less than 5 m in our study) were out of reach of the river (Busch et al., 1992; Thorburn and Walker, 1994). Instead, they took up riparian deep soil water/groundwater recharged by river water, which likely restricted the RWC to the transpiration of riparian trees. Second, the ecohydrological separation (Brooks et al., 2010; Evaristo et al., 2015; Allen et al., 2019; Sprenger et al., 2019) possibly resulted in large isotopic discrepancies between fast-moving water flow and immobile water for plant water uptake. Although the residence time of recharged groundwater from river water was extremely short (less than 0.28 days) (Table 1), only one-third of riparian groundwater was replaced by the lateral seepage of river water (Fig. 9). Our finding probably indicates that river water recharged mobile groundwater quickly but could not completely replace water held tightly in the soil pores (Brooks et al., 2010; Evaristo et al., 2015; Allen et al., 2019). This was consistent with Sprenger et al. (2019) who found that the lateral seepage of river water or rising groundwater level could briefly saturate riparian soils but could not entirely replace/flush immobile waters or homogenize different water pools isotopically. Third, several recent studies showed that phreatophytic/deep-rooted trees predominantly extended roots into fine pores to take up immobile soil water (Evaristo et al., 2015; Maxwell and Condon, 2016; Evaristo et al., 2019). As mentioned above, the immobile water could not be completely replaced by infiltrating river water, which eventually resulted in a small contribution of river water to

deep-rooted riparian trees. This ecohydrological separation perspective that plant-accessible water pools were separated from the fast-moving water can also be supported by our findings that no significant difference in RWCs to the transpiration of riparian trees was observed between rainy and drought seasons in both dry and wet years (Fig. 9). This is because riparian *S. babylonica* trees preferred to rely on immobile water in fine soil pores and they would not change the river water uptake patterns when the fast-moving precipitation input increased (Brooks

et al., 2010; Sprenger et al., 2019).

       Compared to the small RWC to the transpiration of riparian *S. babylonica* trees (less than 25%) in our study, Alstad et al. (1999) found that riparian *Salix monticola* trees near a losing river on the northeast side of the Rocky Mountain National Park, Colorado relied on rivers for approximately 80% of their transpiration. The significant difference in the RWC to the transpiration of riparian trees between the two studies can be attributed to the

potential water source determination as well as the calculation method. First, Alstad et al. (1999) only considered river water and precipitation as potential water sources for riparian *S. monticola*, which resulted in an overestimation of the RWC to the transpiration of riparian *S. monticola*. This is because the RWC estimation in Alstad et al. (1999) also included the proportions of the indirect river water, in-situ soil water, and in-situ groundwater contributions to the transpiration of riparian *S. monticola*. Second, river water can seep into the

saturated/vadose zone across the riparian riverbank and be further taken up by riparian trees indirectly in the form of river-recharged deep soil water/groundwater. In our study, we separated and determined the contributions of indirect river water sources (i.e., the river-recharged deep soil water in the 80−170 cm layer and groundwater also contained river water) to the transpiration of riparian trees. Accurately quantifying the indirect RWC to deep-rooted riparian trees could assist us to determine the effect of riparian plant water use on river runoff along the

losing river.

       We observed substantial variations in the RWC to the transpiration of riparian trees at interannual (between two years) and spatial (between three distances from the riverbank) scales (Fig. 10). The RWC to the transpiration of riparian *S. babylonica* trees in the wet year of 2021 was 1.4 times greater on average than in the dry year of 2019 (Fig. 10). Nevertheless, riparian *S. babylonica* trees presented similar root architecture (i.e., phreatophyte)

associated with similar water source proportions between dry and wet years (Fig. 7). This suggested that the higher groundwater level in the wet year induced higher RWC to riparian deep water compared to the dry year, which further resulted in a higher indirect RWC to the transpiration of riparian phreatophyte trees in the wet year than

in the dry year. Although there was no significant difference in the deep water (below the 80 cm layer) contributions to the transpiration of riparian trees between the three plots, we observed a substantial impact of the declining groundwater level with increasing distance from the riverbank on the decreased indirect RWC to the transpiration of riparian trees in the dry year of 2019 (Fig. 10). The declining water table and increasing residence time of recharged groundwater from D05 to D45 could consequently lead to the decreasing RWC to riparian deep water along the distance away from the riverbank. Thus, the interannual and spatial variabilities of the RWC to the transpiration of riparian *S. babylonica* trees were generally attributed to the various RWCs to riparian deep water rather than the water uptake patterns of riparian trees. This result is in contrast to that of a previous study conducted by Qian et al. (2017) who reported a significant increase of the RWC to the transpiration of *G. biloba* trees in response to the groundwater level decline. This discrepancy was ascribed to the fact that riparian *G. biloba* had a dimorphic root system and shifted their main water sources from the shallow soil layer to the deeper soil layer. However, the potential root growth rate of riparian phreatophyte *S. babylonica* trees can reach 1-13 mm/day, which allows the riparian *S. babylonica* trees to remain in contact with a rising/declining groundwater level and to maintain constant water uptake proportions from deep strata below the 80 cm depth (Naumburg et al., 2005).

## 4.2 Link between RWC/WUE/WTD and the implications

The water uptake patterns of riparian *S. babylonica* trees generally followed the characteristics of a phreatophyte. We observed that leaf WUE of all *S. babylonica* trees across three plots in both dry and wet years was negatively correlated with the indirect RWC to the transpiration of riparian trees and positively related to WTD (Figs. 10, 11b, and 11c). These relationships are consistent with previous studies (Cao et al., 2020; Ding et al., 2020; Behzad et al., 2022). Higher leaf WUE associated with lower RWC to the transpiration of riparian trees and lower groundwater levels are likely because water stress restricts the stomatal conductance and further reduces the transpiration rate of riparian trees. Specifically, the dry year of 2019 was characterized by higher water demand (indicated by higher VPD) and lower water availability compared to the wet year of 2021, but the energy resource (indicated by net radiation) for riparian trees was similar between the two years (Figs. S1-S2). Hence, we suppose that water limitation rather than energy limitation regulates the leaf-level stomatal conductance of riparian *S. babylonica* trees. The high water demands but low river water availability in the dry year likely resulted in the stomatal closure of riparian trees to minimize water loss, which eventually led to a decrease in transpiration rate and even photosynthetic rate (Fabiani et al., 2021; Behzad et al., 2022). Aguilos et al. (2018) further found that

water stress would enhance radiation-normalized WUE because the lack of water availability induced a stronger reduction in transpiration than photosynthesis. With no difference in the average net radiation between dry and wet years, the lower river water availability in a dry year probably increased leaf WUE. It can be inferred that riparian *S. babylonica* trees took up more river water and possibly exhibited a consumptive river-water-use pattern in the wet year compared to the dry year. This agreed well with previous investigations during which the woody plants showed lower leaf WUE and consumptive water use patterns in the rainy season, while they showed higher leaf WUE and conservative water use patterns with lower soil water availability in the dry season (Horton and Clark, 2001; Cao et al., 2020; Behzad et al., 2022). However, consumptive river water taken up by riparian trees could result in a great loss of river water, which should be avoided in the riparian zone of a losing river that is under restoration by "ecological water".

The WTD played a critical role in the river water uptake of riparian trees near a losing river (Mensforth et al., 1994; Horton and Clark, 2001; Qian et al., 2017; Zhou et al., 2017). We observed that the proportional contributions of the river water to the transpiration of riparian trees decreased linearly in response to groundwater level decline, leading to a proportional increase in leaf WUE (Fig. 11a and b). Our finding was consistent with Horton and Clark (2001) who reported an exponential increase in the leaf WUE of riparian *Salix gooddingii* with increasing WTD. As mentioned above, we emphasized the key role of reduced water availability in decreasing transpiration rate and thus in enhancing leaf WUE in this study. Nevertheless, there were some controversial views that the leaf WUE of plant species increased initially and then decreased with increasing WTD (Antunes et al., 2018; Xia et al., 2018). This can be justified by the fact that riparian trees can tolerate reduced water availability only within a species-specific threshold, beyond which xylem cavitation and even crown mortality occur (Naumburg et al., 2005). This indicates that optimal WTD for plant species is related to the highest leaf WUE, under which plant species can consume less water for transpiration to maximize $CO_2$ assimilation (Antunes et al., 2018; Xia et al., 2018). The breakpoint of WTD was not observed in this study (Fig. 11a and b). Further investigations would need to be conducted under deeper groundwater levels (WTD > 4 m) to optimize the WTD and riparian plant-water relations.

Our results have important implications for untangling the trade-offs between riparian tree water use and river runoff management. The proportion of the RWC to the transpiration of riparian trees was compared between dry and wet years to investigate the impacts of river water availability on the water use characteristics of riparian

trees. The riparian *S. babylonica* trees showed the highest leaf WUE and the lowest river water uptake proportion under the lowest groundwater level condition (with a WTD of 4 m). The rising groundwater level may encourage riparian trees to exhibit a consumptive river-water-use pattern, which should not be recommended in revegetated riparian zones beside an ecological water-recharged losing river. Thus, the relationships between the RWC to the transpiration of riparian trees, leaf-level physiological characteristics (e.g., leaf WUE), and hydro-meteorological conditions are critical to protecting the revegetated riparian zones and maintaining river runoff sustainability.

## 4.3 Advantages and limitations of the MixSIAR model and the iteration method

The iteration method in combination with the MixSIAR model and water stable isotopes is particularly useful for separating and quantifying the proportional contributions of river water to the transpiration of riparian trees near a losing river. This integration of methods is more accurate than previous studies (Alstad et al., 1999; Zhou et al., 2017; White and Smith, 2020), which only considered river water as a direct water source of riparian trees without considering their distances from the riverbank and the extents of the lateral roots. The primary advantage of the combined method is that

it explicitly identifies the direct and indirect water sources of riparian trees based on the distance from the riverbank, the extent of lateral roots, and the process of riparian deep-water recharging by the river. To ensure the convergence of the MixSAIR model, both the trace plots and three diagnostic tests (i.e., Gelman–Rubin, Heidelberger–Welch, and Geweke) were adopted (Stock and Semmens, 2013). Besides, the MixSIAR model explicitly considers the uncertainties in the isotopic values and the estimates of source contributions compared to the simpler linear mixing models (Stock and Semmens, 2013; Ma et al., 2016). The strength of the newly proposed multi-iteration method is that it can determine the total contributions of the indirect river water source to the transpiration of riparian trees. The multi-iteration will not stop until there is no significant difference between the results of the last two iterations. This reduces the calculation errors of the RWC to the transpiration of riparian trees.

However, there are still some limitations that should be further investigated in future studies. First, the riparian deep-water sources were identified using the water isotopic data collected in campaigns taking place at an interval of about one month. The riparian soil water movement was complex, and the water stable isotopes might not be uniform between the two campaigns along the losing river. Nevertheless, the isotopic changes from t-1 to t (such as fractionation during this period) were negligible when calculating the contribution of upper soil

water (i.e., in the 0-80 cm or 0-170 cm layers) at t-1 to deep moisture (i.e., soil water in the 80-170 cm layer or groundwater). Assuming the isotopic uniformity over such a time interval may cause uncertainties in estimating the RWC to the transpiration of riparian deep water. Second, we supposed that the contributions of old river water (before initial time (t-1)) to riparian in-situ deep water were identical to the contributions of current river water (during the observation period between t-1 and t) to riparian in-situ deep water. This can induce some uncertainties in the estimations of the RWC to riparian deep water and the RWC to the transpiration of riparian trees. To minimize this issue, water samples would need to be collected more frequently to quantify the contributions of river water to riparian deep water and tree transpiration. Third, we inferred the approximate lateral root extent based on the projected edge of the canopy of *S. babylonica*, which indicated that *S. babylonica* trees could not tap into the river or take up river water directly. However, the lateral roots of *S. babylonica* trees should be directly investigated in further research to confirm our inference. Fourth, the riparian WTD along the studied reach of Chaobai River (from Dam 5 to Dam 4) ranged from 0.2 m to 4.3 m in two studied years (these data have not been published yet). The selected site in this study was the most representative site since there was a significant water table variation (ranging from 0.3 m to 4.0 m) in the two studied years. However, the implications of quantifying the effects of river water on the water use of riparian trees in this study are only applicable to relatively shallow water table conditions (with the WTD less than 4 m). Further investigations should be conducted at deep-WTD sites to better understand and regulate river runoff and tree's water needs.

## 5 Conclusion

We presented a new iteration method in combination with the MixSIAR model and water stable isotopes ($\delta^2H$ and $\delta^{18}O$) to separate and quantify the proportional contributions of river water to the transpiration of riparian *S. babylonica* in the dry year of 2019 and the wet year of 2021 along a losing river in Beijing, China. We found that the infiltrating river water was exchanged with riparian mobile water quickly but was not completely mixed with waters held tightly in the fine pores. Riparian trees near the losing river generally extended roots into fine pores to access immobile water sources. The isotopic discrepancies between the fast-moving water flow and the immobile water taken up by the roots led to a small RWC (20.3%) to the transpiration of riparian trees. The water deficit in the dry year probably induced stomatal closure and a larger reduction in transpiration compared to the

photosynthesis of riparian trees, thus leading to an evident increase of leaf WUE compared to the wet year. The leaf WUE exhibited a negative correlation with the RWC to the transpiration of riparian trees but was positively linearly related to WTD ($p < 0.01$). Riparian *S. babylonica* trees maintained the highest leaf WUE and the lowest river water uptake proportion under deep groundwater conditions (with a WTD of 4 m). This suggests that rising groundwater levels may encourage riparian trees to increase the river water uptake and show a consumptive river-water-use pattern, which cannot be beneficial to the water resource management of a losing river that is under restoration by ecological water. This study provides valuable insights into riparian afforestation that is related to water use and ecosystem health.

**Data availability**: The data that support the findings of this study are available from the corresponding author upon request.

**Author contributions**: YL: Investigation, Methodology, Formal analysis, Writ–ng - original draft, Writ–ng - review & editing; YM: Methodology, Formal analysis, Conceptualization, Writ–ng - review & editing; XFS: Supervision, Writ–ng - review & editing, Project administration; QZ: Methodology. LXW: Writ–ng - review & editing.

**Competing interests**: At least one of the (co-)authors is a member of the editorial board of Hydrology and Earth System Sciences.

**Acknowledgments:** This work was supported by the National Natural Science Foundation of China (41730749) and the National Key R&D Program of China (2021YFC3201203). Sincere thanks go to Xue Zhang, Yiran Li, Lihu Yang, and Binghua Li for their assistance in experiments.

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

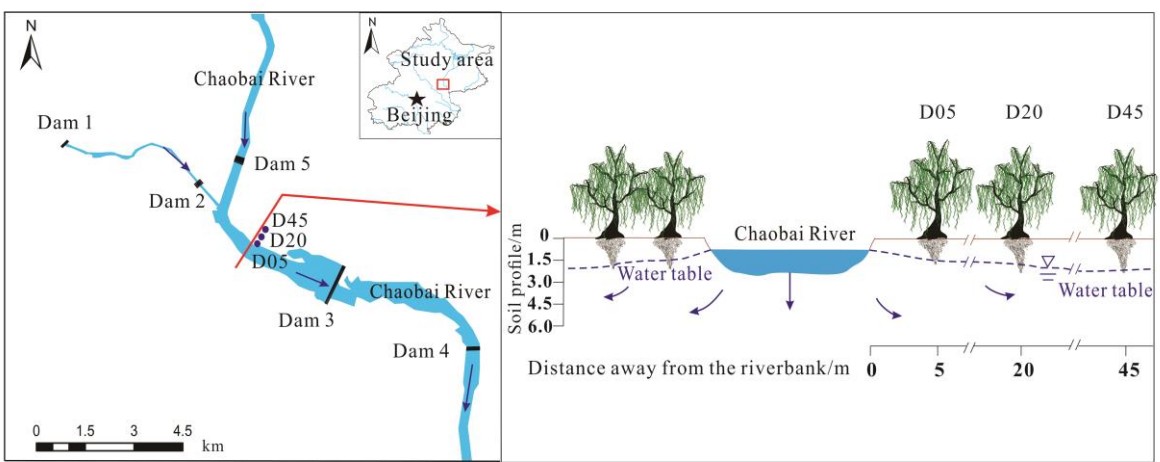

**Figure 1: Schematic diagram of the study area and the three sampling plots (D05, D20, and D45). D05, D20, and D45 are the plots at distances of 5 m, 20 m, and 45 m from the riverbank, respectively.**

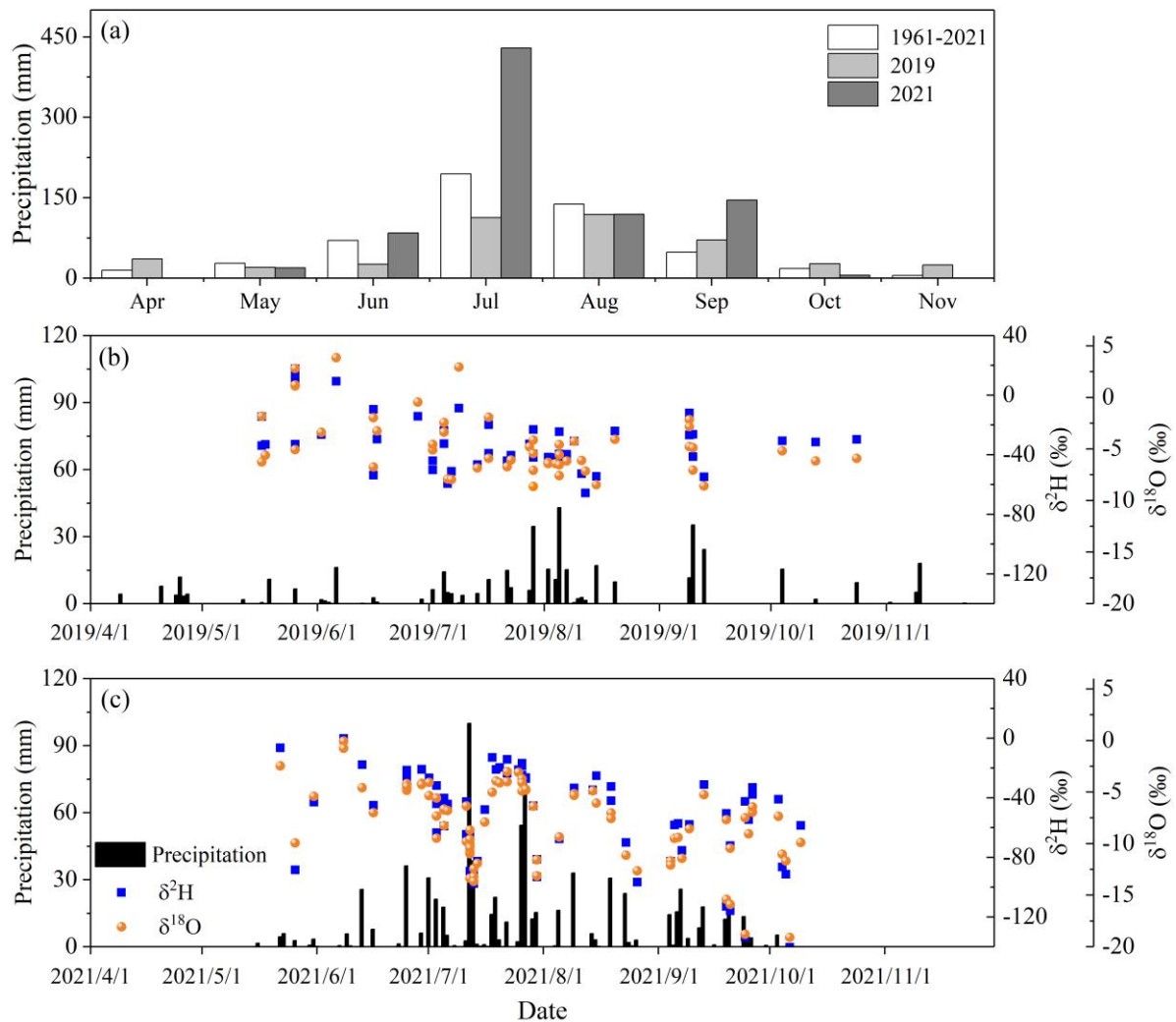


**Figure 2: Monthly average precipitation amount from 1961 to 2021 and monthly total precipitation amount for the observation years of 2019 and 2021 (a), daily precipitation amount and precipitation isotopes during 2019 (b), and 2021 (c).**

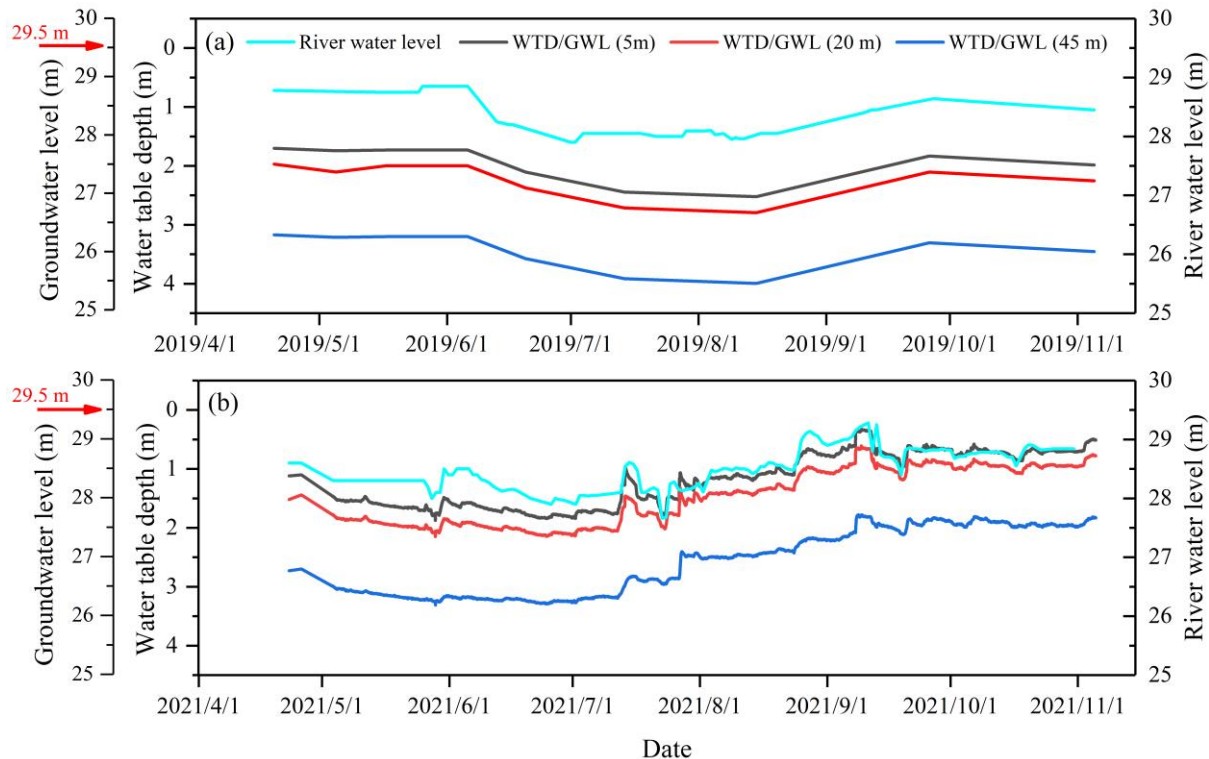


**Figure 3: Seasonal variations of the river water level and the water table depth (WTD)/ groundwater level (GWL) at distances of 5 m, 20 m, and 45 m from the riverbank during the observation period in 2019 (a) and 2021 (b). The red arrow indicates the riparian ground surface level (29.5 m). The riverbed level is 26 m.**


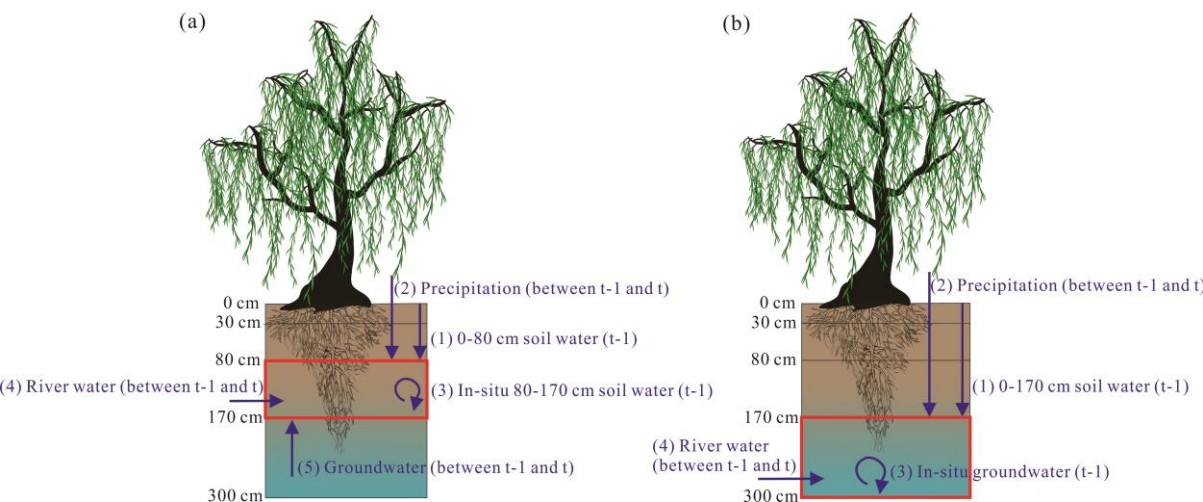

**Figure 4: Schematic diagram of potential water sources: riparian deep soil water in the 80−170 cm layer (a) and groundwater (b). The red box represents riparian deep soil water in the 80−170 cm layer in panel (a) and groundwater in panel (b), respectively. The dark blue arrow indicates different potential water sources of riparian deep water.**

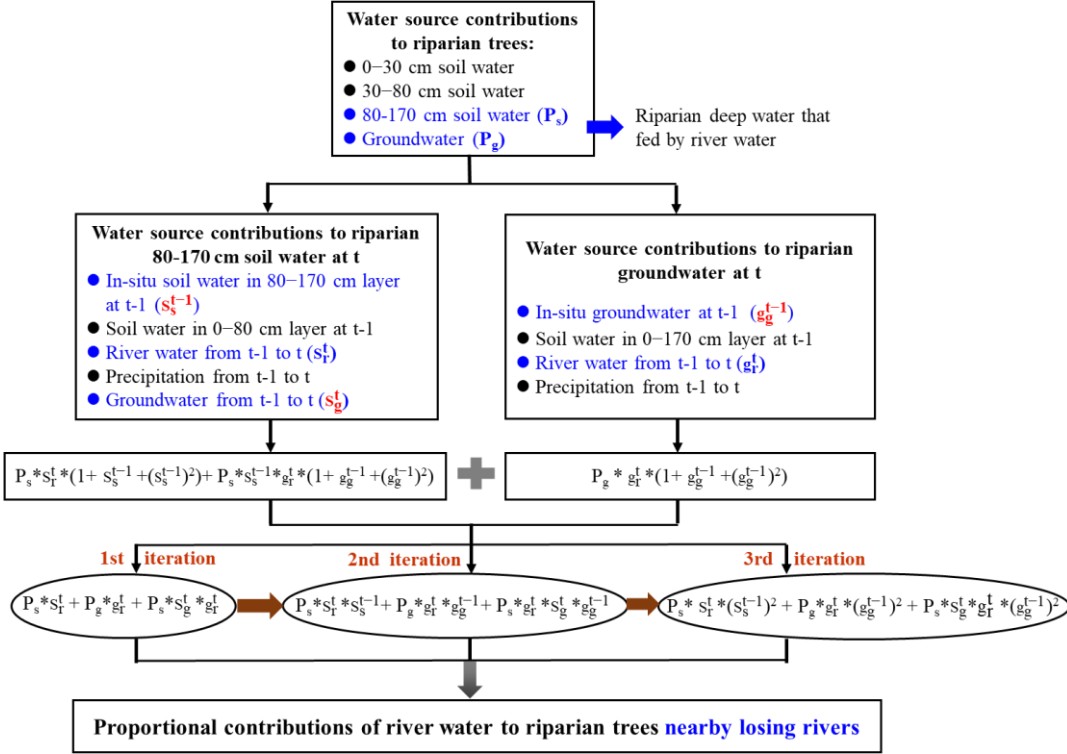

**Figure 5: Flowchart for quantifying the proportional contributions of river water to the transpiration of riparian trees. $P_s$ and $P_g$ denote the contributions of riparian deep soil water in the 80−170 cm layer and groundwater to the transpiration of riparian trees, respectively. $s_r^{t-1}$ and $g_r^{t-1}$ represent the proportional contributions of the old river water (before t-1) to riparian deep soil water in the 80−170 cm layer and groundwater, respectively. $s_s^{t-1}$, $s_r^t$, and $s_g^t$ signify the proportional contributions of in-situ soil water in the 80−170 cm layer at t-1, river water during t-1 to t, and groundwater during t-1 to t for riparian deep soil water in the 80−170 cm layer at t, respectively. $g_g^{t-1}$ and $g_r^t$ stand for proportional contributions of in-situ groundwater at t-1 and river water from t-1 to t to riparian groundwater at t, respectively.**

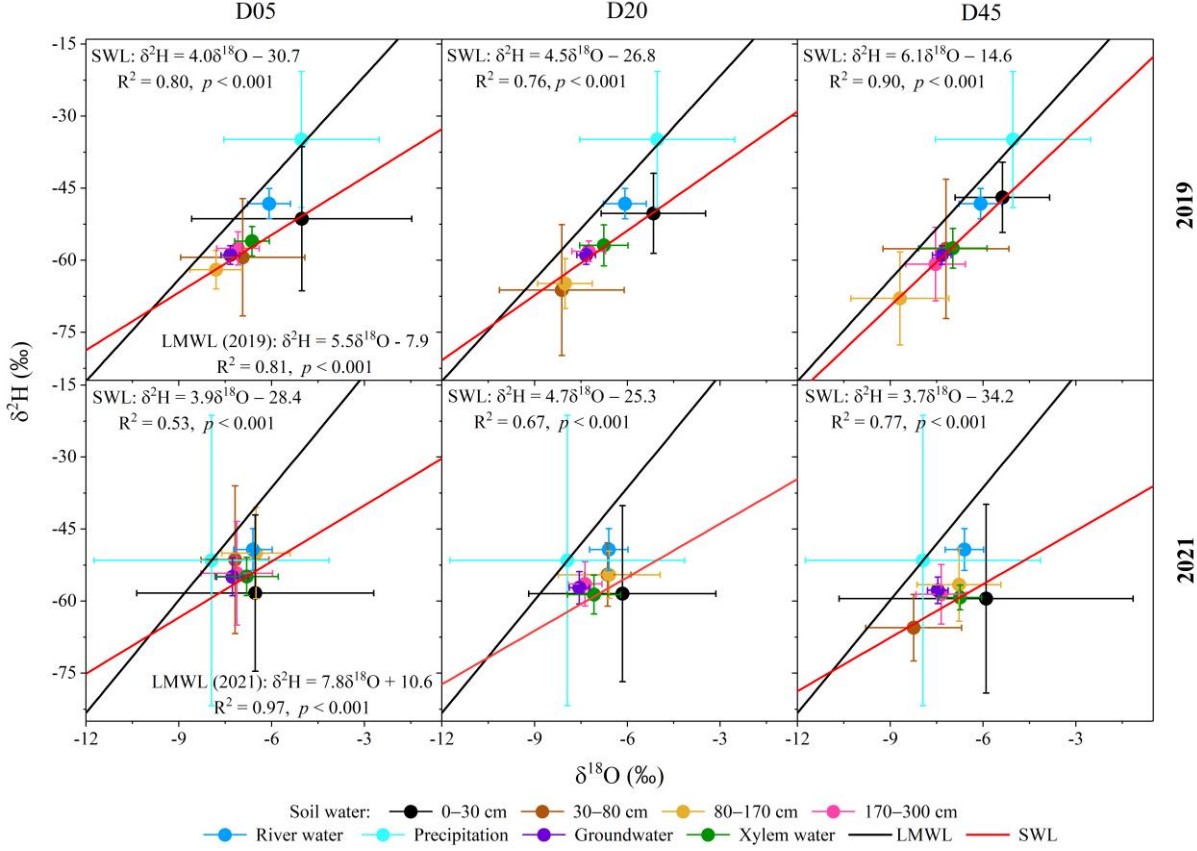

Figure 6: Dual-isotope ($\delta^2$H and $\delta^{18}$O) biplots of different water bodies in the three plots (D05, D20, and D45) for the observation years of 2019 and 2021. The local meteoric water line (LMWL) was determined for each year from the precipitation samples taken over each year. The soil water line (SWL) was determined for each year and each plot using the soil water samples taken over each year. D05, D20, and D45 are the plots at distances of 5 m, 20 m, and 45 m from the riverbank, respectively. The error bars indicate standard deviations.

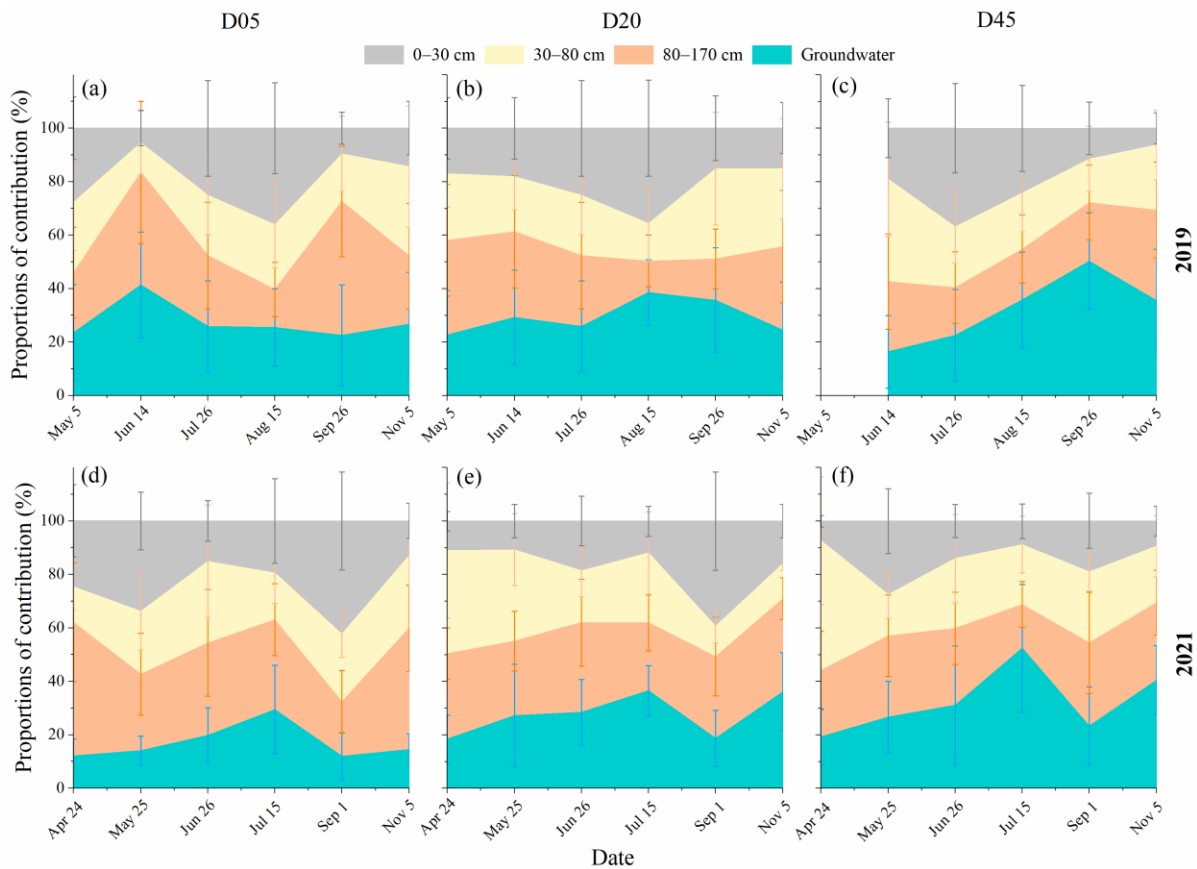

**Figure 7: Seasonal variations in the proportional contributions of soil water and groundwater to the transpiration of riparian trees in the three plots (D05, D20, and D45) for the observation years of 2019 (a−c) and 2021 (d−f). D05, D20, and D45 are the plots at distances of 5 m, 20 m, and 45 m from the riverbank, respectively. The error bars indicate standard deviations.**

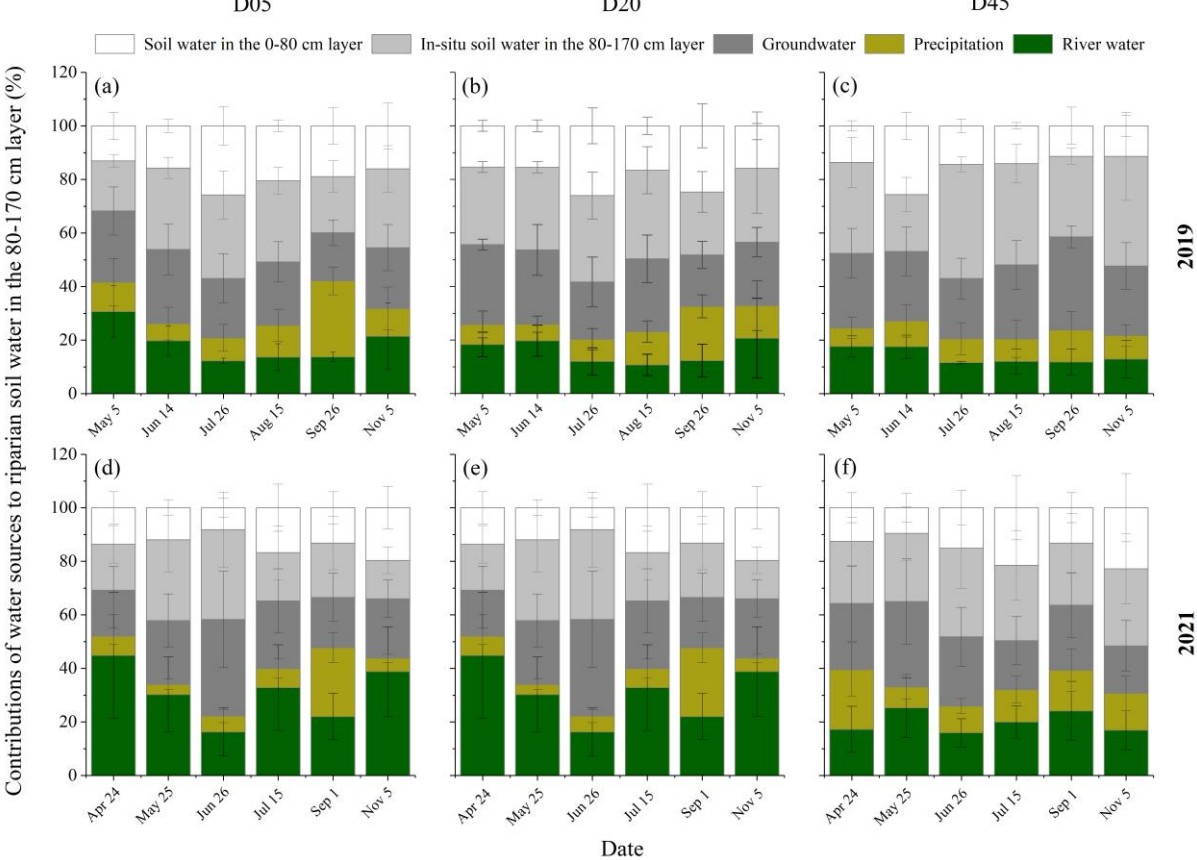

**Figure 8: Seasonal variations in the contributions of different water sources to riparian deep soil water in the 80−170 cm layer in the three plots (D05, D20, and D45) for the observation years of 2019 (a−c) and 2021 (d−f). D05, D20, and D45 are the plots at distances of 5 m, 20 m, and 45 m from the riverbank, respectively. The error bars represent standard deviations.**

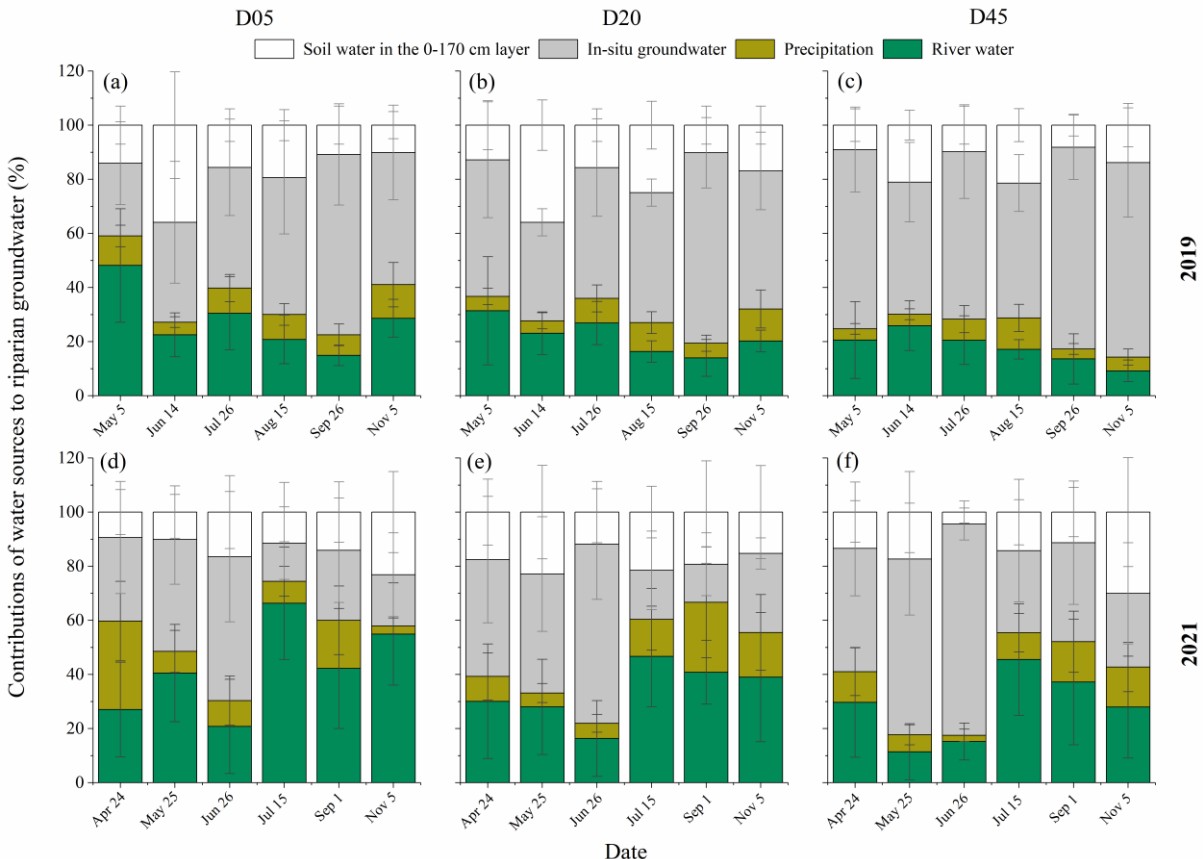

**Figure 9: Seasonal variations in the contributions of different water sources to riparian groundwater in the three plots (D05, D20, and D45) for the observation years of 2019 (a−c) and 2021 (d−f). D05, D20, and D45 are the plots at distances of 5 m, 20 m, and 45 m from the riverbank, respectively. The error bars indicate standard deviations.**

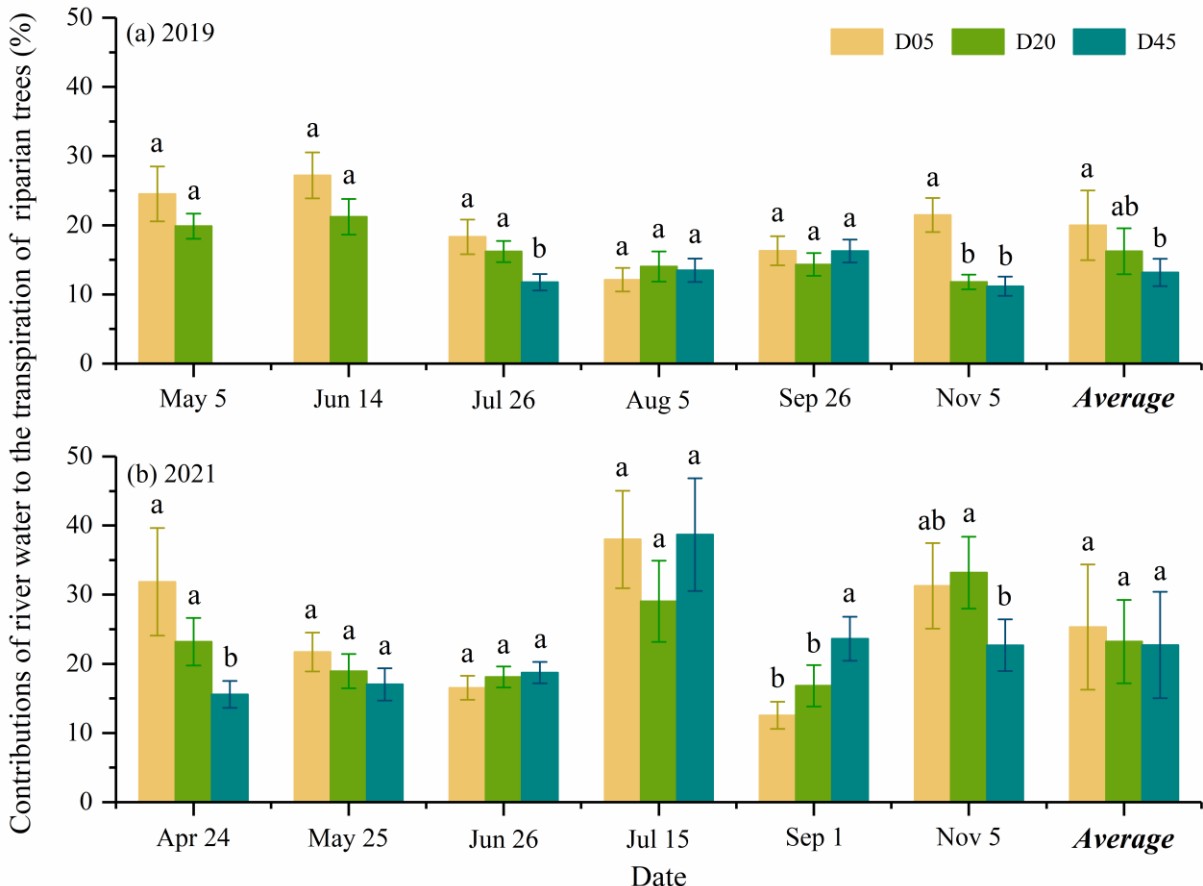

**Figure 10: Contributions of river water to the transpiration of riparian trees in the three plots (D05, D20, and D45) for each sampling campaign for the observation years of 2019 (a) and 2021 (b). Different letters show a significant difference in the river water contribution to the transpiration of riparian trees between three plots for each sampling campaign ($p < 0.05$). D05, D20, and D45 are the plots at distances of 5 m, 20 m, and 45 m from the riverbank, respectively.**

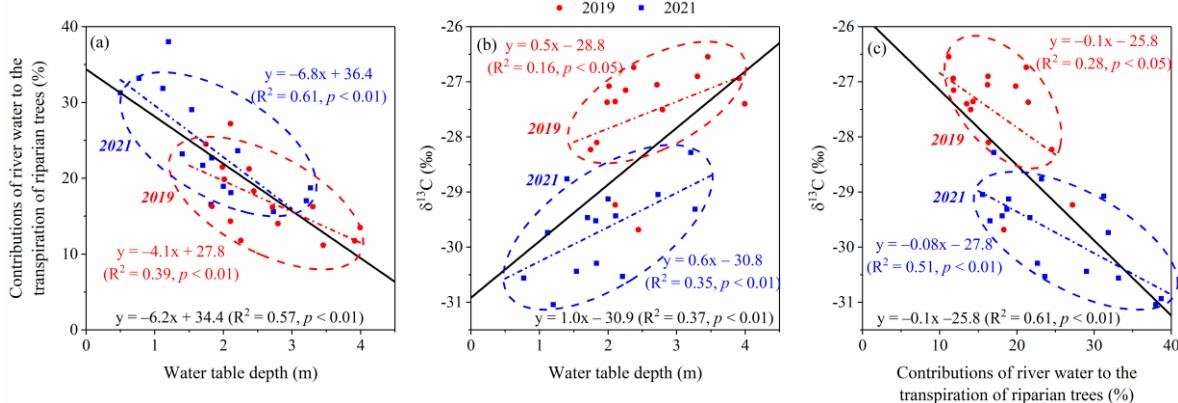

**Figure 11: Relationships between the contributions of river water to the transpiration of riparian trees and the water table depth (a), between the leaf δ¹³C values and the water table depth (b), and between the leaf δ¹³C values and proportions of river water contributions to riparian trees (c). The red line represents the linear relationship fitted by the monthly data in three plots in 2019, while the blue line represents the linear relationship fitted by the monthly data in three plots in 2021. The black line represents the linear relationship fitted by the monthly data in three plots in both years. The WTD, leaf δ¹³C values, and river water contributions to the transpiration of riparian *S. babylonica* are monthly data at each plot at a distance of 5 m, 20 m, and 45 m from the riverbank during the observation period in both years.**

**Table 1: The $^{222}$Rn values in river water, background groundwater and riparian groundwater in three plots (D05, D20, and D45), and the average residence time of recharged groundwater from river water ($T_{res}$, day) in 2021. The background groundwater indicates groundwater in aquifers more than 100 m away from the riverbank. The "negative $T_{res}$ values" were set to "0".**

| | River water | Background groundwater | Riparian groundwater | | |
| --- | --- | --- | --- | --- | --- |
| | | | D05 | D20 | D45 |
| $^{222}$Rn value (Bq/m$^3$) | 610.1 ± 212.3 | 7400 ± 35.4 | 494.5 ± 107.5 | 763.3 ± 118.3 | 787.4 ± 153.2 |
| $T_{res}$ (days) | 0 | Null | 0 | 0.13 ± 0.1 | 0.15 ± 0.13 |

Notes: D05, D20, and D45 are the plots at distances of 5 m, 20 m, and 45 m from the riverbank, respectively.

**Table 2: Leaf δ¹³C values of riparian *S. babylonica* in the three plots (D05, D20, and D45) during the observation period in 2019 and 2021.**

| | Leaf $\delta^{13}C$ value (‰) | | | | | | | |
|---|---|---|---|---|---|---|---|---|
| | 2019 | | | | | | | |
| | May 5 | Jun 14 | Jul 26 | Aug 15 | Sep 26 | Nov 5 | Mean | STD |
| D05 | −28.8 | −29.2 | −29.7 | −30.4 | −28.1 | −27.4 | −28.8 | 1.0 |
| D20 | −27.1 | −26.7 | −27.1 | −27.5 | −27.4 | −27.2 | −27.1 | 0.2 |
| D45 | Null | −27.2 | −26.9 | −27.4 | −26.9 | −26.5 | −27.0 | 0.3 |
| | 2021 | | | | | | | |
| | Apr 24 | May 25 | Jun 26 | Jul 14 | Sep 1 | Nov 5 | Mean | STD |
| D05 | −29.7 | −29.5 | −29.5 | −31.0 | −29.5 | −29.1 | −29.7 | 0.6 |
| D20 | −28.8 | −29.1 | −29.4 | −30.4 | −30.1 | −30.3 | −29.7 | 0.7 |
| D45 | −29.0 | −29.0 | −29.4 | −30.8 | −30.1 | −30.0 | −29.7 | 0.9 |

Note: D05, D20, and D45 are the plots at distances of 5 m, 20 m, and 45 m from the riverbank, respectively.

STD represents standard deviations.

**Table 3 Acronym dictionary**

| | |
|---|---|
| RWC | River water contribution |
| WUE | Leaf-level water use efficiency |
| WTD | Water table depth |
| T | Temperature |
| RH | Relative air humidity |
| $ET_0$ | Reference evapotranspiration |
| VPD | Vapor pressure deficit |
| SWC | Soil water content |
| IRIS | Isotopic ratio infrared spectroscopy system |
| IRMS | Isotope Ratio Mass Spectrometry system |
| VSMOW | Vienna Standard Mean Ocean Water |
| $C_{Water}$ | $^{222}$Rn concentration of the water samples |
| $C_{Air}$ | Air $^{222}$Rn concentration of the water samples |
| $C_{System}$ | Air $^{222}$Rn concentration of the measurement system |
| $V_{System}$ | The interior volume of the measuring set-up |
| $V_{Sample}$ | The volume of water sample |
| $T_{res}$ | The average residence time of recharged groundwater from river water |
| k | The $^{222}$Rn distribution coefficient of water/air |
| λ | The decay coefficient |
| $C_e$ | The $^{222}$Rn concentration of background groundwater when the equilibrium between radon production and decay is reached |
| $C_r$ | The $^{222}$Rn concentration of river water |
| $C_g$ | The $^{222}$Rn concentration of riparian groundwater |
| PWL | The potential water source line |
| $a_p$ | The slope of the PWL |
| $b_p$ | The intercept of the PWL |
| $PW_{excess}$ | The $δ^2H$ deviation of riparian tree xylem water from the PWL |
| $S_r$ | The total RWC to riparian deep soil water in the 80−170 cm layer (throughout the river losing-flow period since 2007) |
| $G_r$ | The total RWC to riparian groundwater (throughout the river losing-flow period since 2007) |

| | |
|---|---|
| $P_s$ | The contribution of riparian deep soil water in the 80−170 cm layer to riparian trees |
| $P_g$ | The contribution of riparian groundwater to riparian trees |
| $s_r^{t-1}$ | The proportional contribution of the old river water (before t-1) to riparian deep soil water in the 80−170 cm layer |
| $g_r^{t-1}$ | The proportional contribution of the old river water (before t-1) to riparian groundwater |
| $s_s^{t-1}$ | The proportional contribution of in-situ soil water in the 80−170 cm layer at t-1 to riparian deep soil water in the 80−170 cm layer at t |
| $s_r^t$ | The proportional contributions of river water from t-1 to t to riparian deep soil water in the 80−170 cm layer at t |
| $s_g^t$ | The proportional contribution of groundwater from t-1 to t to riparian deep soil water in the 80−170 cm layer at t |
| $g_g^{t-1}$ | The proportional contribution of in-situ groundwater at t-1 to riparian groundwater at t |
| $g_r^t$ | The proportional contribution of river water from t-1 to t to riparian groundwater at t |
| ANOVA | One-way analysis of variance |
| LMWL | Local meteoric water line |

845