# Peer review of "Quantifying river water contributions to the transpiration of riparian trees along a losing river: Lessons from stable isotopes and iteration method"

_Hydrology and Earth System Sciences, 2022_

## Author Comment (AC1)

We have carefully addressed all the comments made by Dr. Remy Schoppach and one anonymous reviewer on our manuscript (hess-2022-327) entitled "Quantifying river water contributions to riparian trees along a losing river: Lessons from stable isotopes and iteration method". The comments have helped us greatly improve the overall quality of the manuscript. The following is the point-point response to all the comments.

**Response to Anonymous Referee #1:**

**General comments:**

1.  Comment:

This manuscript investigates the river water contribution to riparian trees. It does it by (1) determining and quantifying the sources of riparian trees water in soil and groundwater, (2) determining the sources of deep soil water and groundwater (precipitation, in-situ soil and groundwater, river water…) to finally (3) determine the proportion of river water that fed the riparian trees. This is an important work to better understand how riparian trees can affect stream flow and how we can better manage riparian zones in order to both protect rivers and riparian vegetation.

The manuscript is well-organized but I found the flow difficult to follow due to the grammar and wording mistakes throughout the manuscript (especially discussion section). The uncertainties of the MixSIAR model and iteration method are not really discussed, the MixSIAR model is also not well-enough presented in the method section. The discussion section is too light, I found that the discussion of the potential processes was too limited, as well as how do these results compare with other studies and why.

The figures are very good, there are just some grammar/wording mistakes in some captions.

**Response:** *Thank you for the positive comments and insightful suggestions. We have double-checked and revised the grammar and wording mistakes throughout the manuscript (including the figures). The iteration method and radon ($^{222}$Rn) concentration are supplemented in the Introduction section. We have added more information on the MixSIAR model to the M&M section (See the responses to Comments #14 and #15), and supplemented hydro-meteorological data (e.g., VPD, net radiation, relative humidity and temperature) to the Result section (Fig. S1). We substantially*

*discussed the uncertainties of this model in the Discussion section. Especially for the major concerns, we have thoroughly revised the Discussion and discussed the potential processes. We have also compared our results with previous work and clearly explained the mechanisms.*

*The Discussion has been revised from the following three aspects. Firstly, we added the strengths/weaknesses and implications of the MixSIAR model and the iteration method. Secondly, we discussed the potential processes of the RWC to transpiration flux of riparian trees, and further explained the effects of the distance from the stream as well as dry/wet year on RWC to transpiration flux of riparian trees. Finally, we discussed the link between RWC/WUE/WTD and its implications on management of rivers and riparian vegetation. The following is the revised Discussion section:*

*"4. Discussion*

*4.1 The strengths/weaknesses and implications of the MixSIAR model and the iteration method*

*Most previous studies considered river water as a direct water source for riparian trees, which may lead to large errors in estimating the RWC to transpiration flux of riparian trees. In comparison, the newly proposed iteration method and the MixSIAR model coupled with water stable isotopes are particularly useful for separating and quantifying the proportional contributions of indirect river water source to transpiration flux of riparian trees nearby a losing river. The primary advantage is that it explicitly identifies the direct and indirect water sources of riparian trees according to the distance from the riverbank, the extents of lateral roots, and the process of river recharging riparian deep water. Furthermore, the MixSIAR model has considered the uncertainties in isotope values and the estimates of source contributions compared with the simpler linear mixing models (Ma et al., 2016; Stock and Semmens, 2013). Both the trace plots and three diagnostic tests are used to check that the MixSAIR model has converged (Stock and Semmens, 2013). The strength of the multi-iteration method is that the total indirect RWC to transpiration flux of riparian trees nearby a losing river can be determined. The multi-iteration stops until there is no significant difference between the results of the last two iterations, which reduces the uncertainty of the calculated RWC to transpiration flux of riparian trees.*

*Although the MixSIAR model in combination with the iteration method were successfully used to quantify the indirect RWC to those riparian trees along a losing river; Nevertheless, it will not be available if the groundwater level is higher than river water level and groundwater feeds river (i.e., "gaining" river). Because the riparian trees along the gaining river mainly rely on the riparian*

*deep water recharged by precipitation or upland infiltration via downhill flow rather than the lateral river water seepage (Miguez-Macho and Fan, 2021). Secondly, we assumed that the contributions of old river water (before initial time (t-1)) to riparian in-situ deep water were identical with those contributions of current river water (during the observation period between t-1 and t) to riparian in-situ deep water in this study. This might increase uncertainties on the estimations of the RWC to riparian deep water and RWC to riparian trees. Additional water samples need to be collected before the initial time to determine the contributions of old river water to riparian deep water.*

*4.2 The RWC to riparian trees and the effects of the distance from the stream on RWC*

*The deep-rooted riparian trees nearby a losing river were identified to use a small proportion of river water (less than 25%) for transpiration in this study (Fig. 10). It agreed well with previous work that the lateral roots of riparian trees further than 5 m away from the riverbank rarely took up river water when their outer projected edge of canopy (less than 5 m in our study) were out of reach of the river (Busch et al., 1992; Thorburn and Walker, 1994). Moreover, the ecohydrology separation reported in previous studies (Brooks et al., 2010; Evaristo et al., 2015; Allen et al., 2019; Sprenger et al., 2019) might result in considerable isotopic discrepancies between fast-moving water flow and immobile water for plant water uptake. The residence time of recharged groundwater from river water was extremely short (no more than 0.28 days) (Table 2). Only one third of the riparian groundwater was replaced by the infiltrating river water with an immobile groundwater proportion of 46.5% (Fig. 9). This probably indicated that the river water recharged mobile groundwater quickly but could not completely replace water held tightly in the soil pores (Brooks et al., 2010; Evaristo et al., 2015; Allen et al., 2019). It was consistent with Sprenger et al. (2019) who found that the lateral seepage of river water or rising water table could briefly saturate riparian soils but not entirely replace/flush immobile waters or isotopically homogenize different water pools. On the other hand, several recent studies showed that the tree species even phreatophytic/deep-rooted trees predominantly extended roots into fine pores to take up immobile soil water (Evaristo et al., 2015; Maxwell and Condon, 2016; Evaristo et al., 2019). As mentioned above, the immobile water could not be completely replaced by infiltrating river water, which eventually resulted in a small contribution of indirect river water to deep-rooted riparian trees.*

*However, our results are quite different from Alstad et al. (1999) who found that approximately 80% of the transpiration water of riparian Salix trees came from a losing river on the northeast side*

of Rocky Mountain National Park, Colo. This is probably because that Alstad et al. (1999) only considered the river water and precipitation as potential water sources for riparian Salix. In fact, the river water seeps into the saturated/vadose zone across riparian riverbank and it is usually taken up by riparian trees in the form of river-recharged soil water/groundwater. In our study, we separated the contributions of indirect river water source (i.e., river-recharged deep soil water in the 80−170 cm layer and groundwater) for riparian trees. However, the RWC to riparian Salix trees calculated by Alstad et al. (1999) included all the proportional contributions of indirect river water, in-suit soil water as well as in-suit groundwater for riparian Salix trees.

In this study, the RWC to riparian S. babylonica trees in wet 2021 was 1.4 times higher on average than in dry 2019 (Fig. 10). This is mainly because that the higher water table in wet year induced higher RWC to riparian deep water (including deep soil water in the 80−170 cm layer and groundwater) and consequently higher indirect RWC to riparian phreatophyte trees compared with dry year. Although there was no significant difference in the deep water (below the 80 cm layer) contributions to riparian trees between three plots, we observed the substantial effect of the declining water table with increasing distance from the riverbank on the reduced indirect RWC to riparian trees in dry 2019 (Fig. 10). Therefore, the temporal and spatial variabilities of the RWC to riparian S. babylonica trees were generally attributed to the various RWCs to riparian deep water rather than the water use patterns of riparian trees. Our result contrasts with the previous study by Qian et al. (2017) who reported a significant increase of the RWC to G. biloba trees in response to the water table decline. It was ascribed to that riparian G. biloba had a dimorphic root system and shifted their main water sources from shallow soil layer to deeper soil layer. Nevertheless, the potential root growth rate of riparian phreatophyte S. babylonica trees can reach 1-13 mm/day, which allows the riparian S. babylonica trees to remain in contact with a rising/declining water table and keep constant water uptake proportions from deep strata below the 80 cm depth in this study (Naumburg et al., 2005)."

4.3 The link between RWC/WUE/WTD and its implications

The water uptake patterns of riparian S. babylonica trees generally remained the characteristics of phreatophyte. We observed that the leaf WUE of all S. babylonica trees across three plots in both dry and wet years were negatively correlated with the indirect RWC to riparian trees and positively related to the WTD, respectively (Figs. 10, 11b, and 11c). These relationships

*are in consistent with previous studies (Behzad et al., 2022; Cao et al., 2020; Ding et al., 2020). Higher leaf WUE associated with lower RWC to riparian trees and lower groundwater levels are likely due to that the water stress restricts the stomatal conductance and further reduces transpiration rate of riparian trees. Specifically, dry 2019 was characterized as higher water demand (indicated by higher VPD) and lower water availability compared with wet 2021, but the energy resource (indicated by net radiation) for riparian trees was similar between two years (Figs. S1 and S2). We supposed that the water limitation rather than energy limitation regulated the leaf-level stomatal conductance of riparian S. babylonica trees. The high water demands but low river water availability in dry year probably resulted in stomatal closure of riparian trees to minimize the water loss, which could eventually lead to a decrease of transpiration rate and even photosynthetic rate (Behzad et al., 2022; Fabiani et al., 2021). Aguilos et al. (2018) further found that the water stress would enhance radiation-normalized WUE, because the lack of water availability induced a more reduction in transpiration than photosynthesis. On account of no difference in the average net radiation between dry and wet years, the lower river water availability in dry year probably resulted in an increase of the leaf WUE. It can be inferred that riparian S. babylonica trees took up more river water and probably showed a consumptive water use strategy in wet year compared to dry year. This agreed well with previous studies that the woody plants showed lower leaf WUE and consumptive water-use patterns in rainy season, while they showed higher leaf WUE and conservative water-use patterns with lower soil water availability in dry season (Behzad et al., 2022; Cao et al., 2020; Horton and Clark, 2001). However, the consumptive water use strategy of riparian trees could result in an overconsumption of the river water, which should be avoided in the riparian zone of a losing river.*

*It was evident that the WTD played a critical role in river water acquisition of riparian trees nearby losing rivers (Mensforth et al., 1994; Horton and Clark, 2001; Qian et al., 2017; Zhou et al., 2017). We observed that the declining water table linearly reduced the proportional contributions of river water sources for riparian trees, which consequently led to a proportional increase of leaf WUE (Fig. 11a). It was consistent with Horton and Clark (2001) who found an exponential growth function between the leaf WUE of riparian Salix gooddingii and WTD. As mentioned above, we emphasized the key role of reduced water availability to account for the decreasing transpiration rate' control on enhancing leaf WUE in this study. Nevertheless, there were*

*some controversial views that the leaf WUE firstly increased and then decreased with the increasing WTD (Antunes et al., 2018; Xia et al., 2018). This could be due to that riparian trees could tolerate declining water availability only up to a species-specific threshold, beyond which xylem cavitation and even crown mortality occurs (Naumburg et al., 2005). These indicated that the optimal WTDs for plant species were corresponding to the highest leaf WUE, under which plant species could consume lower transpiration water to maximize $CO_2$ assimilation. The break point of WTD was not observed in this study, suggesting that further investigations should be conducted under deeper water tables (> 4 m) to quantify the optimal WTD and further optimize the riparian plant-water relations.*

*Our results have important implications for untangling the trade-off relationships between riparian tree water use and river runoff management. Considering various water and energy resources as well as the leaf WUE of riparian trees, the relative amount of RWC to riparian trees has been compared between dry and wet year to investigate the effects of river water availability on physiological characteristics of riparian trees. The riparian S. babylonica trees likely remained the highest WUE (i.e., utilize lower transpiration water to maximize $CO_2$ assimilation) as well as the lowest river water uptake proportion under the lowest water table condition (with the WTD of 4 m). In this study, the lower the groundwater level is, the more beneficial it is to optimize the riparian plant-water relations. Therefore, the relationships between the RWC to riparian trees, leaf-level physiological characteristics (e.g., leaf WUE) and hydro-meteorological conditions are critical and helpful for the better protection of the riparian forest while maintaining sustainable river runoff."*

[Figure]

*Figure 9: Seasonal variations in the different water source contributions to riparian groundwater in the three plots (D05, D20, and D45) for observation years 2019 (a–c) and 2021 (d–f). D05, D20, and D45 are the plots at distance of 5 m, 20 m, and 45 m away from the riverbank, respectively. The error bars indicate standard deviations.*

[Figure]

*Figure 10: River water contributions (RWCs) to riparian trees in the three plots (D05, D20, and D45) for each sampling campaign during the observation period in 2019 (a) and 2021 (b). Different letters show a significant difference in the RWC to riparian trees between three plots (p < 0.05). D05, D20, and D45 are the plots at distance of 5 m, 20 m, and 45 m away from the riverbank, respectively.*

[Figure]

*Figure 11: Relationships between river water contributions to riparian S. babylonica and the water table depth (a), between leaf $\delta^{13}C$ values and the water table depth (b), and between leaf $\delta^{13}C$ values and river water contributions to riparian S. babylonica (c).*

[Figure]

*Figure S1. Daily mean temperature (°C) and daily mean vapor pressure deficit (VPD, kPa) during the observation period in 2019 (c) and 2021 (c). Daily reference evapotranspiration (ET$_0$, mm/day) and daily mean net radiation (W/m$^2$) during the observation period in 2019 (c) and 2021 (c).*

[Figure]

*Figure S2: Seasonal variations of soil water content (SWC) in the 0−30 cm, 30−80 cm, 80−170 cm, and 170−300 cm layers on twelve campaigns (a−h) for the observation years 2019 (a−d) and 2021 (e−h). Different letters (a, b, and c) show a significant difference in the SWC between three plots (p < 0.05). D05, D20, and D45 are the plots at distance of 5 m, 20 m, and 45 m away from the riverbank, respectively.*

**Specific comments:**

INTRODUCTION

2.   Comment:

Line 41: I feel that the jump to paragraph 2 (contribution of river water to riparian trees) is too quick, maybe have a small paragraph beforehand presenting the different sources of water for riparian trees (groundwater, soil water, river water?) first?

**Response:** *We have added a sentence "The potential water sources of riparian trees along a losing river are typically considered as a mix of soil water at different depths, groundwater, and river water (Alstad et al., 1999; White and Smith, 2020)." at the beginning of the second paragraph. Then it shifts to introduce the debate on whether river water is a potential water source for riparian trees or not and how it becomes available to plants.*

3.   Comment:

Line 41: Can you be more specific about "data comparison"? It is unclear.

**Response:** *The "data comparison" has been specified as the direct comparison of stable isotope values between stem water and different source waters. This method is similar to the "graphical inference" method, which could determine the possible root water uptake depth when the stem water isotope is same as the isotopic value of soil water at a certain depth. However, both the "data comparison" and "graphical inference" methods cannot quantify the proportional contribution of different water sources to plants. We have deleted the words "data comparison" and "graphical inference", and changed this sentence to "The statistical two- or multi-source linear mixing models (Ehleringer and Dawson, 1992; Alstad et al., 1999) and Bayesian mixing models (MixSIR, SIAR, SISUS, MixSIAR) (Ma et al., 2016; Wang et al., 2019b; White and Smith, 2020; Li et al., 2021) accompanied with stable water isotopes ($\delta^2H$ and $\delta^{18}O$) have been widely used to estimate the RWC to riparian trees.".*

**M&M**

4. Comment:

Lines 112-125: I think you should develop on the storage in the field and until you put the samples in the fridge (where/in what did you stored the samples? what method did you use to limit evaporation for the water samples?), this is important and not really described in this section. Also, use the (full) technical name of the tools you used in the field for sampling (e.g., "hydrophore"?) or storage of samples (e.g., "brown bottle" is unclear).

**Response:** *Thanks for your professional suggestions. We have added description of the field storage procedure. All the samples (river water, groundwater, soil samples, plant stem samples, plant leaf samples and precipitation) were put into the refrigerating box with several ice bags to avoid evaporation effects in the field. And all the samples were delivered to the laboratory and stored at 4°C/-10°C in the refrigerator. We have provided the full technical names of the tools used in the field for sampling or storage of samples. For example, we have changed "hydrophore" to "Perspex hydrophore water sample collector with capacity of 1L". The "brown bottle" has been changed to "100-ml brown glass vial".*

5. Comment:

Line 115: I am not sure what is it, maybe be more specific and use the full technical name? Maybe "plexiglass hydrophore water sample collector"? But I am not sure about what you used.

**Response:** *We have changed the "hydrophore" to a more specific technical name "plexiglass hydrophore water sample collector with capacity of 1L".*

6. Comment:

Line 115: Why 135 precipitation samples? Before you said precipitation was sampled during the 12 campaigns, I would clarify this point. Also, what was the frequency of sampling? And the location?

**Response:** *We apologize for the oversight. In fact, precipitation was sampled after each precipitation event and not merely collected during the 12 campaigns. A total of 135 precipitation samples were collected throughout the whole years of 2019 and 2021 in order to get the local meteoric water line (LMWL) in the dual-isotope plot. We have changed this sentence to "Precipitation was sampled after each precipitation event via a device consisting of a funnel, a polyethylene bottle and a ping-pong ball. And a total of 135 precipitation samples were collected*

*throughout the whole years of 2019 and 2021."*

7.  Comment:

Lines 120-121: So 3 trees for each plot?

**Response:** *One tree was sampled for each plot. We have changed this sentence to "One riparian S. babylonica tree was chosen in each plot for $\delta^2 H$ and $\delta^{18}O$ measurements in xylem water as well as $\delta^{13}C$ analysis in plant leaves. The mean diameter at breast height of three sampled trees was 28.6 ± 4.4 cm."*

8.  Comment:

Line 122: What do you mean by "several"? Was is not 3/plot? Did you take several samples in the same tree? I would use "sampled" rather than "cut". How did you sampled the stem? What tool did you use? Also, correct to "[…], we removed the bark.".

**Response:** *"Several stems" means that five mature and suberized stems were taken from the same tree in each plot. We have changed "cut" to "sampled". An averruncator with the length of 5 m was used to sample the stem. We have changed the confusing sentence "Several suberized stems were firstly cut from riparian S. babylonica, removed the bark and phloem, and then stored at −10 ℃ until water isotope analysis." to "Five mature and suberized stem samples were taken from the same riparian S. babylonica tree in each plot using an averruncator with the length of 5 m. We removed the bark and phloem of the sampled stems, and then put the remaining xylem samples into a 12-ml brown glass vial sealed with the parafilm."*

9.  Comment:

Line 123: Maybe specify: "stored the remaining xylem at.."? Where/in what did you store it? Glass vial?

**Response:** *We have changed this sentence to "We removed the bark and phloem of the sampled stems, and then put the remaining xylem samples into a 12-ml brown glass vial sealed with the parafilm. The remaining xylem samples were stored in a refrigerating box with several ice bags in the field. Then the xylem samples were transported to the laboratory and kept in a refrigerator at −10 ℃ before water extraction and isotope analysis."*

10. Comment:

Line 124: In what did you stored the leaves before drying them?

**Response:** *The mature leaf samples were stored in a refrigerating box with several ice bags in the field until they were transported to the laboratory. The mature leaves were oven-dried at 65 °C for 72 h on the day of sampling to avoid the effects of microorganisms.*

11. Comment:

Lines 126-130: This is a great section with the details needed (sampling method, storage). Can you clarify: you sampled soil only at 1 location for each plot, right? I would add a reference for the oven-dry method.

**Response:** *Yes, we collected the soil sample only at one location for each plot. We have clarified that "Soils at depths of 0−5, 5−10, 10−20, 20−30, 40−60, 60−80, 90−110, 150−170, 190−210, 250−270, and 280−300 cm in one soil profile nearby the selected S. babylonica trees were sampled by a power auger (CHPD78, Christie Engineering Company, Sydney, Australia)." We have added two references for the oven-dry method as follows "One part of each soil sample was put into a 12-ml brown glass vial and stored at −10 °C before water stable isotope analysis, and the other part was packed into an aluminum box for gravimetric soil water content (SWC) measurement via the oven-drying method (Li et al., 2021; Wang et al., 2019b)."*

*References:*

*Li, Y., Ma, Y., Song, X. F., Wang, L. X., and Han, D. M.: A $\delta^2H$ offset correction method for quantifying root water uptake of riparian trees. Journal of Hydrology. 593, 125811, doi:10.1016/j.jhydrol.2020.125811, 2021.*

*Wang, J., Fu, B., Lu, N., Wang, S., and Zhang, L.: Water use characteristics of native and exotic shrub species in the semi-arid Loess Plateau using an isotope technique. Agriculture, Ecosystems & Environment. 276, 55-63, 2019b.*

12. Comment:

Line 150: Do you have a reference to support this choice for k?

**Response:** *We have added the supporting reference for the parameter "k" (Clever, 1985).*

*The diffusion coefficient 'k' indicates the $^{222}$Rn concentration relationship between a watery phase and the air volume existing above. This relation is dependent on the temperature. With falling temperatures, the quantity of radon soluble in water increases. For example, the diffusion coefficient 'k' increases when temperature drops. The dependency for temperature of the diffusion coefficient for the transition of the phase water/air is presented in Fig. A1.*

*The influence of the radon diffusion coefficient 'k' is low in the temperature range between 10° C and 30° C. A diffusion coefficient k of 0.26 can be applied within the specified temperature range for a typical room temperature of 20° C (Fig. A1).*

[Figure]

*Figure A1. Temperature dependency of the diffusion coefficient 'k' (Clever, 1985)*

*References:*

*Clever, H. L.: Solubility Data Series. Vol.2, Krypton-, Xenon, Radon Gas Solubilities, p. 463-468, Pergamon Press, Oxford, 1985.*

13. Comment:

Lines 168-169: So you calculated the average isotopes values for each of the four layers? You measured soil water isotopes at 11 depths.

**Response:** *In this study, we measured soil water isotopes at 11 depths in the three plots. In order to reduce errors in the analytical procedure, four soil layers (0−30, 30−80, 80−170, and 170−300 cm) were divided according to the seasonal variations in SWC, water isotopes and WTD to identify the main root water uptake depth of riparian trees. The average and standard deviation (SD) isotopes values for each of the four layers (source data) were input to the MixSIAR model.*

14. Comment:

Line 175: I don't understand "set as the mixture data", please clarify. Maybe more info on the MixSIAR model would help, I find it a bit frustrating for the reader to have to look in other papers. It would also help to understand why you talk about sampling times later in the text (lines 184-188, for example). Maybe have a separate section just to present the model (before 2.4.1.)?

Response: *We have added more detailed information on the MixSIAR model before section 2.4.1 "MixSIAR model is a Bayesian mixing model which could apply stable isotope data to quantify the proportions of source contributions to a mixture (Stock and Semmens, 2013). The input data of the MixSIAR model include mixture data, source data, and discrimination data. In this study, the mean and standard deviation (SD) of isotopic values for each of the riparian tree water sources were input as source data into the MixSIAR model, while the raw mixture data (the measured isotopic values for stem water) were directly input into the MixSIAR model. The discrimination data for both $\delta^2H$ and $\delta^{18}O$ were set as zero, because the input $\delta^2H$ and $\delta^{18}O$ values in the MixSIAR were non-fractionated or $\delta^2H$-corrected (See the 2.4.1 and 2.4.2 sections). The Markov Chain Monte Carlo parameter was set to "very long" run length to ensure the highest accuracy. We used both the trace plots and three diagnostic tests (i.e., Gelman–Rubin, Heidelberger–Welch, and Geweke) to decide whether the MixSIAR model has converged or not (Stock and Semmens, 2013). The mean and SD of the different source contributions (median values) to each mixture were finally estimated in the MixSIAR model."*

15. Comment:

Lines 185 and 187: So when you say "soil water in 0-80 cm" and "soil water in 0-170 cm": does the model takes the layer as a whole or does it still account for the different soil layers 0-30, 30-80, 80-170 cm? And how do you integrate the soil water isotopes measured at 11 soil depths in the model? I think more information about that is needed in the manuscript (especially for the readers - like me - who are not familiar with the model).

Response: *When we state "soil water in 0-80 cm" and "soil water in 0-170 cm", it still accounts for the different soil layers in 0-30 cm, 30-80 cm, and 80-170 cm in the MixSIAR model. The proportional contributions of soil water in the 0-30 cm, 30-80 cm, and 80-170 cm layer for riparian deep water were separately estimated in the MixSIAR model. We finally add up the proportional*

*contributions of different soil layers.*

*More information for integrating the soil water isotopes measured at 11 soil depths in the model has been added as follows: "The average soil water isotopes values at depths of 0−5, 5−10, 10−20, and 20−30 cm were calculated for the 0-30 cm soil layer, because water isotopes went through strong evaporation and the SWC varied significantly seasonally. The soil water isotope values at depths of 40−60 and 60−80 cm were averaged for the 30-80 cm soil layer, and those values at 90−110 and 150−170 cm depths were averaged for the 80−170 cm soil layer as the water isotopes and SWC were relatively stable. The average isotopes values of soil water at deep depths (190−210, 250−270, and 280−300 cm) were calculated for the 170-300 cm soil layer, which varied with the fluctuations of water tables."*

16. Comment:

Line 194: Can you provide a reference for the decay coefficient?

**Response:** *We have added a reference for the decay coefficient as "where $\lambda$ represents the decay coefficient (0.181 day$^{-1}$) (Hoehn and Von Gunten, 1989)."*

*Reference:*

*Hoehn, E. and Von Gunten, H. R.: Radon in groundwater: A tool to assess infiltration from surface waters to aquifers. Water Resources Research. 25(8), 1795-1803, 1989.*

17. Comment:

Line 195: How was this value determined?

**Response:** *We have added more information about the determination of $C_e$. The $C_e$ represents the $^{222}$Rn concentration of background groundwater when the equilibrium between radon production and decay is reached. The measuring $^{222}$Rn concentration of groundwater in aquifers more than 100 m away from the riverbank remains constant in this study (with an average value of $7400.0 \pm 35.4$ Bq/m$^3$), suggesting that $C_e$ can be defined as 7400.0 Bq/m$^3$.*

18. Comment:

Lines 205-222: It is an important method here but I found this part difficult to follow. There are A LOT of symbols, maybe use a table instead of the 7 lines of text so it's more readable? I also did

not remember what was Ps and Pg, I would explain these terms here again. I would suggest to change some of the terms, or try to include the time as subscript (t-1, t) so it's easier to follow.

**Response:** *This is a good comment. We have simplified this equation to make it more readable. The Eq (4) has been changed to:*

*"$RWC = P_s * S_r + P_g * G_r$*

$$= P_s*(s_r^t + s_r^{t-1}) + P_g*(g_r^t + g_r^{t-1})$$

$$= P_s*(s_r^t + s_r^t*s_s^{t-1} + s_r^t*(s_s^{t-1})^2 + s_g^t*g_r^t + s_g*g_r^t*g_g^{t-1} + s_g^t*g_r^t*(g_g^{t-1})^2) + P_g*(g_r^t + g_r^t*g_g^{t-1} + g_r^t*(g_g^{t-1})^2)$$

$$= (P_s*s_r^t + P_g*g_r^t + P_s*s_g^t*g_r^t) + (P_s*s_r^t*s_s^{t-1} + P_g*g_r^t*g_g^{t-1} + P_s*g_r^t*s_g^t*g_g^{t-1}) + (P_s*s_r^t*(s_s^{t-1})^2 + P_g*g_r^t*(g_g^{t-1})^2 +$$

$$P_s*s_g^t*g_r^t*(g_g^{t-1})^2)  \qquad\qquad (4)"$$

*In order to make the text clearer, we added a supplement Table S1 to display the abbreviations of all the variables (See Table S1). In addition, we have explained the $P_s$ and $P_g$ terms here again. And we modified the terms that has been used to represent time information as subscript (t-1, t) to make them clearer.*

**Table S1 Acronym dictionary**

| | |
|---|---|
| *RWC* | *River water contribution* |
| *WUE* | *Leaf-level water use efficiency* |
| *WTD* | *Water table depth* |
| *T* | *Temperature* |
| *RH* | *Relative air humidity* |
| *$ET_0$* | *Reference evapotranspiration* |
| *VPD* | *Vapour pressure deficit* |
| *SWC* | *Soil water content* |
| *IRIS* | *Isotopic ratio infrared spectroscopy system* |
| *IRMS* | *Isotope Ratio Mass Spectrometry system* |
| *VSMOW* | *Vienna Standard Mean Ocean Water* |
| *$C_{Water}$* | *$^{222}$Rn concentration of the water samples* |
| *$C_{Air}$* | *Air $^{222}$Rn concentration of the water samples* |
| *$C_{System}$* | *Air $^{222}$Rn concentration values of the measurement system* |
| *$V_{System}$* | *The interior volume of the measuring set-up* |

| | |
|---|---|
| $V_{Sample}$ | The volume of water sample |
| $T_{res}$ | The average residence time of recharged groundwater from river water |
| $k$ | The $^{222}$Rn distribution coefficient of water/air |
| $\lambda$ | The decay coefficient |
| $C_e$ | The $^{222}$Rn concentration of background groundwater when the equilibrium between radon production and decay is reached |
| $C_r$ | The $^{222}$Rn concentration of river water |
| $C_g$ | The $^{222}$Rn concentration of riparian groundwater |
| $PWL$ | The potential water source line |
| $a_p$ | The slope of the PWL |
| $b_p$ | The intercept of the PWL |
| $PW_{excess}$ | The $\delta^2 H$ deviation of riparian tree xylem water from the PWL |
| $S_r$ | The total (during the entire period of river losing since 2007) RWCs to riparian deep soil water in the $80-170$ cm layer |
| $G_r$ | The total (during the entire period of river losing since 2007) RWCs to riparian groundwater |
| $P_s$ | The contributions of riparian deep soil water in the $80-170$ cm layer to riparian trees |
| $P_g$ | The contributions of riparian groundwater to riparian trees |
| $s_r^{t-1}$ | The proportional contributions of the old river water (before t-1) to riparian deep soil water in the $80-170$ cm layer |
| $g_r^{t-1}$ | The proportional contributions of the old river water (before t-1) to riparian groundwater |
| $s_s^{t-1}$ | The proportional contributions of in-situ soil water in the $80-170$ cm layer at t-1 for riparian deep soil water in the $80-170$ cm layer at t |
| $s_r^{t}$ | The proportional contributions of river water during t-1 to t for riparian deep soil water in the $80-170$ cm layer at t |
| $s_g^{t}$ | The proportional contributions of groundwater during |

$$g_g^{t-1} \qquad \text{The proportional contributions of in-situ groundwater at t-1 for riparian groundwater at t}$$

$$g_r^t \qquad \text{The proportional contributions of river water from t-1 to t for riparian groundwater at t}$$

*t-1 to t for riparian deep soil water in the 80−170 cm layer at t*

| | |
|---|---|
| *$g_g^{t-1}$* | *The proportional contributions of in-situ groundwater at t-1 for riparian groundwater at t* |
| *$g_r^t$* | *The proportional contributions of river water from t-1 to t for riparian groundwater at t* |
| *ANOVA* | *One-way analysis of variance* |
| *LMWL* | *Local meteoric water line* |

**RESULTS**

19. Comment:

Lines 234-235: I think the slope of the LMWL only gives an indication about how oxygen and deuterium co-evolve, it does not indicate if the value is high or low. You can have 2 measurements of lower isotopes values but still have the same slope as with 2 measurements of higher isotopes values.

**Response:** *The slope of the LMWL indeed mainly gives an indication of the evaporation degree of water samples. We have changed this sentence to "The slope of the local meteoric water line (LMWL) in 2021 (7.8) was significantly higher compared to 2019 (5.5) ($p < 0.05$), which suggested that the falling raindrops undergone stronger sub-cloud evaporation in 2019 (Zhao et al., 2019).".*

*Reference:*

*Zhao, L., Liu, X., Wang, N., Kong, Y., Song, Y., He, Z., Liu, Q. and Wang, L.: Contribution of recycled moisture to local precipitation in the inland Heihe River Basin. Agricultural and Forest Meteorology. 271, pp.316-335, 2019.*

20. Comment:

Line 276: How did you get a negative residence time? Would it make more sense to use "0"?

**Response:** *It will make more sense to use "0" instead of a negative residence time, due to the fact that the river water recharges groundwater frequently in this study. We have changed all the negative residence time values to "0" throughout the text and in Table 2. The confusing sentence "As shown*

*in Table 2, there was a significant increase of $^{222}$Rn activities in groundwater from D05 (494.5 ± 107.5 Bq/m³) to D45 (787.4 ± 153.2 Bq/m³) (p < 0.05). The $T_{res}$ of groundwater that recharged by river to the underlying aquifer and/or riverbank increased from D05 (−0.09 ± 0.09 days) to D45 (0.15 ± 0.13 days) (Table 2)." has been changed to "There was a significant increase of $^{222}$Rn activity in groundwater from D05 (610.1 ± 107.5 Bq/m³) to D45 (787.4 ± 153.2 Bq/m³) (p < 0.05) (Table 2). The $T_{res}$ of recharged groundwater from river water increased from D05 (0 days) to D45 (0.15 ± 0.13 days) (Table 2)."*

*Table 2: The $^{222}$Rn values in river water, background groundwater and riparian groundwater in three plots (D05, D20, and D45), and the average residence time of recharged groundwater from river water ($T_{res}$, day) in 2021. The background groundwater represents groundwater in aquifers more than 100 m away from the riverbank.*

| | *River water* | *Background groundwater* | *Riparian groundwater* | | |
| --- | --- | --- | --- | --- | --- |
| | | | *D05* | *D20* | *D45* |
| *$^{222}$Rn value (Bq/m³)* | *610.1 ± 212.3* | *7400.0 ± 35.4* | *610.1 ± 107.5* | *763.3 ± 118.3* | *787.4 ± 153.2* |
| *$T_{res}$ (days)* | *0* | *Null* | *0* | *0.13 ± 0.1* | *0.15 ± 0.13* |

*Notes: D05, D20, and D45 are the plots at distance of 5 m, 20 m, and 45 m away from the riverbank, respectively.*

21. Comment:

Lines 280-291: I would also remove the first sentence here and go straight to the results. I find the first paragraph difficult to follow because you present a mix of interannual and seasonal differences, think about what you want to present. You could have a first paragraph presenting the interannual differences, a second presenting the seasonal differences (differences between months for a SAME year). From my point of view, the Figure 9 only shows the differences between plots (you only show the stats for this), not between years (despite the same scales for the y axes) and months, so I would only refer to Figure 9 in the second paragraph.

**Response:** *We have removed the first sentence and went straight to the results. We have a first paragraph presenting the interannual differences, and a second presenting the seasonal differences (differences between months for a SAME year). We have only referred to Figure 9 in the third paragraph.*

*The first paragraph has been changed to "The proportional contributions of river water to riparian S. babylonica trees were significantly higher in 2021 (mean of 23.8% ± 7.8%) than in 2019*

*(mean of 16.8% ± 4.7%) (p < 0.05). Specifically, the most significantly monthly difference in the RWC to riparian S. babylonica trees between dry 2019 and wet 2021 was up to 19.8% (p < 0.001). The monthly maximum value of the RWC to S. babylonica trees was significantly higher in wet 2021 (35.2% ± 7.0%) compared with dry 2019 (24.2% ± 3.0%) (p < 0.05)."*

*The second paragraph has been added to present the seasonal differences in RWCs to riparian trees for a SAME year: "The riparian S. babylonica took up the most river water in July (35.2 ± 7.0%) and November (29.0 ± 5.0%) in 2021, whereas the highest RWCs to riparian trees occurred in May (22.2 ± 1.7%) and June (24.2 ± 1.6%) in 2019. The minimum river water uptake for S. babylonica in 2021 was 17.7 ± 2.7% (in September), while riparian trees took up the least water in August 2019 (13.2 ± 1.9%). No significant seasonal-trend of RWCs to riparian trees was observed in both years (p > 0.05)."*

22. Comment:

Lines 288-291: This section reads well but you say in the first sentence that there are significant differences in RWC between the 3 plots in 2021 while there is no difference (Fig 9), please correct.

**Response:** *We have corrected the first sentence as follows: "The water uptake of river water by riparian S. babylonica was significantly different between the three plots in 2019 (p < 0.05), while no difference was observed between the three plots in 2021 (p > 0.05) (Fig. 10)."*

23. Comment:

Lines 299-304: I would add in the text the R2 and p values of the linear models even if they are shown in Figure 10. Why did you fit the model to the whole dataset? And not one model for each year? It should be explained in the data analysis section, maybe I missed this point.

**Response:** *We added the $R^2$ and p values of the linear models in the text as follows: "There was a significant negative relationship between the RWC to riparian trees and the WTD ($R^2$ = 0.57; p = 0.000) (Fig. 11a). The leaf $\delta^{13}C$ of riparian S. babylonica was found to be negatively correlated with the RWCs to S. babylonica ($R^2$ = 0.61; p = 0.000) but positively related to the WTD ($R^2$ = 0.37; p = 0.000) in linear functions (Fig. 11b and c)."*

*We also added the reasons for fitting the model to the whole dataset in the "2.5 data analysis" section. "In order to get the general relationships (not only available for 2019 dry year but also for*

*2021 wet year) between the WTD, leaf $\delta^{13}C$ values and RWCs to riparian trees, the linear regression model using for quantifying their relationships was fitted to the whole dataset in both two years."*

**DISCUSSION**

24. Comment:

I think that this section is not well-enough developed and that the sections should be revised in order to reflect the objectives presented in the introduction. There is also no discussion about the MixSIAR model and the iteration method presented here and on which all the results are based. I would add a section to discuss about their strengths/weaknesses and implications for the discussed results. Then, I would discuss the RWC to riparian trees and the effect of the distance from the stream. Finally, I would discuss about the link between RWC/WUE/WTD and its implications (also include management). I think the discussion about the potential processes and implications should be developed, also how do your results compare with previous work and why?

**Response:** *This is a good comment. We have reorganized the entire discussion section. Firstly, we discussed the strengths/weaknesses and implications of the MixSIAR model and the iteration method. Secondly, we discussed the RWC to riparian trees and the effects of the distance from the stream and dry/wet year on RWCs to riparian trees. In this section, we discussed and developed the potential processes. Finally, we discussed the link between RWC/WUE/WTD and its implications on management of riparian forest and river runoff. We have compared the results with previous work and provided corresponding explanations throughout the revised discussion section. The entirely revised discussion was displayed in the response to comment #1.*

25. Comment:

Lines 336-342: You just look to mostly report previous findings here. First, what YOUR results suggest? Then, HOW does it relates/compare to previous work?

**Response:** *We have revised these sentences to emphasize the implications of our results and compare our results with previous work. The revised part can be found in the first paragraph of the 4.2 discussion section in the response to comment #1.*

26. Comment:

Lines 344-345: This point is super interesting; can you try to make the story about this clearer?

**Response:** *We have clarified this point in detail. The revised story has been specified in the response to the comment #1 (The first paragraph of the 4.2 discussion section).*

27. Comment:

Lines 381-382: I don't think you can really compare your "optimal" WTD with values from other studies because it is not the same site. I would rather discuss the potential reasons of these differences. Clarify the "knee point", I see what you mean but I would reword, a "break point" instead?

**Response:** *We have compared our "optimal" WTD with values from other studies and extensively discussed the potential reasons of these differences. The reorganized discussion part has been specified in the response to the comment #1 (The second paragraph of the 4.3 discussion section)*

*In addition, we changed the "knee point" to "break point".*

28. Comment:

Line 390: You talk about accurate separation and quantification of RWC to riparian trees but we don't know the limitations and uncertainty of the model and iteration method.

**Response:** *We have added the discussion about the advantages, limitations and uncertainties of the model and iteration method in the section 4.1 ("The strengths/weaknesses and implications of the MixSIAR model and the iteration method"). The reorganized discussion section has been specified in the response to the comment #1.*

CONCLUSION

29. Comment:

1. Lines 401-408: I think this is too much results, the conclusion should not be like an abstract. I would focus more on the implications of your findings for riparian zones management and future research.

**Response:** *We have modified the conclusion part and focused more on the implications of our*

*findings for riparian zone management and future research. The revised conclusion part is as follows:*

*"In this study, we presented a new iteration method together with stable water isotopes ($\delta^2 H$ and $\delta^{18}O$) and the MixSIAR model to separate and quantify the indirect RWC to riparian S. babylonica in dry 2019 and wet 2021 along a losing river in Beijing, China. It was found that the infiltrating river water quickly exchanged with riparian mobile water but not mixing with waters held tightly in the fine pores. Riparian trees nearby a losing river generally extended roots into fine pores to access to immobile water sources. The isotopic discrepancies between fast-moving water flow and immobile water for plant water uptake led to a small RWC (20.3%) to riparian trees. The water deficit in dry year could induce stomatal closure and larger reduction in transpiration of riparian trees, leading to an evident increase of WUE compared with that in wet year. The leaf WUE showed a negatively correlation with the RWCs to riparian trees but was positively related to the WTD in linear functions (p < 0.001). These suggested that rising water table would trigger riparian trees to increase river water acquisitions and show a consumptive water use strategy, which would not be recommended for the water resources management of a losing river. The maximum WTD of 4 m seemed to be optimal for protecting both the river runoff and riparian revegetation, maintaining highest water use efficiency, and minimizing the plant transpiration. This study provides valuable insights into riparian afforestation related to water use and healthy riparian ecosystem enhancement."*

**Technical corrections:**

**INTRODUCTION**

30. Comment:

Line 34: English is not my mother tongue but should it be "replenishment" instead of "replenishing"?

**Response:** *We have changed " replenishing" to "replenishment".*

31. Comment:

Lines 38-40: This is a very interesting question but some parts of the sentence need to be edited to have a clearer sentence. By "where" I guess you mean the sources? At the first read I thought you

were talking about where along the river, I would use "source" to be clearer. I would change "responses to the variations in the water table" to "response to water table variations". Also, be more precise about what you are talking about: is it the river water or groundwater level? When you say "revegetated riparian species" it means that the species are revegetated, which is false, I would change to "tree water requirement of revegetated riparian zones/areas" for exemple (it is the riparian zone/area that have been revegetated). Finally, I am not sure that the word "balance" is the best one to use here, I would improve the wording of the sentence.

**Response:** *We have modified the "where and how much water riparian tree take up" to "what water sources and how much river water is used by riparian trees". In addition, we have changed "responses to variations in the water table" to "responses of plant water use characteristics to groundwater level variations". The "revegetated riparian species" has been changed to "revegetated riparian zones". We have deleted the word "balance" and improved the wording of this sentence as follows: "Therefore, understanding what water sources and how much river water are used by riparian trees as well as the responses of plant water use characteristics to groundwater level variations can help to control the river runoff and tree water requirement of revegetated riparian zones."*

32. Comment:

Line 47: Can you be more specific about the "different waters"? Different "water sources"?

**Response:** *We have specified the "different waters" as "different water sources and plant stem water"*

33. Comment:

Line 51: Do you mean "change in river water level"?

**Response:** *Yes. We have changed "changes in river water" to "the changes in river water level"*

34. Comment:

Lines 62-66: I would improve the wording of the sentences. From my point of view, it does not read that well while it is important to state clearly the knowledge gap/issue here. Also, "estimations" of what?

**Response:** *We have rewritten these sentences and clarified the knowledge gap as follows: "There is growing evidence indicating that riparian trees at a certain distance away from the riverbank rarely took up river water directly, because their lateral roots could not reach the river (Mensforth et al., 1994; Thorburn and Walker, 1994). Nevertheless, riparian trees could indirectly utilize the river water seeped into riparian deep zone (including deep soil water and groundwater) when their roots tapped into the groundwater level (Mensforth et al., 1994; Wang et al., 2019b). Treating river water as a direct water source might lead to inaccurate estimations of the RWC to transpiration flux. However, it remains unclear that how to separate and quantify the contribution of the indirect river water source that recharges riparian deep water to transpiration flux of riparian trees nearby losing rivers."*

35. Comment:

Lines 67-81: Maybe I'm too picky but be more specific when you use WUE: is it WUE of plant (lines 69, 70)? Trees (lines 73, 79)? Similarly for RWC, I would specify "RWC to riparian trees" (line 67).

**Response:** *We have added the tree or leaf in front of "WUE" and specified the "RWC to riparian trees" throughout the manuscript.*

36. Comment:

Line 75: Water table depth: of groundwater?

**Response:** *We have changed the "water table depth" to "depth of water table".*

37. Comment:

Lines 82-88: Great, very clear objectives here.

**Response:** *Thanks for your positive comments.*

38. Comment:

Line 86: I would specify "tree WUE".

**Response:** *We have specified WUE as "tree WUE".*

**M&M**

39. Comment:

Line 94: I think "dried up from X to X" would be more correct. Or "during X up to X"?

**Response:** *We have changed "dried up during 1999 to 2007" to "dried up from 1999 to 2007".*

40. Comment:

Lines 95-96: The end of the sentence is a bit unclear due to the wording. I would change to "more than 33 km$^2$ of riparian zone has been revegetated until 2020" or "from 2007 to 2020". Have the trees been planted?

**Response:** *Yes, trees have been planted. We have changed this sentence to "more than 33 km$^2$ of riparian zone has been revegetated until 2020".*

41. Comment:

Line 100: I would change to "from April to November 2019 and 2021".

**Response:** *We have changed it to "from April to November in 2019 and 2021".*

42. Comment:

Line 101: "were collected", not "was".

**Response:** *We have changed "was collected" to "were collected".*

43. Comment:

Line 104: I think it is "water level gauge".

**Response:** *We have changed "water gauge" to "water level gauge".*

44. Comment:

Line 105: "from April to November".

**Response:** *We have changed "during April to November" to "from April to November".*

45. Comment:

Line 106: "with a total precipitation of".

**Response:** *We have changed "with total precipitation of" to "with a total precipitation of".*

46. Comment:

Line 107: I would correct and say "which was 1.8 times higher than for the drier year 2019 (445.6 mm)".

**Response:** *We have changed "which was 1.8 times of that in dry year of 2019 (445.6 mm)" to "which was 1.8 times higher than for the drier year 2019 (445.6 mm)".*

47. Comment:

Line 108: I would also correct here: "fluctuated between X and X m" and "mean WTD across the three plots".

**Response:** *We have corrected "fluctuated at 27.9−28.9 m in 2019 and 27.3−29.7 m in 2021" as "fluctuated between 27.9 and 28.9 m in 2019 and between 27.3 and 29.7 m in 2021". The "mean WTD in three plots" has been changed to "mean WTD across the three plots".*

48. Comment:

Line 109: Use "higher" not "larger" to compare values. Also, change to "higher than in X".

**Response:** *We have used "higher" to compare values. The "larger than that in 2021" has been changed to "higher than in 2021".*

49. Comment:

Lines 109-110: Is it not the opposite? The WTD is lower (shallower GW) closer to the stream (see Figures 1 and 3).

**Response:** *We have changed this sentence to "The WTD increased with increasing distances from the riverbank in both 2019 and 2021 (Fig. 3)."*

50. Comment:

Lines 112-113: Please write the months in full.

**Response:** *We have spelled out the months. The corrected sentence is as follows: "Twelve sampling campaigns on May 5, June 14, July 26, August 15, September 26, November 5 in 2019 and April 24, May 25, June 26, July 15, September 1, November 5 in 2021 were conducted to collect groundwater, river water, soil, stem, and leaf samples."*

51. Comment:

Line 132: I would say "extract water from xylem and soil samples" instead.

**Response:** *We have changed "extract water in stem and soil samples" to "extract water from xylem and soil samples".*

52. Comment:

Line 133: "above 99%".

**Response:** *We have changed "more than 99%" to "above 99%".*

53. Comment:

Line 137: I would use "xylem" rather than "stem" in this section.

**Response:** *We have used "xylem" rather than "stem" in this section.*

54. Comment:

Line 138: Correct to "for both the IRIS and IRMS systems".

**Response:** *We have changed "between the IRIS and IRMS systems" to "for both the IRIS and IRMS systems".*

55. Comment:

Lines 141-151: It is a clear section, but I am not familiar with $^{222}$Rn and don't really understand the sentence on lines 145-146 "to ensure… less than 80 Bq/m$^3$". Can you clarify?

**Response:** *We have added more information about "to ensure… less than 80 Bq/m$^3$". "The air $^{222}$Rn concentration values in the $^{222}$Rn monitor ($C_{Air}$, Bq/m$^3$) were recorded at 10-minute intervals. The air inside the measurement set-up had contained a certain $^{222}$Rn concentration right before injecting the water sample ($C_{System}$, Bq/m$^3$). It is generally assumed that the already existing $C_{System}$ can be*

*ignored accordingly when $C_{System}$ is around or lower than 80 Bq/m³. In this study, more than four intervals were conducted to ensure that the $C_{System}$ was less than 80 Bq/m³. The measurement range of $C_{Air}$ was 2–2,000,000 Bq/m³ with a measurement precision of 3%. The above measured $C_{Air}$ value was not yet the ²²²Rn concentration in the measured water sample ($C_{water}$), because the ²²²Rn driven out had been diluted by the air within the ²²²Rn monitor and a small part of the ²²²Rn remained diluted in the watery phase."*

56. Comment:

Lines 153-156: I feel that these sentences do not belong here but in the introduction, for me the section starts on line 156 at "in this study…".

**Response:** *We have deleted these sentences in the M&M part and moved them to the introduction part.*

57. Comment:

Line 157: I would correct with "isotopes were integrated/used within/in the MixSIAR model and an iteration method was proposed to identify.". What do you mean by the "original"?

**Response:** *We have corrected with "In this study, water stable isotopes ($\delta^2H$ and $\delta^{18}O$) were integrated within the MixSIAR model and an iteration method was proposed to identify the contributions of the indirect river water that recharged riparian deep water to riparian S. babylonica trees (Figs. 4 and 5)". The "original" means that the total river water contribution to riparian deep water during the entire period of river losing flow to riparian deep zone since 2007. We have changed "original" to "total" and added more explanation about the total river water contributions in section 2.4.3. Here is the revised part: "It was worth noting that riparian deep soil water (80−170 cm layer) and groundwater can be recharged by river water continuously, when the groundwater levels lied below the riverbeds (i.e., losing rivers). The total RWC to riparian deep water should be explicitly identified during the entire period of river losing flow to riparian deep zone since 2007, although the proportional contribution of old (before t-1) river water for riparian deep water might be small."*

58. Comment:

Lines 158-159: I am not sure "merge" is the correct word, please correct the wording and grammar of the sentence.

**Response:** *We have changed the "merged" to "recharged". This sentence has been corrected as follows: "In this study, water stable isotopes ($\delta^2H$ and $\delta^{18}O$) were integrated within the MixSIAR model and an iteration method was proposed to identify the contributions of the indirect river water that recharged riparian deep water to riparian S. babylonica trees (Figs. 4 and 5)".*

59. Comment:

Lines 159-163: I would check the wording of these sentences, I found it difficult to understand (maybe follow the section titles you used for 2.4.1., 2.4.2. and 2.4.3.? – they are clear). For example, what do you mean by "root water uptake patterns"? The sources? "Without considering river water as a direct water source"? Also, using "figured out" connotes a lack of accuracy, so I would use "determined" instead, for example.

**Response:** *We have changed the "root water uptake patterns" to "direct water source (including soil water at three different layers and groundwater) contributions to riparian trees". We have deleted "without river water as a direct water source" and changed "figured out" to "determined". The sentences have been corrected as follows: "Firstly, the direct water source (including soil water at three different layers and groundwater) contributions to riparian trees were determined via $\delta^2H$ and $\delta^{18}O$ in different waters and the MixSIAR model. Secondly, the proportional contributions of river water to riparian deep water (i.e., riparian groundwater and deep soil water in the $80-170$ cm layers) were determined by the MixSIAR model and water isotopes. Finally, the proposed iteration method was used to quantify the proportions of the indirect RWC to riparian trees (Figs. 4 and 5)."*

60. Comment:

Lines 165-168: I found these sentences difficult to understand, I would check the wording. What do you mean by "which was mixed proportionally", "relatively stable" in terms of seasonal variations in SWC, water isotopes and WTD"? I am also not sure these sentences are needed here.

**Response:** *We have carefully checked the wording of theses sentences and deleted "which was mixed proportionally with precipitation, old soil water, or even river water and groundwater" as*

well as *"relatively stable"*. *However, these sentences were needed to define the direct water sources for riparian trees in the MixSIAR model, which was critical to identify the contributions of indirect river water source to riparian trees.*

*"Soil water at different depths was taken up by riparian S. babylonica directly. We measured soil water isotopes at 11 depths in the three plots. In order to reduce errors in the analytical procedure, four soil layers (0−30 cm, 30−80 cm, 80−170 cm, and 170−300 cm) were divided to identify the main root water uptake depth of riparian trees according to seasonal variations in the SWC, water isotopes and WTD. The average soil water isotopes values at depths of 0−5 cm, 5−10 cm, 10−20 cm, and 20−30 cm were calculated for the 0−30 cm soil layer, because the water isotopes went through strong evaporation and the SWC varied significantly seasonally. The soil water isotope values at depths of 40−60 cm and 60−80 cm were averaged for the 30-80 cm soil layer, and those values at 90−110 cm and 150−170 cm depths were averaged for the 80−170 cm soil layer as the water isotopes and SWC were relatively stable. The average isotopes values of soil water at deep depths (190−210 cm, 250−270 cm, and 280−300 cm) were calculated for the 170−300 cm soil layer, which varied with the fluctuations of water tables. Groundwater could also be regarded as a direct water source for phreatophyte riparian trees (Dawson and Ehleringer, 1991; Busch et al., 1992). As the isotopic composition of soil water in the 170−300 cm layer was similar to that of groundwater, they were considered to be one water source (groundwater)."*

61. Comment:

Line 170: I don't understand why you refer to Figures 2, 3 and S1 here, I don't see the link with why you separated the soil in 4 layers. Correct "in the 170-230 cm layer".

**Response:** *We have deleted the Figures 2, 3, and S1 here and changed "in 170-300 cm layer" to "in the 170-300 cm layer".*

62. Comment:

Line 172: I would change "determined" to "used as direct water sources".

**Response:** *We have revised it as suggested.*

63. Comment:

Lines 173-174: I would change "stem" to "xylem" since you measured isotopes of xylem water.

**Response:** *We have changed "stem water" to "xylem water" when we referred to "measured isotopes of xylem water" throughout the manuscript.*

64. Comment:

Lines 177-179: I don't think these sentences are needed here since you develop this point in section 2.4.2.

**Response:** *We have deleted this sentence in this paragraph.*

65. Comment:

Line 181: I would specify "deep soil water (80-170 cm) and groundwater", also check grammar of the sentence.

**Response:** *We have changed this sentence to "riparian deep soil water (80−170 cm) and groundwater can be recharged by river water continuously, when the groundwater levels lied below the riverbeds (i.e., losing rivers).". We also corrected the wording and the grammar mistakes throughout the manuscript.*

66. Comment:

Lines 181-184: There are some wording and grammatical mistakes I think. Change "could be" by "can be"? "were applied" by "were used"? "in the 80-170 cm layer".

**Response:** *We have corrected "could be" as "can be" and changed "were applied" to "were used". We have modified all the "in … cm layer" to "in the … cm layer" throughout the manuscript.*

67. Comment:

Lines 183-190: I like this part, it's clear, and the Figure S2 is great! I just wonder if it would be better to have the figure in the main text rather than in the Supplement.

**Response:** *We have moved the Figure S2 to the main text (See Fig. 4).*

[Figure]

*Figure 4: Schematic diagram for potential water sources of (a) riparian deep soil water in 80−170 cm layer and (b) groundwater.*

68. Comment:

Lines 191-192: I am not sure about the wording here: "recharged from the river to the underlying aquifer and/or riverbank"… or maybe I misunderstood.

**Response:** *We have changed "the residence time of groundwater recharged from the river to the underlying aquifer and/or riverbank" to "the residence time of recharged groundwater from river water".*

69. Comment:

Line 199: I would not use "figured out" in a manuscript, I would use "determined" instead. Please correct the "figured out" throughout the manuscript.

**Response:** *We have changed "figured out" to "determined" throughout the manuscript.*

70. Comment:

Line 200: I am not sure about the use of "merge" here.

**Response:** *We have changed "merged" to "recharged".*

71. Comment:

Line 202: What do you mean by "be consistent"? Can you change "proportions" to "contribution"?

**Response:** *We have changed "be consistent" to "be same" in this sentence. And we have modified "proportions" to "contributions".*

72. Comment:

Line 222: I am not sure "recharge" is the best word, maybe say "we estimated the proportions of old and current river water in the riparian deep water", but should it be "in the riparian trees" instead (the aim of the section 2.4.3.)?

**Response:** *Yes, we have changed this sentence to "Using this proposed iteration method, we estimated the total proportions of old and current river water in the riparian trees.".*

73. Comment:

Lines 224-229: This section is a bit difficult to read, the first sentence it too long, I would try to separate it. Also, check the grammar and wording. The "regression analysis method" is unclear.

**Response:** *We have carefully checked the grammar and wording of this section. The revised section is as follows: "One-way analysis of variance (ANOVA) incorporating with Kolmogorov-Smirnov, Levene's and post-hoc Tukey's tests ($p < 0.05$) were used to investigate the statistic differences of different variables. These variables included the WTDs, SWC, $\delta^2H$ and $\delta^{18}O$ in different water sources and xylems, $^{222}Rn$ concentration of river water and groundwater, source water contribution to riparian deep water as well as trees, and leaf $\delta^{13}C$ values in the three plots in 2019 and 2021. In order to get the general relationships (not only available for 2019 dry year but also for 2021 wet year) between the WTD, leaf $\delta^{13}C$ values and RWCs to riparian trees, the linear regression model using for quantifying their relationships was fitted to the whole dataset in both two years. The statistical analysis was performed in the Excel (v2016) as well as SPSS (24.0, Inc., Chicago, IL, USA)."*

**RESULTS**

74. Comment:

This section is well-organized and the results are presented concisely and quite clearly. There are some grammar and wording mistakes, especially when you present the isotopes results. You can't say "d$^2$H in precipitation was more depleted", but say "precipitation was more depleted in d2H" or "d2H in precipitation was higher/lower than…". Check throughout the section 3.1 (lines 232, 239, 240, 242-243, 244-245…). Also, you use a lot of "than these/that" (line 233) or "with that in" (line 236) to compare results between years, I would correct and use "than in" or "compared to" instead. Check also the use of "the".

Response: *Thanks for your helpful comments and suggestions. We have corrected the grammar and wording mistakes throughout the manuscript. The presentation of the isotopes results has been corrected as suggested throughout the manuscript. The incorrect words (including "than these/that", "with that in", and et al) were changed to "than in" or "compared to" in order to compare results between years. We also corrected the use of "the".*

75. Comment:

Line 232: I would not use "it was evident", just present the results clearly and briefly.

Response: *We have deleted "it was evident".*

76. Comment:

Line 234: Use "higher" rather than "larger" to compare values.

Response: *We have changed "larger" to "higher" throughout the manuscript.*

77. Comment:

Line 236: I would rather say "SWC of each soil layer" than say "of all four layer", at the first read I thought you combined all the soil layers together but you analyzed the difference between the plots for each soil layer separately from what I understood.

Response: *We have changed "SWC of all four soil layers" to "SWC of each soil layer".*

78. Comment:

Lines 246-253: This part reads better, it's clear and concise.

**Response:** *Thanks for your positive comments.*

79. Comment:

Line 250: I would correct to "decreased with increasing distance from the riverbank".

**Response:** *We have corrected to "decreased with increasing distance from the riverbank".*

80. Comment:

Line 253: What do you mean by "evidently"? Is it significant? and "plummeted"?

**Response:** *We have changed "evidently" to "significantly ($p< 0.05$)" and added the significance after plummeted "plummeted significantly ($p< 0.05$)".*

81. Comment:

Lines 255-278: This section reads well and is well organized, check the mistakes I referred to previously. I would remove the first sentence and would go straight to the results.

**Response:** *We have corrected the wording, grammar, and other mistakes throughout the whole section. We have also removed the first sentence of this section and went straight to the results.*

82. Comment:

Line 257: I am not sure about the use of the word "in-situ" to refer to the water that is already in the deep soil or groundwater compartment… But I don't really what word you could use instead so I'm not very helpful on this point.

**Response:** *Thanks for your comments. We have explained the meaning of "in-situ" word at the first use in the text. Here is the revised part: "The potential water sources of riparian deep soil water in the 80−170 cm layer at t included the in-situ (i.e., the water that is already in the deep soil or groundwater compartment) soil water in this layer at t-1, soil water in the 0−80 cm layer at t-1, river water between t-1 and t, precipitation between t-1 and t, and groundwater between t-1 and t (Fig. 4a)."*

83. Comment:

Line 261: "15.7%".

**Response:** *We have changed "15.7" to "15.7%".*

84. Comment:

Line 262: "deep soil water", "the lowest", "and in June".

**Response:** *We have corrected them as suggested.*

85. Comment:

Line 265: I would correct: "significant interannual and seasonal differences in the water sources…".

**Response:** *Thanks a lot for your suggestion. But we have deleted this sentence and went straight to the results according to the reviewer's comment # 81.*

86. Comment:

Lines 268-269: Please correct the wording of this sentence.

**Response:** *We have changed this sentence to "The average contribution of river water to riparian groundwater was 28.1 ± 12.1% during the observation period.".*

87. Comment:

Lines 271, 272: You should refer to Figure 3 here to help the reader understand since you present some of the WTD results.

**Response:** *We have referred to Figure 3 in this sentence.*

88. Comment:

Lines 275-276: As I mentioned before, I still don't understand this sentence, maybe a wording mistake or me… Check the grammar as well.

**Response:** *We have changed the negative residence time "−0.09 ± 0.09 days" to "0" in this sentence. We also corrected the grammar mistake and revised it as follows: "The $T_{res}$ of recharged groundwater from river water increased from D05 (0 days) to D45 (0.15 ± 0.13 days) (Table 2)." (see the response to the comment #20)*

89. Comment:

Line 282: Please put the unit "%" after each result, "mean of X% ± X%", check and correct throughout the manuscript.

**Response:** *We have put the unit "%" after each result and corrected throughout the manuscript.*

90. Comment:

Lines 282, 284, 285, 286: Correct the grammar: "higher" not "more" or "larger", "lowest" not "least", "highest" not "most" to compare values. You can say "X was higher/lower than X" or "X was the highest/lowest in…".

**Response:** *We corrected this grammar mistake throughout the manuscript.*

91. Comment:

Lines 288-291: Use "between" not "among" to make comparisons. The first part of second sentence is perfect but the end is unclear "whereas….. in 2021" ("corresponding value"?, "along the distances"?), please correct the wording and grammar, or you could just say that there was no significant differences in 2021.

**Response:** *We have changed "among" to "between" to make comparisons throughout the manuscript. The end part of second sentence has been changed to "whereas there was no significant differences in 2021 (p > 0.05) (Fig. 10)."*

92. Comment:

Line 292: I would slightly change the title to: "relationships between leaf $d^{13}C$, RWC to riparians trees and WTD".

**Response:** *We have corrected this presentation as suggested throughout the manuscript.*

93. Comment:

Lines 293-298: Check the wording and grammar mistakes ("the" missing, "higher" not "larger", "significant" not "significantly"…) as commented above, write the months in full, use "significant" instead of "remarkably" (line 294).

**Response:** *We have corrected this paragraph as suggested. The revised version is as follows:*

*"The leaf $\delta^{13}C$ of riparian S. babylonica trees was significantly higher in 2019 ($-27.7\% \pm 1.0$ ‰) than in 2021 ($-29.7\% \pm 0.7$ ‰) ($p < 0.05$) (Table 1). There was a significant increase of the leaf $\delta^{13}C$ from D05 ($-28.8$‰) to D45 ($-27.0$‰) in 2019 ($p < 0.05$), while no significant difference in the leaf $\delta^{13}C$ was observed between the different distances in 2021 ($p > 0.05$). The lowest leaf $\delta^{13}C$ value of riparian trees occurred on August 15 in 2019 and July 14 in 2021, before when the intense rainfall occurred in both years."*

94. Comment:

Lines 299-304: This section reads better. Correct "RWC to riparian trees". I would move the last sentence to the discussion section.

**Response:** *We have corrected "RWC to riparian trees". And we have moved the last sentence to the discussion section.*

DISCUSSION

95. Comment:

The flow of your thoughts is difficult to follow, I would try to be clearer in my explanations. There are wording and grammar mistakes that need to be corrected, also check the tense you use.

**Response:** *We have rewritten the whole discussion part and corrected the wording and grammar mistakes as well as tense mistakes (see the revised discussion section in response to the comment #1).*

96. Comment:

Line 307-326: The section is well-organized but you don't need to repeat the results in this section (lines 307-310, 316-317, 324-325).

**Response:** *We have deleted the results and discussed the potential processes and implications in this section. We also compare our results with previous work and provided more explanations for the reasons.*

97. Comment:

Line 311: I find it difficult to understand what do you mean by "contradictions", please develop your thoughts and the processes involved.

**Response:** *We have deleted the vague "contradictions" and rewritten this part. We also developed and made a complete story including my thoughts, potential processes, and implications. The revised part has been specified in the response to comment #1 (The first paragraph of the 4.2 discussion part).*

98. Comment:

Line 311: I would be more precise and not use "interactions", we don't know if it is river-GW flow or GW-river flow, in your study you only looked at river-GW flow, say it.

**Response:** *In our study, there was only the process of river water recharging the groundwater system. We have changed the confusing word "interactions" to "river recharging the groundwater system" throughout the manuscript.*

99. Comment:

Line 321: Same comment for "exchange", it is not clear enough.

**Response:** *We have also changed the confusing word "exchange" to "river recharging the groundwater system" throughout the manuscript.*

100. Comment:

Line 323: I don't think that "weakened" is the right word to use here.

**Response:** *We have deleted "weakened" and rewritten these sentences. The revised part is as follows: "Although there was no significant difference in the deep water (below the 80 cm layer) contributions to riparian trees between three plots, we observed the substantial effect of the declining water table with increasing distance from the riverbank on the reduced indirect RWCs to riparian trees in dry 2019 (Fig. 10). Therefore, the temporal and spatial variabilities of the RWC to riparian S. babylonica trees generally attributed to the various RWCs to riparian deep water rather than the water use patterns of riparian trees."*

101. Comment:

Line 326: "distance" from what?

**Response:** *We have specified "distance from the riverbank" throughout the manuscript.*

102. Comment:

Lines 329-335: The section here is also well-organized but I still find it difficult to follow your story.

**Response:** *We have rewritten this part to make a clearer story including my thoughts, potential explanations for small RWCs to riparian trees, comparison with previous studies and corresponding explanations. The revised part has been specified in the response to comment #1 (The first and second paragraphs of the 4.2 discussion section).*

103. Comment:

Line 329: You can say "smaller than.." or say "small", please correct.

**Response:** *We have changed "smaller" to "small".*

104. Comment:

Lines 331-335: I like this part, just check the grammar and wording ("or that stored").

**Response:** *Thanks for your positive comment. We have combined this part with the potential explanations for small RWCs to riparian trees in the 4.2 section part to make the story clearer. The revised part has been specified in the response to comment #1 (The first paragraph of the 4.2 discussion section).*

105. Comment:

Line 346: Here and throughout the manuscript, don't forget: "RWC to riparian trees".

**Response:** *We have corrected "RWC to riparian trees" throughout the manuscript.*

106. Comment:

Line 347-349: These 2 sentences should be switched, I would first briefly remind the result and then say what it suggests, you did the opposite here.

**Response:** *We have switched these two sentences and rewritten this part to make it clearer.*

107. Comment:

Line 349: I would reword "along the gradient of distance".

**Response:** *We have changed "along the gradient of distance" to "with increasing distance from the riverbank".*

108. Comment:

Line 351: The technical word "dimorphic" should be explained at first use to help the nonexpert reader to understand what you mean.

**Response:** *We have explained the meaning of "dimorphic" root systems at first use. The "dimorphic" root systems can help plant species to shift their main water sources between shallow and deep layers. Here is the revised part: "Our result contrasts with the previous study by Qian et al. (2017) who reported a significant increase of the RWC to G. biloba trees in response to the water table decline. It was ascribed to that riparian G. biloba had a dimorphic root system and shifted their main water sources from shallow soil layer to deeper soil layer. Nevertheless, the potential root growth rate of riparian phreatophyte S. babylonica trees can reach 1-13 mm/day, which allows the riparian S. babylonica trees to remain in contact with a rising/declining water table and keep constant water uptake proportions from deep strata below the 80 cm depth in this study (Naumburg et al., 2005)."*

109. Comment:

Lines 351-355: Interesting difference between tree species, try to reword to clarify this point. The sentence on lines 354-355 is repeating the one on lines 351-353.

**Response:** *We have rewritten these sentences to clarify the difference between tree species. The revised part has been shown in the comment #108.*

110. Comment:

Line 357: I don't think "balance and coordination" are the right words here.

**Response:** *We have deleted the word "balance and coordination" and improved the wording of this sentence as follows: "Therefore, understanding what water sources and how much river water are used by riparian trees as well as the responses of plant water use characteristics to groundwater*

*level variations can help to control the river runoff and tree water requirement of revegetated riparian zones."*

111. Comment:

Lines 361-369: I like the ideas here but I would improve the wording and check the grammar to improve the flow.

**Response:** *We have rewritten and improved the flow of this part to make a complete story including our thoughts, potential processes, and implications. The revised part has been specified in the response to comment #1 (The first paragraph of the 4.2 discussion section part).*

112. Comment:

Line 362: "profligate" is not the right word here; it can't be used to describe water-use strategy.

**Response:** *We have modified "profligate water use strategy" to "consumptive water use strategy"*

113. Comment:

Line 366: Say "river water" not "river flow".

**Response:** *We have corrected it throughout the manuscript.*

114. Comment:

Lines 371-374: This is results, not discussion.

**Response:** *We have deleted these results and rewritten this part.*

115. Comment:

Lines 375-380: I like the ideas here; grammar and wording need to be improved.

**Response:** *We have rewritten this part and improved the grammar and wording. The revised part has been specified in the response to comment #1 (The second and third paragraphs of the 4.3 discussion section).*

116. Comment:

Line 375: Correct the grammar: "previous studies that showed an…".

**Response:** *We have corrected as suggested.*

117. Comment:

Line 376: No need to add the equation.

**Response:** *We have deleted the equation.*

118. Comment:

Line 377: I think "coordinate" is not the right word here, maybe "optimize"?

**Response:** *We have changed "coordinate" to "optimize".*

119. Comment:

Line 379: I would reword "balancing the relationship" and "flow reservation".

**Response:** *We have deleted these words and rewritten these sentences. The revised part is as follows: "Therefore, the relationships between the RWC to riparian trees, leaf-level physiological characteristics (e.g., leaf WUE) and hydro-meteorological conditions are critical and helpful for the better protection of the riparian forest while maintaining sustainable river runoff."*

120. Comment:

Line 383: I would say "groundwater table" rather than "water table", it is clearer, please check throughout the manuscript

**Response:** *Thanks a lot for your comments. But we prefer to use "water table" throughout the manuscript. The "water table" is more official to define "the upper surface of the saturated zone". It has been widely used in many official websites and academic journals (e.g., https://www.usgs.gov/faqs/what-groundwater and https://education.nationalgeographic.org/resource/water-table).*

121. Comment:

Lines 386-397: I find the flow of this section hard to follow, thoroughly check the wording and grammar, don't hesitate to ask an English native speaker.

**Response:** *We have rewritten this part thoroughly and invited an English native speaker to help*

*check the wording and grammar.*

CONCLUSION

122. Comment:

Lines 399-412: Check and correct the wording and grammar.

**Response:** *We have corrected the wording and grammar in this part.*

123. Comment:

Lines 399-401: I like this part, reminding the objective of the study.

**Response:** *Thank you for the positive comments.*

FIGURES

124. Comment:

Figure 1: Great figure!! It is very clear, easy to read and show all the info needed. As I see dams along the river, I wonder if water was released during the study periods and how it could have affected river flow and the results?

**Response:** *Thanks for your positive comments. Due to continuous drought and groundwater overexploitation, the Chaobai River dried up from 1999 to 2007. The ecological water (including reclaimed water, reservoir water, and diverted water by the South-to-North Water Transfer Project) has been supplied to restore this dry river via a systematic water release by dams since 2007. A total of 51.1 million and 380 million cubic meters of ecological water sources were released to the Chaobai River in 2019 and 2021, respectively. This significantly different amount of ecological water release between two years led to a remarkable discrepancy in water table between dry 2019 and wet 2021 (Fig. 3). Moreover, the dams along the Chaobai River are used to regulate the river water level especially in flood season, which has a great effect on the river runoff and thus the riparian groundwater level. However, little was known about the effects of the river water level/groundwater level on the water use characteristics of riparian trees. Therefore, our study aims*

*at understanding the RWC to riparian trees and their responses to groundwater level variations.*
*This could help to control the river runoff and tree water requirement of revegetated riparian zones.*

125. Comment:

Figure 2: Check the reference to Figure 2 in the text, lines 106-107 should only refer to Figure 2a. I would change the caption to "Monthly average precipitation amount from 1961 to 2021 and monthly total precipitation amount for the observation years 2019 and 2021 (a), Daily total precipitation amount and precipitation isotopes during 2019 (b) and 2021 (c).

**Response:** *We have changed the reference to "Figure 2b and c" in the text to "Figure 2a". And we modified the caption to "Monthly average precipitation amount from 1961 to 2021 and monthly total precipitation amount for the observation years 2019 and 2021 (a), Daily total precipitation amount and precipitation isotopes during 2019 (b) and 2021 (c)".*

126. Comment:

Figure 3: The figure is clear but I would show groundwater level (on same scale as river water level) instead of water table depth so we can actually compare with river water level, but maybe I'm being too picky if the aim of the figure is only to show the seasonal variations.

**Response:** *This is a good comment. We have shown the groundwater level, water table depth, river water level in Fig. 3. We have also added the elevation of riverbed (26.0 m) as well as the riparian ground surface elevation (29.5 m) in the captions in order to indicate the groundwater level. The revised Fig. 3 is as follows:*

[Figure]

*Figure 3: Seasonal variations of the river water level and depth of water table (WTD)/groundwater level (GWL) at distances of 5 m, 20 m, and 45 m away from the riverbank during the observation period in 2019 (a) and 2021 (b). The red arrow indicates the riparian ground surface level (29.5 m). The riverbed level is 26 m.*

127. Comment:

Figure 4: Nice flowchart, the reference is missing in the text.

**Response:** *Thanks for your positive comment. We have added the reference of Figure 4 in the text.*

128. Comment:

Figure 5: This is again a very nice figure. I would check and correct the wording of the caption (3 first sentences). Maybe try to increase the front size? I would use xylem instead of stem, as suggested before.

**Response:** *Thanks for your positive comment. We have checked and corrected the wording of the three first sentences in the caption. We also increased the front size and changed "stem water" to "xylem water".*

[Figure]

*Figure 6: Dual-isotope ($\delta^2H$ and $\delta^{18}O$) biplots of different water bodies in the three plots (D05, D20, and D45) for the observation years 2019 and 2021. The local meteoric water line (LMWL) was fitted by precipitation isotopes for each year. The soil water line (SWL) was fitted by the soil water isotopes in the four layers across three plots (D05, D20, and D45) for each year. D05, D20, and D45 are the plots at distance of 5 m, 20 m, and 45 m away from the riverbank, respectively. The error bars indicate standard deviations.*

129. Comment:

Figure 6: Very nice figure, I would however correct the caption with "Seasonal variations in the (proportional) contributions of soil water and groundwater to riparian trees in the three plots.". Try to increase the front size.

**Response:** *Thanks for your positive comment. We have changed the confusing sentences "Seasonal variations in the proportional contributions of soil water and groundwater to riparian trees in the three plots…" to "Seasonal variations in the proportional contributions of soil water and groundwater to riparian trees in the three plots (D05, D20, and D45) for the observation years*

*2019 (a−c) and 2021 (d−f).". And we also increased the front size.*

[Figure]

*Figure 7: Seasonal variations in the proportional contributions of soil water and groundwater to riparian trees in the three plots (D05, D20, and D45) for the observation years 2019 (a−c) and 2021 (d−f). D05, D20, and D45 are the plots at distance of 5 m, 20 m, and 45 m away from the riverbank, respectively. The error bars indicate standard deviations.*

130. Comment:

Figure 7: Nice figure, correct the "in-situ" in the legend at the top of the figure and the y axis "contributions of water sources to riparian soil water in the 80-170 cm layer in the three plots", check the grammar in the caption. I would increase the front size as well.

**Response:** *We have corrected the "in-situ" in the legend at the top of the figure and the y axis "contributions of water sources to riparian soil water in the 80-170 cm layer in the three plots". We also modified the grammar in the caption and increased the front size.*

[Figure]

*Figure 8: Seasonal variations in the different water source contributions to riparian deep soil water in the 80−170 cm layer in the three plots (D05, D20, and D45) for the observation years 2019 (a−c) and 2021 (d−f). D05, D20, and D45 are the plots at distance of 5 m, 20 m, and 45 m away from the riverbank, respectively. The error bars indicate standard deviations.*

131. Comment:

Figure 8: Nice figure as well, just correct the "in-situ" in the legend, check the caption and try to increase the front size.

**Response:** *We have modified the "in-situ" in the legend as well as the grammar in the caption. We also increased the front size (see the Figure 9 in the response to Comment #1).*

132. Comment:

Figure 9: Nice and clear figure, correct the y axis "contributions of river water to riparian trees", I would try to better highlight the yearly average, maybe using bold? I did not notice it at first look. Check and correct the caption: "different letters show a significant difference in the RWC to riparian

trees between two plots", maybe say "RWC to riparian trees in the three plots for each sampling campaign" rather than "seasonal variation" since you only show the statistical results of the differences between plots for a same campaign. The stats results for September 2021 look weird, D45 looks different than D05 and D20 not different from D05 and D45, can you check that?

**Response:** *We have corrected the y axis "contributions of river water to riparian trees", the caption "different letters show a significant difference in the RWC to riparian trees between two plots", and "RWC to riparian trees in the three plots for each sampling campaign". We highlighted the yearly average using bold fond. Additionally, we have modified the stats results of September 2021 as suggested (see the Figure 10 in the response to Comment #1).*

133. Comment:

Figure 10: Very nice figure as well, maybe just try to increase the front size. Check the grammar in the caption.

**Response:** *Thanks for your positive comment. We have increased the front size of the caption and modified the grammar in the caption.*

**Response to Referee #2 (Dr. Remy Schoppach):**

**General comments:**

1. Comment:

This manuscript from Li et al. aims at quantifying the contribution of river water to the transpiration flux of trees growing in the riparian area. This is an important topic, of interest for the community. Globally, the paper quality suffers from a clear lack of structure, elusive objectives and a poor discussion. Figures are of relatively good quality. Clearly the authors are not native speakers, but the paper remains easily readable from a language perspective, except for the discussion. The effort put in the language is appreciated but the structure and the reasoning need to be substantially improved.

**Response:** *Thank you for positive comments and insightful suggestions. We have substantially revised the structure, objectives, discussion as well as the wording and grammar mistakes throughout the manuscript. We have reorganized the structure especially in the Introduction, Materials & methods, Results, and Discussion sections.*

*In the Introduction, we have made a straightforward flow of ideas leading to a scientific gap that this paper aims to fill. Some sentences which belong to the Introduction rather than M&M have been moved into the Introduction section. We have introduced the use of radon as an indicator and the need of an iteration method (See the response to Comment #2).*

*In the M&M and Result sections, we have added a separate part of "3.1 Hydro-meteorological conditions" and moved all the hydro-meteorological conditions in the M&M part to the 3.1 section.*

*We have reorganized the entire Discussion. Firstly, we discussed the strengths/weakness and implications of the MixSIAR model and the iteration method. Secondly, we discussed the RWC to riparian trees and the effects of the distance from the stream and dry/wet year on RWCs to riparian trees. In this part, we have discussed and developed the potential processes. Finally, we discussed the link between RWC/WUE/WTD and developed its implications on management of riparian forest and river runoff. We have compared the results with previous work and provided corresponding explanations throughout the revised Discussion section. The discussion section has been revised thoroughly by means of developing the potential processes, reasons, and implications of our findings.*

2.    Comment:

Introduction lack of reasoning on the scientific gap. In some parts, it really reads like a discussion where the authors compare contrasting results from the literature without highlighting the questions it raises. Some concepts are not even introduced (e.g., the use of radon as an indicator or the need of an iteration method). The introduction requires a straight flow of ideas leading to a scientific gap that this paper aims to fill. The lack of structure is also visible as some parts of the introduction are displayed in the M&M.

**Response:** *Thank you for positive comments and insightful suggestions. We have reorganized the Introduction section and improved the flow of ideas leading to a scientific gap that this paper aims to fill. We have highlighted the questions it raised rather than displayed or compared different results from the literature. We added more introductions about the use of radon as an indicator and the need of an iteration method. Some sentences which belong to the Introduction rather than the M&M have been moved into the Introduction.*

**Specific comments:**

**Abstract**

3.    Comment:

Line 16: write active. We propose instead of "were proposed"

**Response:** *We have changed this sentence to "We proposed a new iteration method in combination with the MixSIAR model to quantify the proportional river water contribution (RWC) to riparian S. babylonica and its correlations with the depth of water table (WTD) as well as leaf $\delta^{13}C$."*

4.    Comment:

Line 19: contributed by

**Response:** *We have corrected it as suggested.*

5.    Comment:

Line 20: why using "but" instead of "and". Is the increase in river water acquisition in contradiction with the decrease in leaf $\delta^{13}C$? If yes, you need to explain why.

**Response:** *We have changed "but" to "and". The increase in river water acquisition is not in contradiction with the decrease in leaf $\delta^{13}C$.*

6. Comment:

There is no explanation of the decrease in leaf $\delta^{13}C$ the abstract?

**Response:** *We have explained the decrease in leaf $\delta^{13}C$ in the Abstract. Here is the revised part: "Significantly increasing river water acquisitions (by 7.0%) and decreasing leaf $\delta^{13}C$ (by −2.0‰) of riparian trees were observed as the WTD changed from 2.7 m in dry 2019 to 1.7 m in wet 2021 (p < 0.05). The lower water availability in drier year probably resulted in the plant stomatal closure to minimize the water loss, which consequently enhanced the leaf $\delta^{13}C$."*

7. Comment:

Line 24. How the rising water table would "stimulate" trees to maximize transpiration? I think this is simply the results of a higher water availability leading to a less negative water potential in the root-zone and subsequent lower stomata regulation. There is no clue for any "stimulation" of the plant?

Moreover, the reasoning is a bit ambiguous as you say that a rising water table increases the transpiration and the water extraction from the river (but not from the groundwater?). This is puzzling to me.

**Response:** *We have deleted the words "stimulate riparian trees to maximize transpiration water consumptions". Riparian trees can take up groundwater directly, while the river water seeped into the groundwater is an indirect water source for riparian trees.*

*We have clarified and changed this sentence to "The rising water table would trigger riparian trees to increase the water uptake from the groundwater/river water and show a consumptive water use strategy, which could not be recommended in order to both protect rivers and riparian vegetation."*

8. Comment:

Line 27: why using capital letters for Groundwater-Soil-Atmosphere Continuum?

**Response:** *We have corrected as "groundwater-soil-atmosphere continuum".*

**Introduction**

9. Comment:

Line 30: I'm not aware of a consequence of groundwater overexploitation on the alteration of precipitation regime. Maybe consider re-writing.

**Response:** *We have deleted the "precipitation regime".*

10. Comment:

Line 35: contributed "by"

**Response:** *We have changed "contributed 40%" to "contributed by 40%".*

11. Comment:

Line 38: delete "deeply"

**Response:** *We have deleted "deeply".*

12. Comment:

Line 39: their responses to the variation in the water table (response of what, transpiration? Variation in what, level?) Please be specific.

**Response:** *We have changed this sentence to "Therefore, understanding what water sources and how much river water are used by riparian trees as well as the responses of plant water use characteristics to groundwater level variations can help to control the river runoff and tree water requirement of revegetated riparian zones."*

13. Comment:

Line 39: Also, I don't understand how a "deep understanding" could help "balancing the river flow and the revegetated riparian species"? It could help implementing management strategies maybe? But the understanding will not balance anything.

**Response:** *We have deleted "balance" in this sentence. We changed this sentence to "Therefore,*

*understanding what water sources and how much river water are used by riparian trees as well as the responses of plant water use characteristics to groundwater level variations can help to control the river runoff and tree water requirement of revegetated riparian zones"*

14. Comment:

Line 39: Is "revegetated species" an already used term? I'm not native speaker but I don't think a species can be revegetated. A riparian area could be, but not a species, right?

**Response:** *We have changed "revegetated riparian species" to "revegetated riparian zones".*

15. Comment:

Line 41 to 43: contribution to what? Transpiration flux I guess, but please write it. Please also insert the references within the sentence after each corresponding argument.

**Response:** *We have changed "the river water contribution (RWC) to riparian trees" to "the river water contribution (RWC) to transpiration flux". We also inserted the references within the sentence after each corresponding argument. The revised part is as follows: "The statistical two- or multi-source linear mixing models (Ehleringer and Dawson, 1992; Alstad et al., 1999) and Bayesian mixing models (MixSIR, SIAR, SISUS, MixSIAR) (Ma et al., 2016; Wang et al., 2019b; White and Smith, 2020; Li et al., 2021) accompanied with stable water isotopes ($\delta^2H$ and $\delta^{18}O$) have been widely used to estimate the RWC to riparian trees."*

*References:*

*Alstad, K. P., Welker, J. M., Williams, S. A., and Trlica, M. J.: Carbon and water relations of Salix monticola in response to winter browsing and changes in surface water hydrology: an isotopic study using delta C-13 and delta O-18. Oecologia. 120, 375-385, 1999.*

*Ehleringer, J. R. and Dawson, T. E.: Water-uptake by plants-perspectives from stable isotope composition. Plant Cell and Environment. 15, 1073-1082, 1992.*

*Li, Y., Ma, Y., Song, X. F., Wang, L. X., and Han, D. M.: A δ2H offset correction method for quantifying root water uptake of riparian trees. Journal of Hydrology. 593, 125811, doi:10.1016/j.jhydrol.2020.125811, 2021.*

*Ma Y. and Song X. F.: Using stable isotopes to determine seasonal variations in water uptake of summer maize under different fertilization treatments. Science of the Total Environment. 550: 471-483, 2016.*

*Wang, J., Fu, B., Lu, N., Wang, S., and Zhang, L.: Water use characteristics of native and exotic shrub species in the*

*semi-arid Loess Plateau using an isotope technique. Agriculture, Ecosystems & Environment. 276, 55-63, 2019b.*

*White, J. C. and Smith, W. K.: Water source utilization under differing surface flow regimes in the riparian species Liquidambar styraciflua, in the southern Appalachian foothills, USA. Plant Ecology. 221, 1069-1082, 2020.*

16. Comment:

Line 45: "a separate water source". Separate from what, other sources? If yes, maybe list them.

**Response:** *We have changed "a separate water source" to "a direct potential water source for riparian trees".*

17. Comment:

Line 53: There was a debate or there is?

**Response:** *We have changed "There was a debate" to "There is a debate"*

18. Comment:

Line 63: inaccurate estimation of what? Please consider re-writing the entire sentence

**Response:** *We have rewritten these sentences and clarified the knowledge gap as follows: "There is growing evidence indicating that riparian trees at a certain distance away from the riverbank rarely took up river water directly, because their lateral roots could not reach the river (Mensforth et al., 1994; Thorburn and Walker, 1994). Nevertheless, riparian trees could indirectly utilize the river water seeped into riparian deep zone (including deep soil water and groundwater) when their roots tapped into the groundwater level (Mensforth et al., 1994; Wang et al., 2019b). Treating river water as a direct water source might lead to inaccurate estimations of the RWC to transpiration flux. However, it remains unclear that how to separate and quantify the contribution of the indirect river water source that recharges riparian deep water to transpiration flux of riparian trees nearby losing rivers."*

19. Comment:

Line 65: You can't just state that "how to separate and quantify the contribution of … is a great challenge" without reasoning it. Why is it a great challenge? If you don't explain it, the reader gets

confused and can only believe you. You need first to introduce the reasons making this a challenge.

**Response:** *We have added the explanation of why is "how to separate and quantify the contribution of ..." is a great challenge (see the response to Comment #18).*

20. Comment:

Line 70 to 80 reads like a discussion more than an introduction.

**Response:** *We have deleted most of the sentences from line 72 to 80 and focused on the scientific gap as well as its corresponding reasons. The modified sentences have been shown as follows: "The RWC to transpiration flux of riparian trees can be quantified indirectly by determining both the RWC to riparian deep water and the water use patterns of riparian trees. A multi-iteration method will help to calculate the proportional contributions of total (old and current) river water to riparian deep water, which could further improve the estimation accuracy of the RWC to tree transpiration flux. The radioactive isotope ($^{222}Rn$) has been widely used for tracing groundwater origins and the corresponding pathways in riparian zone (Close et al., 2014; Zhao et al., 2018). However, it is unclear about the residence time of recharged groundwater from river water and its effects on the RWC to plant transpiration flux. Moreover, the fluctuation of the depth of water table (WTD) in the riparian zone resulting from changing river water level plays a critical role in the RWC to riparian trees (Horton and Clark, 2001; Liu et al., 2017; Xia et al., 2018). Enhancing understanding of the quantitative relationship between WTD and the RWC to riparian trees will help to determine the optimal water table for maintaining the water source sustainability in the riparian zone beside a losing river. Nevertheless, little attention has been paid to quantifying the relationships between the RWCs to riparian trees and WTD.*

*Since leaf $\delta^{13}C$ values are positively related to tree WUE, the leaf $\delta^{13}C$ has been widely used as an indicator of tree WUE for C3 photosynthesis plants (Farquhar et al., 1989). For example, Thorburn and Walker (1994) found that the riparian Eucalyptus camaldulensis beside the ephemeral stream had higher tree WUE with more frequent access to river water based on the leaf $\delta^{13}C$ measurements. However, few studies focused on quantifying the relationships between leaf WUE and the RWC to riparian trees nearby a losing river."*

21. Comment:

Line 83: the first objective is the propose an iteration method. This comes from nowhere as no part of the introduction introduce the issue related to iteration methods? Also, what is an iteration method together with water stable isotopes?

**Response:** *This is a good comment. We have added the detailed information about the newly proposed iteration method in the introduction part. The added items are shown as follows: "The RWC to transpiration flux of riparian trees can be quantified indirectly by determining both the RWC to riparian deep water and the water use patterns of riparian trees. A multi-iteration method will help to calculate the proportional contributions of total (old and current) river water to riparian deep water, which could further improve the estimation accuracy of the RWC to tree transpiration flux."*

*We have changed "an iteration method together with water stable isotopes" to "an iteration method together with the MixSIAR model and water stable isotopes".*

**M&M**

22. Comment:

Line 94: has been seriously degraded instead of degraded seriously.

**Response:** *We have corrected it as suggested.*

23. Comment:

Line 95: What is ecological water? How is that "ecological water" supplied? This part of context is worth being developed a bit more. Is it via a systematic water release by dams?

**Response:** *We have modified this part as "Due to continuous drought and groundwater overexploitation, the Chaobai River dried up from 1999 to 2007. The "ecological water" (including reclaimed water, reservoir water, and diverted water by the South-to-North Water Transfer Project) has been supplied via a systematic water release by dams to restore this dry river since 2007. A total of 51.1 million and 380 million cubic meters of ecological water sources were released to the Chaobai River in 2019 and 2021, respectively."*

24. Comment:

Line 103: via a pressure stage gauge

**Response:** *We have corrected it as suggested.*

25. Comment:

Line 108: Fluctuated between

**Response:** *We have corrected it as suggested.*

26. Comment:

Line 109: The mean WTD in the three plots was significantly (p<0.5) deeper in 2019 (value) than in 2021 (value).

**Response:** *We have corrected it as suggested.*

27. Comment:

Line 105 to 110: Should this section displayed in the Results instead of M&M?

**Response:** *We have moved these sentences from M&M to the Results section. We added a separate part (3.1 Hydro-meteorological conditions) in the Results section and moved all the hydro-meteorological conditions in the M&M to the 3.1 section.*

28. Comment:

Line 118: There is not mention of the $^{222}$Rn in the introduction. Therefore, the reader has no idea what $^{222}$Rn is and why you determine the its concentration?

**Response:** *This is a good comment. We have presented the significance of determining the $^{222}$Rn concentration in the Introduction section in this study. The revised part is as follows: "The radioactive isotope ($^{222}$Rn) has been widely used for tracing groundwater origins and the corresponding pathways in riparian zone (Close et al., 2014; Zhao et al., 2018). However, it is unclear about the residence time of recharged groundwater from river water and its effects on the RWC to plant transpiration flux."*

*References:*

*Close M., Matthews M., Burbery L., Abraham P. and Scott D.: Use of radon to characterise surface water recharge*

*to groundwater. Journal of Hydrology. 53(2): 113-127, 2014.*

*Zhao D., Wang G., Liao F., Yang N., Jiang W., Guo L., Liu C. and Shi Z.: Groundwater-surface water interactions derived by hydrochemical and isotopic ($^{222}$Rn, deuterium, oxygen-18) tracers in the Nomhon area, Qaidam Basin, NW China. Journal of Hydrology. 565, 650-661, 2018.*

29. Comment:

Line 120: Is it Three trees in each of the three plots or one three per plot? Please re-write to avoid confusion.

**Response:** *We have modified this sentence to "One riparian S. babylonica tree was chosen in each plot for $\delta^2H$ and $\delta^{18}O$ measurements in xylem water as well as $\delta^{13}C$ analysis in plant leaves. The mean diameter at breast height of three sampled trees was 28.6 ± 4.4 cm."*

30. Comment:

Line 133: Did you measure the efficiency of the extraction and how? By weighting your fresh and dry samples? Please explain it.

**Response:** *We have added more information about how to calculate the efficiency of the extraction in this paragraph. For example: "We weighted all the xylem and soil samples before and after extraction and subsequently calculated the efficiency of water extraction in order to ensure the water extraction efficiency higher than 99% and no isotopic fractionation during water extraction process."*

31. Comment:

Line 141: Is there something missing in this sentence. What is exactly $C_{Air}$?

**Response:** *We have rewritten this paragraph to make it clear. "The radon ($^{222}$Rn) concentration in the groundwater and river water samples ($C_{Water}$, Bq/l) was determined based on the air $^{222}$Rn concentration values ($C_{Air}$, Bq/m$^3$) measured by a $^{222}$Rn monitor (Alpha GUARD PQ2000 PRO, Bertin Instruments, Germany). 100 ml of the water sample was slowly poured into the air-tight glass bottles and then purged with air in a closed gas cycle system. The air $^{222}$Rn concentration values in the $^{222}$Rn monitor ($C_{Air}$, Bq/m$^3$) were recorded at 10-minute intervals. The air inside the measurement set-up had contained a certain $^{222}$Rn concentration right before injecting the water sample ($C_{System}$, Bq/m$^3$). It is generally assumed that the already existing $C_{System}$ can be ignored*

*accordingly when $C_{System}$ is around or lower than 80 Bq/m³. In this study, more than four intervals were conducted to ensure that the $C_{System}$ was less than 80 Bq/m³. The measurement range of $C_{Air}$ was 2–2,000,000 Bq/m³ with a measurement precision of 3%. The above measured $C_{Air}$ value was not yet the ²²²Rn concentration in the measured water sample ($C_{water}$), because the ²²²Rn driven out had been diluted by the air within the ²²²Rn monitor and a small part of the ²²²Rn remained diluted in the watery phase."*

32. Comment:

Line 153 to 156: This is introduction, not M&M.

**Response:** *These sentences have been moved into the Introduction part.*

33. Comment:

Line 160: Based on what did you decided to not consider river water as a direct source for tree water uptake? You have to justify this choice one way or another. Or at least explain it.

**Response:** *We have added more information on the reason of why we did not consider river water as a direct water source for tree water uptake in the 2.4.1 section.*

*"Mensforth et al. (1994) and Thorburn and Walker (1994) characterized the outer projected edge of canopy as the extension range of lateral roots, which could indicate whether riparian trees take up river water directly or not. In this study, the outer projected edge of canopy was less than 5 m for riparian S. babylonica tree closest to the river (5 m away from the riverbank). It indicated that the lateral roots of S. babylonica trees could not tap into the river water. Therefore, the river water was not considered as a direct potential water source for tree water uptake, while the soil water in the 0−30 cm, 30−80 cm, and 80−170 cm layers, and groundwater were considered as the direct water sources for riparian S. babylonica."*

34. Comment:

Line 173: What is the correction proposed by Li et al. (2021). Please explain briefly how this works.

**Response:** *We have added the explanation on how the correction method –PWL correction method works in the section 2.4.1. Here is the revised part: "In this study, the $\delta^2H$ offsets between xylem water in riparian trees and its corresponding potential source water were also observed, which*

*could be explained by the $\delta^2H$ fractionation occurring in the plant water use processes (Li et al., 2021). These $\delta^2H$ offsets could lead to inaccuracy estimations in the MixSAIR model. In order to eliminate the $\delta^2H$ offsets of xylem water from its potential water sources, the measured xylem water $\delta^2H$ values were corrected via the potential water source line (PWL) proposed by Li et al. (2021). The PW-excess (PW-excess = $\delta^2H - a_p\delta^{18}O - b_p$; $a_p$ and $b_p$ were the slope and intercept of the PWL) was calculated to indicate the $\delta^2H$ deviation from the PWL, which was subsequently subtracted from the measured xylem water $\delta^2H$ values. The corrected $\delta^2H$ and raw $\delta^{18}O$ in xylem water were set as the mixture data in the MixSIAR model to quantify the contributions of direct water sources to riparian S. babylonica."*

35. Comment:

Line 183: Try to avoid saying "As shown in figS2a". Just write your sentence and refer to the figure at the end.

**Response:** *We have corrected this mistake throughout the manuscript.*

36. Comment:

Line 194: Please add a reference for Eq. 2 and for the coefficient values.

**Response:** *We have added a reference (Hoehn and Von Gunten, 1989) for the equation and the coefficient.*

*Reference:*

*Hoehn, E. and Von Gunten, H. R.: Radon in groundwater: A tool to assess infiltration from surface waters to aquifers. Water Resources Research. 25(8), 1795-1803, 1989.*

37. Comment:

Line 195: this is the first time you indicate that [222]Rn is Radon. Should come earlier in my opinion.

**Response:** *We have indicated that [222]Rn is Radon in the Introduction section.*

**Results**

38. Comment:

Line 262: was the lowest

**Response:** *We have corrected it as suggested.*

39. Comment:

Line 276: that is recharged

**Response:** *We have corrected it as suggested.*

40. Comment:

Line 282: were significantly higher

**Response:** *We have corrected it as suggested.*

41. Comment:

Line 285-286-287: Please consider re-writing

**Response:** *We have written these sentences as follows:*

*"The proportional contributions of river water to riparian S. babylonica trees were significantly higher in 2021 (mean of 23.8% ± 7.8%) than in 2019 (mean of 16.8% ± 4.7%) (p < 0.05). Specifically, the most significant difference in monthly RWC to riparian S. babylonica trees between dry 2019 and wet 2021 was up to 19.8% (p < 0.001). The maximum value of monthly RWC to S. babylonica trees was significantly higher in wet 2021 (35.2% ± 7.0%) compared with dry 2019 (24.2% ± 3.0%) (p < 0.05).*

*The riparian S. babylonica took up the most river water in July (35.2 ± 7.0%) and November (29.0 ± 5.0%) in 2021, whereas the highest RWC to riparian trees occurred in May (22.2 ± 1.7%) and June (24.2 ± 1.6%) in 2019. The minimum river water uptake for S. babylonica in 2021 was 17.7 ± 2.7% (in September), while riparian trees took up the least water in August 2019 (13.2 ± 1.9%). No significant seasonal-trend of the RWC to riparian trees was observed in both years (p > 0.05)."*

42. Comment:

Line 293: reference the corresponding figure

**Response:** *We have corrected as suggested.*

43. Comment:

Line 294: -27.7 is not remarkably larger than -29.7

**Response:** *The leaf $\delta^{13}C$ value of C3 plants generally ranged from -25.0‰ to -31.0‰. The statistical analysis showed that the leaf $\delta^{13}C$ in 2019 ($-27.7 \pm 1.0$ ‰) was significantly higher than in 2021 ($-29.7 \pm 0.7$ ‰) (p < 0.05).*

44. Comment:

Line 295: a significant increase

**Response:** *We have corrected it as suggested.*

45. Comment:

Line 297: before when (chose one)

**Response:** *We have deleted "when".*

46. Comment:

Line 303: This statement is dropped without any explanation. I suggest to move it to the discussion and to actually discuss it.

**Response:** *We have moved the last sentence to the Discussion section. Here is the revised sentence: "The riparian S. babylonica trees likely remained the highest WUE (i.e., utilize lower transpiration water to maximize $CO_2$ assimilation) as well as the lowest river water uptake proportion under the lowest water table condition (with the WTD of 4 m). In this study, the lower the groundwater level is, the more beneficial it is to optimize the riparian plant-water relations. Therefore, the relationships between the RWC to riparian trees, leaf-level physiological characteristics (e.g., leaf WUE) and hydro-meteorological conditions are critical and helpful for the better protection of the riparian forest while maintaining sustainable river runoff."*

Discussion

47. Comment:

Line 311: what are the interactions? Do you mean exchange of water?

**Response:** *In this study, there was only one process that the river water recharges the groundwater system along a losing river. We have deleted the confusing words "interactions" and "exchange" as well as clarified this process throughout the manuscript.*

48. Comment:

Line 311: "These contradictions". What contradictions?

**Response:** *We have deleted the "contradictions" and rewritten this part to make the story clearer. We have further discussed the potential explanations for the small RWC to riparian trees in the 4.2 section (see the response to the comment #1 by the Anonymous reviewer).*

49. Comment:

Line 312: might be due to that the. Please re-write

**Response:** *We have rewritten this sentence to make it clearer. Here is the revised sentence: "This probably indicated that the river water recharged mobile groundwater quickly but could not completely replace water held tightly in the soil pores (Brooks et al., 2010; Evaristo et al., 2015; Allen et al., 2019)."*

50. Comment:

Line 314: previous studies "showing". There is a word missing here

**Response:** *We have rewritten this sentence. The revised sentence is as follows: "It was consistent with Sprenger et al. (2019) who found that the lateral seepage of river water or rising water table could briefly saturate riparian soils but not entirely replace/flush immobile waters or isotopically homogenize different water pools."*

51. Comment:

Line 314: You mention previous studies, but cite only one.

**Response:** *We have deleted "previous studies" and changed this sentence to: "It was consistent with Sprenger et al. (2019) who found that the lateral seepage of river water or rising water table*

*could briefly saturate riparian soils but not entirely replace/flush immobile waters or isotopically*

*homogenize different water pools."*

52. Comment:

Line 319: The rising water table stimulated exchanges. Stimulated isn't an appropriate word here. Maybe triggered?

**Response:** *We have changed "stimulated" to "triggered".*

53. Comment:

Line 319: I don't understand this part of the discussion. It sounds obvious to me that if the water table reach 1.7 m below the surface it will exchange water with the soil layer standing at that same depth and in a larger proportion compared to a situation where the groundwater level is 1m deeper.

**Response:** *We have deleted this part of the Discussion, and discussed the potential processes that river water recharges riparian deep soil water and groundwater.*

54. Comment:

Line 326: Again, you mentioned previous studies and cite only one.

**Response:** *We have deleted this sentence.*

55. Comment:

Line 329: a smaller proportion than what?

**Response:** *We have changed "smaller" to "small".*

56. Comment:

Line 331: "is similar" and "Nearby perennial streams"

**Response:** *We have corrected it as suggested.*

57. Comment:

Line 334: what is "that" in the sentence?

**Response:** *We have deleted this sentence and rewritten this section.*

58. Comment:

Line 340: The authors actually did not measure the isotopic signature of the bound water in fine pore and neither the exchange between this water and the river water, so you can't state that it rarely exchanges. What you can do is to speculate and use this argument as an explanation.

**Response:** *This is a good comment. We have deleted the misleading statement (i.e., it rarely exchanges) and rewritten this part. The revision has been presented in the first paragraph of 4.2 discussion part (see the response to comment #1 by the previous Anonymous reviewer).*

59. Comment:

Line 348: significantly more river water. Actually no, you don't know. A higher proportion maybe, but I would bet that it is the opposite for the amounts.

**Response:** *We have deleted the "significantly more river water", because we did not measure the amounts of riparian tree transpiration flux. Only a higher proportion of the RWC to riparian trees in 2021 can be determined via the iteration method accompanied with the MixSIAR model. Nevertheless, we added the hydro-meteorological data (including evaporative demand indicated by VPD and net radiation) in this study (section 3.1). These hydro-meteorological data help us to speculate the trends of leaf-level characteristics (e.g., transpiration rate and photosynthetic rate). The details have been specified in the response to the comments #62, #63, and #64.*

60. Comment:

Line 356: The was a balance and coordination…. because obvious differences were found. What you did not find these differences? This entire sentence isn't clear and I don't see the point of it.

**Response:** *We have deleted this sentence and rewritten the whole section.*

61. Comment:

Line 359: If you say the tree grew more reliance on… you need to compare it with something. Otherwise, you should not use "more". More than what? More than who, another species?

**Response:** *We have deleted "more" or other comparison when we don't compare it with other thing throughout the manuscript.*

62. Comment:

Line 360 to 370: I don't understand how the authors can actually infer a profligate water use strategy whereas they have no ideas on the water fluxes. They have no transpiration measurements, no evaporative demand measurements. What about the radiation amount? This is a key variable driving water-use efficiency. If the radiation over 2021 growing season was twice less than the radiation in 2019, it is easily imaginable that the photosynthetic rate was substantially impacted and subsequently the WUE.

**Response:** *This is a good comment. We have added more descriptions of meteorological data in the Results section 3.1 and Figure S1. Extensive discussions about a consumptive water use strategy of riparian trees have been added in this study.*

*The average VPD during the observation period was significantly higher in 2019 (1.1 KPa) than in 2021 (0.9 KPa) (p < 0.05) (Fig. S1). Nevertheless, no significant difference in average net radiation during the observation period was found between two years (p > 0.05) (Fig. S1 a and b).*

*Higher leaf WUE associated with lower RWC to riparian trees and lower groundwater levels are likely due to that the water stress restricts the stomatal conductance and further reduces transpiration rate of riparian trees. Specifically, dry 2019 was characterized as higher water demand (indicated by higher VPD) and lower water availability compared with wet 2021, but the energy resource (indicated by net radiation) for riparian trees was similar between two years (Figs. S1 and S2). We supposed that the water limitation rather than energy limitation regulated the leaf-level stomatal conductance of riparian S. babylonica trees. The high water demands but low river water availability in dry year probably resulted in stomatal closure of riparian trees to minimize the water loss, which could eventually lead to a decrease of transpiration rate and even photosynthetic rate (Behzad et al., 2022; Fabiani et al., 2021). Aguilos et al. (2018) further found that the water stress would enhance radiation-normalized WUE, because the lack of water availability induced a more reduction in transpiration than photosynthesis. On account of no difference in the average net radiation between dry and wet years, the lower river water availability in dry year probably resulted in an increase of the leaf WUE. It can be inferred that riparian S. babylonica trees took up more river water and probably showed a consumptive water use strategy in wet year compared to dry year. This agreed well with previous studies that the woody plants showed lower leaf WUE and consumptive water-use patterns in rainy season, while they showed higher leaf WUE and*

*conservative water-use patterns with lower soil water availability in dry season (Behzad et al., 2022; Cao et al., 2020; Horton and Clark, 2001).*

*Despite the lack of transpiration flux measurements, we could infer that riparian S. babylonica took up more proportions of river water and probably showed a consumptive water use strategy in wet year compared with dry year based on the plant water uptake patterns, leaf WUE characteristics, and hydro-meteorological data (e.g., evaporative demand, net radiation, soil water content, and water table depth). Nevertheless, the lack of transpiration flux data did not allow us to determine the absolute amount of river water contribution for riparian S. babylonica trees. Further studies need to investigate the root water uptake patterns in combination with the transpiration flux to validate the water use strategy of riparian trees.*

63. Comment:

Line 378: In my opinion the argument developed here is misleading. The water table is low, likely because of the high evapotranspiration, which indicates a high transpiration rate, and very likely an even higher photosynthetic rate, leading to a high WUE. Under the same radiation conditions but with a much higher WTD, the WUE would be as high and probably even higher, and not the opposite as suggested by the authors.

**Response:** *This is a good comment. In this study, the dams along the Chaobai River are used to regulate the river water level especially in flood season, which has a great effect on the river runoff and thus the riparian groundwater level. A total of 51.1 million and 380 million cubic meters of ecological water sources were released to the Chaobai River in 2019 and 2021, respectively. Therefore, the significant difference in water table between dry 2019 and wet 2021 was mainly attributed to the significantly different amount of released ecological water between two years (see Fig. 3 in the response to Comment #126 by previous Anonymous reviewer).*

*The high water demands but low river water availability in dry year probably resulted in stomatal closure of riparian trees to minimize the water loss, which could eventually lead to a decrease of transpiration rate and even photosynthetic rate (Behzad et al., 2022; Fabiani et al., 2021). Aguilos et al. (2018) also found that the water stress would enhance radiation-normalized WUE, because the lack of water availability induced a more reduction in transpiration than photosynthesis (see the third paragraph in the response to Comment #62).*

64. Comment:

Line 411: That is wrong, the authors did not measure the transpiration so they don't know if a WTD of 4m minimize the plant transpiration.

**Response:** *We have deleted "the WTD of 4 m minimize the plant transpiration." We mainly discussed the relationship between WTD and the RWC to riparian trees and as well as its corresponding potential explanations. The revised discussion has been shown in the second and third paragraphs of "4.3 The link between RWC/WUE/WTD and its implications" (see the response to Comment #1 by previous Anonymous reviewer).*

**Figure**

65. Comment:

Fig 1.: I like the figure; it describes the sampling site very well. Please indicate what dotted line represents in right panel (I guess water table level). Also indicate if it represents a measured level or just a schematic representation.

**Response:** *Thanks for your positive comment. We have added the indication of dotted line on the right panel. The dotted line is a schematic representation of the water table.*

66. Comment:

Fig 3.: The figure indicates an average river water level of 29m. What does it mean exactly? What is the reference, the zero level? Could it be the distance from the river bed? Is it such a deep river? I think most large European rivers displayed a depth of 2 to 5 m. I've never been to this place but 29m deep sounds more like a lake than a river. BTW, how a 29m deep river can lose flow?

**Response:** *The figure indicates the elevation of river water level. The reference zero level is the sea water level. We have added the elevation of riverbed (26.0 m) as well as riparian ground surface elevation (29.5 m) in the captions. The river water depth is ranging from 1.9 m to 2.9 m in dry 2019, whereas it fluctuated between 1.7 m and 3.3 m in wet 2021.*

[Figure]

*Figure 3: Seasonal variations of the river water level and water table depth (WTD)/groundwater level (GWL) at distances of 5 m, 20 m, and 45 m away from the riverbank during the observation period in 2019 (a) and 2021 (b). The red arrow indicates the riparian ground surface level (29.5 m). The riverbed level is 26 m.*

67. Comment:

Fig 4.: The Flowchart and its symbols must be explained in the legend.

**Response:** *We have added the explanation of flowchart and its symbols in the legend.*

[Figure]

*Figure 5: Flowchart for quantifying the proportional contributions of river water to riparian trees. The $P_s$ and $P_g$ represent the contributions of riparian deep soil water in the 80−170 cm layer as well as groundwater to riparian trees, respectively. The $s_r^{t-1}$ and $g_r^{t-1}$ represent the proportional contributions of the old river water (before t-1) to riparian deep soil water in the 80−170 cm layer and groundwater, respectively. The $s_s^{t-1}$, $s_r^t$, and $s_g^t$ represent the proportional contributions of in-situ soil water in the 80−170 cm layer at t-1, river water during t-1 to t, and groundwater during t-1 to t for riparian deep soil water in the 80−170 cm layer at t, respectively. The $g_g^{t-1}$ and $g_r^t$ represent the proportional contributions of in-situ groundwater at t-1 and river water from t-1 to t for riparian groundwater at t, respectively.*

68. Comment:

Table 2: units of residence time

**Response:** *We have added the units of residence time in the captions.*

*"Table 2: The $^{222}$Rn values in river water, background groundwater and riparian groundwater in three plots (D05, D20, and D45), and the average residence time of groundwater ($T_{res}$, day) in 2021. The background groundwater represents groundwater in aquifers more than 100 m away from the riverbank."*

---

## Author Response (AR1)

Dear Dr. Markus Hrachowitz, editor of Hydrology and Earth System Sciences:

Thanks very much for giving us the opportunity to revise this paper. Upon your request, we have carefully revised our manuscript (hess-2022-327) entitled "Quantifying river water contributions to riparian trees along a losing river: Lessons from stable isotopes and iteration method" after considering all the comments made by you, Dr. Remy Schoppach and one anonymous reviewer. The comments have helped us greatly improve the overall quality of the manuscript. The following is the point-to-point response to all the comments. The page and line numbers in the following response refer to the revised manuscript with changes marked.

**Response to the editor (Dr. Markus Hrachowitz):**

Dear authors,

As you have seen, two reviewers have assessed your manuscript. They both appreciate the overall idea of the study. However, they also raise a number of critical concerns and point out that the manuscript is not sufficiently developed to meet the standards for publication in HESS. Not only are large parts of the manuscript, and in particular the Discussion, difficult to follow for the reader due to rather poor English, but also due to a lack of a clear and logical structure. In addition, and more importantly, the level of refection throughout the manuscript is very limited. For example, the scientific knowledge gaps and science questions to be addressed remain very vague. Similarly, the entire experiment but in particular the results are not placed into sufficient context with existing literature, leaving the discussion rather hollow. What do the results mean? What can be learned from that? And what are the wider implications of that (also with respect to existing literature)?

Without much deeper consideration of the above shortcomings, the manuscript has the flavor of a technical report of a rather limited case study.

For a revised version of the manuscript to be potentially considered for publication, the above points, in addition to all remaining reviewer comments, need to be addressed in depth and detail. I therefore strongly encourage the authors to invest quite some more effort to considerably improve on the above points, with special consideration of a much stronger reflection and discussion.

**Response:** *We are grateful to you and two reviewers for your constructive and valuable comments on our manuscript. We have substantially revised the manuscript according to the comments (see the replies to Anonymous Referee #1 and Referee #2 (Dr. Remy Schoppach) in detail).*

*The primary revisions include the following four aspects:*

1. *In the Introduction section, we have reorganized the structure and given a straightforward flow of ideas to address clearer objectives and scientific knowledge gaps of this study. Firstly, we have introduced the significance of determining what water sources and how much rive water is taken up by riparian trees nearby losing rivers (see P.2, Lines 40-52). Secondly, we have introduced a debate on whether the river water is a potential water source of riparian trees or not and how it becomes available to riparian plants (see P.2, Lines 40-52). Thus, we have highlighted the question that how to separate and quantify the contributions of indirect river water source (which recharges riparian deep soil water/groundwater) to riparian trees nearby losing rivers (see P.2, Lines 53 to P. 4, Line 89). Thirdly, we have introduced several methods (e.g. MixSIAR model, water tracers ($\delta^2H$, $\delta^{18}O$, and $^{222}Rn$), iteration method) to identify the water source contributions to riparian plants. We also addressed the need to combine these methods to give a more reliable quantification of the river water contribution (RWC) to riparian trees (see P.4, Lines 90-104). Fourthly, we have introduced the effects of the RWC to riparian trees on the leaf water use efficiency (WUE) and the critical role of the water table depth (WTD) in plant water use. Then we have raised the question that what is the quantitative relationship between the RWC, WUE and WTD? (see P.4, Lines 105 to P. 5, Line 135). Finally, we have provided three objectives of this study (see P.5, Lines 136 to P. 6, Line 142).*

   *We have highlighted the questions this study raised rather than displayed or compared different results from the literature. Some sentences which belong to the Introduction rather than the M&M have been moved into the Introduction section. (see P.3, Lines 77-84).*

2. *In the M&M section, we have added more details about the sample collection (see P.6, Lines 159 to P. 8, Line 203), isotopic analyses (see P.8, Lines 204 to P. 9, Line 243) and MixSIAR model (see P.10, Lines 259-269). We have explained how we separated the soil in 4 layers and averaged the soil water isotopic values for each layer (see P.10, Lines 272 to P. 11, Line 283). We further clarified the methods for quantifying the direct water source (soil water and groundwater) contributions to riparian trees (see P.10, Lines 272 to P. 12, Lines 307), the water source contributions to deep soil water and groundwater (see P.12, Lines 308-329), and the RWC to riparian trees (see P.12, Lines 330 to P. 14, Line 367), respectively.*

3. *In the Results section, we have added a separate part of "3.1 Hydro-meteorological conditions"*

*and moved all the hydro-meteorological conditions in the M&M section to the section 3.1 (see P.14, Lines 381 to P. 15, Line 401). We also presented the interannual differences (see P.17, Lines 466 to P. 18, Line 476) and seasonal differences (see P.18, Lines 477-485) in the RWC to riparian trees, respectively.*

4. *In the Discussion, we have reorganized and completely rewritten the entire section. Firstly, we have discussed the advantages and limitations of the MixSIAR model and the iteration method (see P.19, Lines 508 to P. 21, Line 556). In this part, we put more effort into explaining the feasibility of the methods used in this study, especially with consideration of their reflection and implications. Secondly, we discussed the potential processes of the RWC to riparian trees (see P.21, Lines 560 to P. 22, Line 597), and explaining the influence of the distance from the river and dry/wet year on the RWC to riparian trees (see P.22, Lines 598 to P. 24, Line 641). Finally, we discussed the link between RWC/WUE/WTD and developed its implications on management of riparian forest and river runoff (see P.24, Lines 642 to P. 27, Line 720). We have compared the results with previous work and provided corresponding explanations throughout the revised Discussion section. The entire experiment and in particular the results are placed into sufficient context with existing literatures to make the discussion more substantial. We have put quite more effort on the Discussion section by means of discussing the potential processes, reasons, and implications of our findings.*

*We have double checked and corrected the writing throughout this manuscript (especially discussion section). The co-author Lixin Wang (Professor in Indiana University-Purdue University Indianapolis (IUPUI), USA) has helped us to further edit and polish the language of the revised manuscript.*

**Response to Anonymous Referee #1:**

**General comments:**

1. Comment:

This manuscript investigates the river water contribution to riparian trees. It does it by (1) determining and quantifying the sources of riparian trees water in soil and groundwater, (2) determining the sources of deep soil water and groundwater (precipitation, in-situ soil and

groundwater, river water…) to finally (3) determine the proportion of river water that fed the riparian trees. This is an important work to better understand how riparian trees can affect stream flow and how we can better manage riparian zones in order to both protect rivers and riparian vegetation.

The manuscript is well-organized but I found the flow difficult to follow due to the grammar and wording mistakes throughout the manuscript (especially discussion section). The uncertainties of the MixSIAR model and iteration method are not really discussed, the MixSIAR model is also not well-enough presented in the method section. The discussion section is too light, I found that the discussion of the potential processes was too limited, as well as how do these results compare with other studies and why.

The figures are very good, there are just some grammar/wording mistakes in some captions.

**Response:** *Thank you for the positive comments and insightful suggestions. We have double-checked and revised the grammar and wording mistakes throughout the manuscript (including the figures). The iteration method and radon ($^{222}$Rn) concentration are supplemented in the Introduction section. We have added more details about the sample collection and isotopic analyses (see P. 6, Line 159 to P.9, Line, 243)) and more information on the MixSIAR model to the M&M section (See P. 10, Lines 259-269). We further clarified the methods for quantifying the direct water source contributions to riparian trees and the water source contributions to deep soil water and groundwater (see P.10, Lines 270 to P.12, Line 329). We have addressed the proposed iteration method to quantify the river water contribution (RWC) to riparian trees in detail (see P.12, Line 339 to P.14, Line 365). In the Results section, we have supplemented the hydro-meteorological conditions (e.g., VPD, net radiation, relative humidity, and temperature) (see P.14, Line 381 to P.15, Line 401, and Fig. S1). We substantially discussed the uncertainties of this model in the Discussion section (see P.19, Lines 508 to P.21, Lines 556). Especially for the major concerns, we have thoroughly revised the Discussion and discussed the potential processes. We have also compared our results with previous work and clearly explained the mechanisms.*

*The Discussion has been revised from the following three aspects. Firstly, we added the strengths/weaknesses and implications of the MixSIAR model and the iteration method. Secondly, we discussed the potential processes of the RWC to riparian trees, and further explained the effects of the distance from the stream as well as dry/wet year on RWC to riparian trees. Finally, we*

*discussed the link between RWC/WUE/WTD and its implications on management of rivers and riparian vegetation.*

**Specific comments:**

INTRODUCTION

2. Comment:

Line 41: I feel that the jump to paragraph 2 (contribution of river water to riparian trees) is too quick, maybe have a small paragraph beforehand presenting the different sources of water for riparian trees (groundwater, soil water, river water?) first?

**Response:** *We have added a sentence "The potential water sources of riparian trees along a losing river are generally considered as a mix of soil water at different depths, groundwater, and river water (Alstad et al., 1999; White and Smith, 2020)." at the beginning of the second paragraph (see P. 2, Lines 53-54). Then it shifts to introduce the debate on whether river water is a potential water source for riparian trees or not and how it becomes available to plants.*

3. Comment:

Line 41: Can you be more specific about "data comparison"? It is unclear.

**Response:** *The "data comparison" refers to the direct comparison of stable isotopic values between plant stem water and different water sources. This method could determine the possible root water uptake depth when the stem water isotope is same as the isotopic value of soil water at a certain depth. We have specified the words "data comparison" and changed this sentence to "The graphical inference and direct comparison of stable isotopic values between plant stem water and different water sources (Dawson and Ehleringer, 1991; Busch et al., 1992; Costelloe et al., 2008; Zhao et al., 2016), statistical two- or multi-source linear mixing models (Alstad et al., 1999; Zhou et al., 2017), and the MixSIAR Bayesian mixing model (Wang et al., 2019a; Wang et al., 2020; White and Smith, 2020; Li et al., 2021) integrated with stable isotopes ($\delta^2H$ and $\delta^{18}O$) have been widely used to identify the potential water sources taken up by riparian trees."(see P. 4, Lines 90-95).*

**M&M**

4. Comment:

Lines 112-125: I think you should develop on the storage in the field and until you put the samples in the fridge (where/in what did you stored the samples? what method did you use to limit evaporation for the water samples?), this is important and not really described in this section. Also, use the (full) technical name of the tools you used in the field for sampling (e.g., "hydrophore"?) or storage of samples (e.g., "brown bottle" is unclear).

**Response:** *Thanks for your professional suggestions. We have added the description of the field storage procedure. All the samples (river water, groundwater, soil samples, plant stem samples, plant leaf samples and precipitation) were put into the refrigerating box with several ice bags to avoid evaporation effects in the field. And all the samples were delivered to the laboratory and stored at 4°C/-10°C in the refrigerator (see P. 7, Lines 181-183 and P. 7, Line 193 to P. 8, Line 195). We have provided the full technical names of the tools used in the field for sampling or storage of samples. For example, we have changed "hydrophore" to "plexiglass hydrophore water sample collector with capacity of 1L" (see P. 7, Line 177). The "brown bottle" has been changed to "100-ml brown glass vials" (see P. 7, Line 184).*

5. Comment:

Line 115: I am not sure what is it, maybe be more specific and use the full technical name? Maybe "plexiglass hydrophore water sample collector"? But I am not sure about what you used.

**Response:** *We have changed the "hydrophore" to a more specific technical name "plexiglass hydrophore water sample collector with capacity of 1L" (see P. 7, Line173).*

6. Comment:

Line 115: Why 135 precipitation samples? Before you said precipitation was sampled during the 12 campaigns, I would clarify this point. Also, what was the frequency of sampling? And the location?

**Response:** *We apologize for the oversight. In fact, precipitation was sampled after each precipitation event and not merely collected during the 12 campaigns. A total of 135 precipitation samples were collected throughout the whole years of 2019 and 2021 in order to get the local meteoric water line (LMWL) in the dual-isotope plot. We have changed this sentence to "Precipitation was sampled after each precipitation event via a device consisting of a funnel, a*

*polyethylene bottle, and a ping-pong ball. A total of 135 precipitation samples were collected*

*throughout the whole years of 2019 and 2021." (see P. 7, Lines 178-181).*

7. Comment:

Lines 120-121: So 3 trees for each plot?

**Response:** *One tree was sampled for each plot. We have changed this sentence to "One riparian S.*

*babylonica tree was chosen in each plot for $\delta^2H$ and $\delta^{18}O$ measurements in xylem water as well as*

*$\delta^{13}C$ analysis in plant leaves. The mean diameter at breast height of three sampled trees was 28.6 ±*

*4.4 cm." (see P. 7, Lines 186-189).*

8. Comment:

Line 122: What do you mean by "several"? Was not 3/plot? Did you take several samples in the

same tree? I would use "sampled" rather than "cut". How did you sample the stem? What tool did

you use? Also, correct to "[…], we removed the bark.".

**Response:** *"Several stems" means that five mature and suberized stems were taken from the same*

*tree in each plot. We have changed "cut" to "sampled". An averruncator with the length of 5 m was*

*used to sample the stem. We have changed the confusing sentence "Several suberized stems were*

*firstly cut from riparian S. babylonica, removed the bark and phloem, and then stored at −10 ℃*

*until water isotope analysis." to "Five mature and suberized stem samples were taken from the same*

*riparian S. babylonica tree in each plot using an averruncator with the length of 5 m. We removed*

*the bark and phloem of the sampled stems, and then put the remaining xylem samples into a 12-ml*

*brown glass vial sealed with parafilm." (see P. 7, Lines 189-192)*

9. Comment:

Line 123: Maybe specify: "stored the remaining xylem at.."? Where/in what did you store it? Glass

vial?

**Response:** *We have changed this sentence to "We removed the bark and phloem of the sampled*

*stems, and then put the remaining xylem samples into a 12-ml brown glass vial sealed with parafilm."*

*(see P.7, Lines 190-192) and "The remaining xylem and mature leaf samples were stored in a*

*refrigeration box with several ice bags in the field. Then the xylem samples were transported to the*

*laboratory and kept in a refrigerator at −10 ℃ before water extraction and isotope analysis." (see*

*P. 7, Line 193 to P.8 Line 195).*

10. Comment:

Line 124: In what did you stored the leaves before drying them?

**Response:** *The mature leaf samples were stored in a refrigeration box with several ice bags in the field until they were transported to the laboratory (see P. 7, Line 193 to P.8 Line 195). The mature leaves were oven-dried at 65 ℃ for 72 h on the day of sampling. (see P. 8, Lines 195-197)*

11. Comment:

Lines 126-130: This is a great section with the details needed (sampling method, storage). Can you clarify: you sampled soil only at 1 location for each plot, right? I would add a reference for the oven-dry method.

**Response:** *Yes, we collected the soil sample only at one location for each plot. We have clarified that "Soils at depths of 0−5, 5−10, 10−20, 20−30, 40−60, 60−80, 90−110, 150−170, 190−210, 250−270, and 280−300 cm in one soil profile near the selected S. babylonica trees were sampled by a power auger (CHPD78, Christie Engineering Company, Sydney, Australia)." (see P. 8, Lines 198-200). We have added two references for the oven-dry method as follows "One portion of each soil sample was put into a 12-ml brown glass vial and stored at −10 ℃ before water stable isotope analysis, and the other portion was packed into an aluminum box for gravimetric soil water content (SWC) measurement via the oven-drying method (Wang et al., 2019b; Li et al., 2021)." (see P. 8, Lines 200-203).*

*References:*

*Li, Y., Ma, Y., Song, X. F., Wang, L. X., and Han, D. M.: A $\delta^2H$ offset correction method for quantifying root water uptake of riparian trees. Journal of Hydrology. 593, 125811, doi:10.1016/j.jhydrol.2020.125811, 2021.*

*Wang, J., Fu, B., Lu, N., Wang, S., and Zhang, L.: Water use characteristics of native and exotic shrub species in the semi-arid Loess Plateau using an isotope technique. Agriculture, Ecosystems & Environment. 276, 55-63, 2019b.*

12.  Comment:

Line 150: Do you have a reference to support this choice for k?

**Response:** *We have added the supporting reference for the parameter "k" (Clever, 1985) (see P. 9, Lines 232-233).*

*The diffusion coefficient 'k' indicates the $^{222}$Rn concentration relationship between a watery phase and the air volume existing above. This relation is dependent on the temperature. With falling temperatures, the quantity of radon soluble in water increases. For example, the diffusion coefficient 'k' increases when temperature drops. The dependency for temperature of the diffusion coefficient for the transition of the phase water/air is presented in Fig. A1.*

*The influence of the radon diffusion coefficient 'k' is low in the temperature range between 10° C and 30° C. A diffusion coefficient k of 0.26 can be applied within the specified temperature range for a typical room temperature of 20° C (Fig. A1).*

[Figure]

*Figure A1. Temperature dependency of the diffusion coefficient 'k' (Clever, 1985)*

*Reference:*

*Clever, H. L.: Solubility Data Series. Vol.2, Krypton-, Xenon, Radon Gas Solubilities, p. 463-468, Pergamon Press, Oxford, 1985.*

13. Comment:

Lines 168-169: So you calculated the average isotopes values for each of the four layers? You measured soil water isotopes at 11 depths.

   **Response:** *In this study, we measured soil water isotopes at 11 depths in the three plots. In order to reduce errors in the analytical procedure, four soil layers (0−30, 30−80, 80−170, and 170−300*

*cm) were ued to identify the main root water uptake depth of riparian trees according to the seasonal variations in the SWC, water isotopes and WTD. The average soil water isotopes values at depths of 0−5 cm, 5−10 cm, 10−20 cm, and 20−30 cm were calculated for the 0−30 cm soil layer, because the water isotopes went through strong evaporation and SWC varied significantly seasonally. The soil water isotope values at depths of 40−60 cm and 60−80 cm were averaged for the 30-80 cm soil layer, and those values at 90−110 cm and 150−170 cm depths were averaged for the 80−170 cm soil layer since the water isotopes and SWC were relatively stable. The average isotopic values of soil water at deep depths (190−210 cm, 250−270 cm, and 280−300 cm) were calculated for the 170−300 cm soil layer, which varied with the fluctuations of groundwater levels. (see P. 10, Line 272 to P.11, Line 283)*

14. Comment:

Line 175: I don't understand "set as the mixture data", please clarify. Maybe more info on the MixSIAR model would help, I find it a bit frustrating for the reader to have to look in other papers. It would also help to understand why you talk about sampling times later in the text (lines 184-188, for example). Maybe have a separate section just to present the model (before 2.4.1.)?

**Response:** *We have added more detailed information on the MixSIAR model before the section 2.4.1 "The MixSIAR model is a Bayesian mixing model which can be integrated with stable isotopes to quantify the proportions of source contributions to a mixture (Stock and Semmens, 2013). The input data of the MixSIAR model include mixture data, source data, and discrimination data. In this study, the mean and standard deviation (SD) of the isotopic values of each water source for riparian trees/riparian deep water were input as source data into the MixSIAR, while the measured isotopic values of xylem water/riparian deep water were input as raw mixture data into the MixSIAR. The discrimination data for both $\delta^2H$ and $\delta^{18}O$ were set to zero, because the input $\delta^2H$ and $\delta^{18}O$ values in the MixSIAR were non-fractionated or $\delta^2H$-corrected. The Markov Chain Monte Carlo parameter was set to the run length "very long". Both the trace plots and three diagnostic tests (i.e., Gelman–Rubin, Heidelberger–Welch, and Geweke) were used to determine whether the MixSIAR model converged or not (Stock and Semmens, 2013). Then, the mean and SD values of different water source contributions could be estimated with the MixSIAR model." (see P. 10, Lines 259-269)*

15.  Comment:

Lines 185 and 187: So when you say "soil water in 0-80 cm" and "soil water in 0-170 cm": does the model takes the layer as a whole or does it still account for the different soil layers 0-30, 30-80, 80-170 cm? And how do you integrate the soil water isotopes measured at 11 soil depths in the model? I think more information about that is needed in the manuscript (especially for the readers - like me - who are not familiar with the model).

**Response:** *When we state "soil water in 0-80 cm" and "soil water in 0-170 cm", it still accounts for the different soil layers of 0-30 cm, 30-80 cm, and 80-170 cm in the MixSIAR model. The proportional contributions of soil water in the 0-30 cm, 30-80 cm, and 80-170 cm layer for riparian deep water were estimated separately in the MixSIAR model. We finally add up the proportional contributions of different soil layers (see P. 10, Line 272 to P.11, Line, 283).*

*More information for integrating the soil water isotopes measured at 11 soil depths in the model has been added as follows: "The average soil water isotopes values at depths of 0−5 cm, 5−10 cm, 10−20 cm, and 20−30 cm were calculated for the 0−30 cm soil layer, because the water isotopes went through strong evaporation and SWC varied significantly seasonally. The soil water isotope values at depths of 40−60 cm and 60−80 cm were averaged for the 30-80 cm soil layer, and those values at 90−110 cm and 150−170 cm depths were averaged for the 80−170 cm soil layer since the water isotopes and SWC were relatively stable. The average isotopic values of soil water at deep depths (190−210 cm, 250−270 cm, and 280−300 cm) were calculated for the 170−300 cm soil layer, which varied with the fluctuations of groundwater levels." (see P. 10, Line 272 to P.11, Line 283)*

16.  Comment:

Line 194: Can you provide a reference for the decay coefficient?

**Response:** *We have added a reference for the decay coefficient as "where λ represents the decay coefficient (0.181 day$^{-1}$) (Hoehn and Von Gunten, 1989)." (see P. 9, Line 238)*

*Reference:*

*Hoehn, E. and Von Gunten, H. R.: Radon in groundwater: A tool to assess infiltration from surface waters to aquifers. Water Resources Research. 25(8), 1795-1803, 1989.*

17.  Comment:

Line 195: How was this value determined?

**Response:** *We have added more information about the determination of $C_e$. The $C_e$ represents the [222]Rn concentration of background groundwater when the equilibrium between radon production and decay is reached. The measuring [222]Rn concentration of groundwater in aquifers more than 100 m away from the riverbank remains constant in this study (with an average value of $7400.0 \pm 35.4$ $Bq/m^3$), suggesting that $C_e$ can be defined as $7400.0$ $Bq/m^3$. (see P. 9, Lines 238-242)*

18. Comment:

Lines 205-222: It is an important method here but I found this part difficult to follow. There are A LOT of symbols, maybe use a table instead of the 7 lines of text so it's more readable? I also did not remember what was Ps and Pg, I would explain these terms here again. I would suggest to change some of the terms, or try to include the time as subscript (t-1, t) so it's easier to follow.

**Response:** *This is a good comment. We have simplified this equation to make it more readable (see P. 13, Lines 343-350). The Eq (3) has been changed to:*

*"$RWC = P_s * S_r + P_g * G_r$*

$$= P_s*(s_r^t + s_r^{t-1}) + P_g*(g_r^t + g_r^{t-1})$$

$$= P_s*(s_r^t + s_r^t*s_s^{t-1} + s_r^t*(s_s^{t-1})^2 + s_g^t*g_r^t + s_g*g_r^t*g_g^{t-1} + s_g^t*g_r^t*(g_g^{t-1})^2) + P_g*(g_r^t + g_r^t*g_g^{t-1} + g_r^t*(g_g^{t-1})^2)$$

$$= (P_s*s_r^t + P_g*g_r^t + P_s*s_g^t*g_r^t) + (P_s*s_r^t*s_s^{t-1} + P_g*g_r^t*g_g^{t-1} + P_s*g_r^t*s_g^t*g_g^{t-1}) + (P_s*s_r^t*(s_s^{t-1})^2 + P_g*g_r^t*(g_g^{t-1})^2 +$$
$$P_s*s_g^t*g_r^t*(g_g^{t-1})^2) \qquad\qquad (3)"$$

*In order to make the text clearer, we added a supplement Table S1 to display the abbreviations of all the variables (See Table S1). In addition, we have explained the $P_s$ and $P_g$ terms here again. And we modified the terms that has been used to represent time information as subscript (t-1, t) to make them clearer.*

**Table S1 Acronym dictionary**

| | |
|---|---|
| *RWC* | *River water contribution* |
| *WUE* | *Leaf-level water use efficiency* |
| *WTD* | *Water table depth* |
| *T* | *Temperature* |
| *RH* | *Relative air humidity* |

| | |
|---|---|
| $ET_0$ | Reference evapotranspiration |
| VPD | Vapor pressure deficit |
| SWC | Soil water content |
| IRIS | Isotopic ratio infrared spectroscopy system |
| IRMS | Isotope Ratio Mass Spectrometry system |
| VSMOW | Vienna Standard Mean Ocean Water |
| $C_{Water}$ | $^{222}Rn$ concentration of the water samples |
| $C_{Air}$ | Air $^{222}Rn$ concentration of the water samples |
| $C_{System}$ | Air $^{222}Rn$ concentration of the measurement system |
| $V_{System}$ | The interior volume of the measuring set-up |
| $V_{Sample}$ | The volume of water sample |
| $T_{res}$ | The average residence time of recharged groundwater from river water |
| $k$ | The $^{222}Rn$ distribution coefficient of water/air |
| $\lambda$ | The decay coefficient |
| $C_e$ | The $^{222}Rn$ concentration of background groundwater when the equilibrium between radon production and decay is reached |
| $C_r$ | The $^{222}Rn$ concentration of river water |
| $C_g$ | The $^{222}Rn$ concentration of riparian groundwater |
| PWL | The potential water source line |
| $a_p$ | The slope of the PWL |
| $b_p$ | The intercept of the PWL |
| $PW_{excess}$ | The $\delta^2H$ deviation of riparian tree xylem water from the PWL |
| $S_r$ | The total RWC to riparian deep soil water in the 80−170 cm layer (throughout the river losing-flow period since 2007) |
| $G_r$ | The total RWC to riparian groundwater (throughout the river losing-flow period since 2007) |
| $P_s$ | The contribution of riparian deep soil water in the 80−170 cm layer to riparian trees |
| $P_g$ | The contribution of riparian groundwater to riparian trees |
| $s_r^{t-1}$ | The proportional contribution of the old river water (before t-1) to riparian deep soil water in the 80−170 cm layer |
| $g_r^{t-1}$ | The proportional contribution of the old river water (before t-1) to |

| | riparian groundwater |
|---|---|
| $s_s^{t-1}$ | *The proportional contribution of in-situ soil water in the 80−170 cm layer at t-1 to riparian deep soil water in the 80−170 cm layer at t* |
| $s_r^t$ | *The proportional contribution of river water from t-1 to t to riparian deep soil water in the 80−170 cm layer at t* |
| $s_g^t$ | *The proportional contribution of groundwater from t-1 to t to riparian deep soil water in the 80−170 cm layer at t* |
| $g_g^{t-1}$ | *The proportional contribution of in-situ groundwater at t-1 to riparian groundwater at t* |
| $g_r^t$ | *The proportional contribution of river water from t-1 to t to riparian groundwater at t* |
| ANOVA | *One-way analysis of variance* |
| LMWL | *Local meteoric water line* |

**RESULTS**

19. Comment:

Lines 234-235: I think the slope of the LMWL only gives an indication about how oxygen and deuterium co-evolve, it does not indicate if the value is high or low. You can have 2 measurements of lower isotopes values but still have the same slope as with 2 measurements of higher isotopes values.

**Response:** *The slope of the LMWL indeed mainly gives an indication of the evaporation degree of water samples. We have changed this sentence to "The slope of the local meteoric water line in 2021 (7.8) was significantly higher than in 2019 (5.5) ($p < 0.05$), which suggested that the falling raindrops undergone stronger sub-cloud evaporation in 2019 (Zhao et al., 2019)." (see P. 15, Lines 405-408).*

*Reference:*

*Zhao, L., Liu, X., Wang, N., Kong, Y., Song, Y., He, Z., Liu, Q. and Wang, L.: Contribution of recycled moisture to local precipitation in the inland Heihe River Basin. Agricultural and Forest Meteorology. 271, pp.316-335, 2019.*

20. Comment:

Line 276: How did you get a negative residence time? Would it make more sense to use "0"?

**Response:** *It will make more sense to use "0" instead of a negative residence time, due to the fact that the river water recharges groundwater frequently in this study. We have changed all the negative residence time values to "0" throughout the text (see P. 17, Line 453) and in Table 1. The confusing sentence "As shown in Table 2, there was a significant increase of $^{222}$Rn activities in groundwater from D05 (494.5 ± 107.5 Bq/m$^3$) to D45 (787.4 ± 153.2 Bq/m$^3$) (p < 0.05). The $T_{res}$ of groundwater that recharged by river to the underlying aquifer and/or riverbank increased from D05 (−0.09 ± 0.09 days) to D45 (0.15 ± 0.13 days) (Table 2)." has been changed to "There was a significant increase of $^{222}$Rn activity in groundwater from D05 (610.1 ± 107.5 Bq/m$^3$) to D45 (787.4 ± 153.2 Bq/m$^3$) (p < 0.05) (Table 1). The $T_{res}$ of recharged groundwater from river water increased from D05 (0 days) to D45 (0.15 ± 0.13 days) (Table 1)." (see P. 17, Line 459-464)*

*Table 1: The $^{222}$Rn values in river water, background groundwater and riparian groundwater in three plots (D05, D20, and D45), and the average residence time of recharged groundwater from river water ($T_{res}$, day) in 2021. The background groundwater represents groundwater in aquifers more than 100 m away from the riverbank.*

| | River water | Background groundwater | Riparian groundwater | | |
|---|---|---|---|---|---|
| | | | D05 | D20 | D45 |
| $^{222}$Rn value (Bq/m$^3$) | 610.1 ± 212.3 | 7400.0 ± 35.4 | 610.1 ± 107.5 | 763.3 ± 118.3 | 787.4 ± 153.2 |
| $T_{res}$ (days) | 0 | Null | 0 | 0.13 ± 0.1 | 0.15 ± 0.13 |

*Notes: D05, D20, and D45 are the plots at distance of 5 m, 20 m, and 45 m away from the riverbank, respectively.*

21. Comment:

Lines 280-291: I would also remove the first sentence here and go straight to the results. I find the first paragraph difficult to follow because you present a mix of interannual and seasonal differences, think about what you want to present. You could have a first paragraph presenting the interannual differences, a second presenting the seasonal differences (differences between months for a SAME year). From my point of view, the Figure 9 only shows the differences between plots (you only show the stats for this), not between years (despite the same scales for the y axes) and months, so I would only refer to Figure 9 in the second paragraph.

**Response:** *We have removed the first sentence and went straight to the results. We have a first paragraph presenting the interannual differences (see P. 17, Line 466 to P18, Line 476), and a second presenting the seasonal differences (differences between months for a SAME year) (see P. 18, Lines 477-485). We have only referred to Figure 10 in the third paragraph (see P. 18, Line 490).*

*The first paragraph has been changed to "The proportional contributions of river water to riparian S. babylonica trees were significantly higher in 2021 (mean of 23.8% ± 7.8%) than in 2019 (mean of 16.8% ± 4.7%) (p < 0.05). Specifically, the most significantly monthly difference in the RWC to riparian S. babylonica trees between dry 2019 and wet 2021 was up to 19.8% (p < 0.001). The monthly maximum RWC to S. babylonica trees was significantly higher in wet 2021 (35.2% ± 7.0%) compared to dry 2019 (24.2% ± 3.0%) (p < 0.05)." (see P. 17, Line 466 to P18, Line 476)*

*The second paragraph has been added to present the seasonal differences in the RWC to riparian trees for a SAME year: "The riparian S. babylonica took up the most river water in July (35.2 ± 7.0%) in 2021, whereas the highest RWC to riparian trees occurred in June (24.2% ± 1.6%) in 2019. The minimum river water uptake for riparian S. babylonica in 2021 was in September (17.7% ± 2.7%), while trees took up the least river water in August 2019 (13.2% ± 1.9%). Although the precipitation amount in rainy season was much higher than in drought season (p < 0.001), no significant difference in the RWC to riparian S. babylonica trees was observed between the rainy and drought seasons in a same year (p > 0.05) (Figs. 2 and 9). The difference values of the RWC to riparian trees between the rainy and dry seasons were not significantly different (p > 0.05) in both 2019 (−4.0%) and 2021 (−4.4%) (Fig. 9). These showed that there were no significant seasonal variations in the RWC to riparian trees within a year (p > 0.05)." (see P. 18, Lines 477-485).*

22. Comment:

Lines 288-291: This section reads well but you say in the first sentence that there are significant differences in RWC between the 3 plots in 2021 while there is no difference (Fig 9), please correct.

**Response:** *We have corrected the first sentence as follows: "The water uptake of river water by riparian S. babylonica was significantly different between the three plots in 2019 (p < 0.05), while no difference was observed between the three plots in 2021 (p > 0.05) (Fig. 10)." (see P. 18, Lines 486-488)*

23. Comment:

Lines 299-304: I would add in the text the R2 and p values of the linear models even if they are shown in Figure 10. Why did you fit the model to the whole dataset? And not one model for each year? It should be explained in the data analysis section, maybe I missed this point.

**Response:** *We added the $R^2$ and p values of the linear models in the text as follows: "There was a significant negative relationship between the RWC to riparian trees and WTD ($R^2 = 0.57$; $p < 0.001$) (Fig. 11a). The leaf $\delta^{13}C$ of riparian S. babylonica was found to be negatively correlated with the RWCs to S. babylonica ($R^2 = 0.61$; $p < 0.001$) but positively related to WTD ($R^2 = 0.37$; $p < 0.001$) in linear functions (Fig. 11b and c)." (see P. 19, Lines 499-502)*

*We also added the reasons for fitting the model to the whole dataset in the section "2.5 Statistical analysis". "The linear regression model was fitted to the whole dataset in both years to get the general relationships between the WTD, leaf $\delta^{13}C$ values and the RWC to riparian trees." (see P. 14, Lines 374-376)*

**DISCUSSION**

24. Comment:

I think that this section is not well-enough developed and that the sections should be revised in order to reflect the objectives presented in the introduction. There is also no discussion about the MixSIAR model and the iteration method presented here and on which all the results are based. I would add a section to discuss about their strengths/weaknesses and implications for the discussed results. Then, I would discuss the RWC to riparian trees and the effect of the distance from the stream. Finally, I would discuss about the link between RWC/WUE/WTD and its implications (also include management). I think the discussion about the potential processes and implications should be developed, also how do your results compare with previous work and why?

**Response:** *This is a good comment. We have reorganized the entire discussion section. Firstly, we discussed the advantages and limitations of MixSIAR model and the iteration method (see P. 19, Line 508 to P.21, Line 556). Secondly, we discussed the RWC to riparian trees and effects of the distance from the river and dry/wet year on the RWC to riparian trees (see P. 21, Line 560 to P.24, Line 641). In this part, we discussed the potential processes. Finally, we discussed the link between*

*RWC/WUE/WTD and its implications on management of riparian forest and river runoff (see P. 24, Line 642 to P.27, Line 733). We have compared the results with previous work and provided corresponding explanations throughout the revised discussion section.*

25. Comment:

Lines 336-342: You just look to mostly report previous findings here. First, what YOUR results suggest? Then, HOW does it relates/compare to previous work?

**Response:** *We have revised these sentences to emphasize the implications of our results and compare our results with previous work. The revised part can be found in the first paragraph of the section 4.2 in the revised manuscript (see P. 21, Line 563 to P. 22, Line 586).*

26. Comment:

Lines 344-345: This point is super interesting; can you try to make the story about this clearer?

**Response:** *We have clarified this point in detail. The revised story has been specified in the first paragraph of the section 4.2 (see P. 21, Line 563 to P. 22, Line 586).*

27. Comment:

Lines 381-382: I don't think you can really compare your "optimal" WTD with values from other studies because it is not the same site. I would rather discuss the potential reasons of these differences. Clarify the "knee point", I see what you mean but I would reword, a "break point" instead?

**Response:** *We have deleted the comparison of the "optimal" WTD with values from other studies. We discussed the potential reasons of these differences. The reorganized discussion part has been specified in the second paragraph of the section 4.3 (see P. 26, Lines 696-710)*

*In addition, we changed the "knee point" to "break point" (see P. 26, Line 708).*

28. Comment:

Line 390: You talk about accurate separation and quantification of RWC to riparian trees but we don't know the limitations and uncertainty of the model and iteration method.

**Response:** *We have added the discussion about the advantages, limitations and uncertainties of the*

*model and iteration method in the section "4.1 Advantages and limitations of MixSIAR model and the iteration method" (see P. 19, Line 508 to P.21, Line 556).*

CONCLUSION

29. Comment:

1. Lines 401-408: I think this is too much results, the conclusion should not be like an abstract. I would focus more on the implications of your findings for riparian zones management and future research.

**Response:** *We have modified the conclusion and focused more on the implications of our findings for riparian zone management and future research. The revised conclusion part is as follows:*

*"In this study, we presented a new iteration method together with the MixSIAR model and stable water isotopes ($\delta^2 H$ and $\delta^{18} O$) to separate and quantify the proportional contributions of river water to riparian S. babylonica in dry 2019 and wet 2021 along a losing river in Beijing, China. It was found that the infiltrating river water exchanged with riparian mobile water quickly but not completely mixing with waters held tightly in the fine pores. Riparian trees near a losing river generally extended roots into fine pores to access the immobile water sources. The isotopic discrepancies between the fast-moving water flow and the immobile water taken up by the roots led to a small RWC (20.3%) to transpiration of riparian trees. The water deficit in the dry year probably induced stomatal closure and larger reduction in transpiration compared to the photosynthesis of riparian trees, thus leading to an evident increase of WUE than in the wet year. The leaf WUE showed a negative correlation with the RWC to riparian trees but was positively related to WTD in linear functions ($p < 0.001$). Riparian S. babylonica trees maintained the highest WUE and the lowest river water uptake proportion under deep groundwater condition (with the WTD of 4 m) in this study. These suggested that rising groundwater level triggered riparian trees to increase the river water uptake and show a consumptive river-water-use pattern, which should not be recommended for the water resource management of a losing river restored by ecological water. This study provides valuable insights into riparian afforestation related to water use and ecosystem health." (see P. 27, Line 735 to P. 28, Line 757).*

**Technical corrections:**

**INTRODUCTION**

30. Comment:

Line 34: English is not my mother tongue but should it be "replenishment" instead of "replenishing"?

**Response:** *We have changed " replenishing" to "replenishment" (see P. 2, Line 45).*

31. Comment:

Lines 38-40: This is a very interesting question but some parts of the sentence need to be edited to have a clearer sentence. By "where" I guess you mean the sources? At the first read I thought you were talking about where along the river, I would use "source" to be clearer. I would change "responses to the variations in the water table" to "response to water table variations". Also, be more precise about what you are talking about: is it the river water or groundwater level? When you say "revegetated riparian species" it means that the species are revegetated, which is false, I would change to "tree water requirement of revegetated riparian zones/areas" for exemple (it is the riparian zone/area that have been revegetated). Finally, I am not sure that the word "balance" is the best one to use here, I would improve the wording of the sentence.

**Response:** *We have modified the "where and how much water riparian tree take up" to "water sources and how much river water is taken up by riparian trees". In addition, we have changed "responses to variations in the water table" to "responses of tree water use characteristics to groundwater level variations". The "revegetated riparian species" has been changed to "revegetated riparian zones". We have deleted the word "balance" and improved the wording of this sentence as follows: "Therefore, determining what water sources and how much river water is taken up by riparian trees and the responses of tree water use characteristics to groundwater level variations can help implement management strategies for maintaining river runoff and tree water need of revegetated riparian zones." (see P. 2, Lines 48-52).*

32. Comment:

Line 47: Can you be more specific about the "different waters"? Different "water sources"?

**Response:** *We have specified the "different waters" as "different water sources and plant stem*

*water" (see P. 3, Lines 63-64)*

33. Comment:

Line 51: Do you mean "change in river water level"?

**Response:** *Yes. But we have deleted this sentence due to the reorganization of the Introduction section.*

34. Comment:

Lines 62-66: I would improve the wording of the sentences. From my point of view, it does not read that well while it is important to state clearly the knowledge gap/issue here. Also, "estimations" of what?

**Response:** *We have rewritten these sentences and clarified the knowledge gap as follows: "Growing evidence showed that riparian trees rarely took up river water directly at a certain distance away from the riverbank because their lateral roots could not reach the river (Mensforth et al., 1994; Thorburn and Walker, 1994); Nevertheless, riparian trees could indirectly utilize river water that recharges deep zone (e.g., deep soil water and groundwater) when their roots tap into the groundwater level (Mensforth et al., 1994; Wang et al., 2019a). If we take river water as a direct water source, the RWC to transpiration of riparian trees may be overestimated. How to separate and quantify the contributions of the indirect river water source to riparian trees near losing rivers is a great challenge." (see P. 3, Line 76 to P.4, Line 89)*

35. Comment:

Lines 67-81: Maybe I'm too picky but be more specific when you use WUE: is it WUE of plant (lines 69, 70)? Trees (lines 73, 79)? Similarly for RWC, I would specify "RWC to riparian trees" (line 67).

**Response:** *We have added the tree or leaf in front of "WUE" and specified the "RWC to riparian trees" in this paragraph (see P. 4, Lines 105-106) and revised them throughout the manuscript.*

36. Comment:

Line 75: Water table depth: of groundwater?

**Response:** *"Water table depth (WTD)" in this manuscript represents "the depth to the water table". The "water table depth" is an official word to define "the depth to the upper surface of the saturated zone".*

37. Comment:

Lines 82-88: Great, very clear objectives here.

**Response:** *Thanks for your positive comments.*

38. Comment:

Line 86: I would specify "tree WUE".

**Response:** *We have specified WUE as "tree WUE" (see P. 4, Lines 106).*

**M&M**

39. Comment:

Line 94: I think "dried up from X to X" would be more correct. Or "during X up to X"?

**Response:** *We have changed "dried up during 1999 to 2007" to "dried up from 1999 to 2007" (see P. 6, Line 150).*

40. Comment:

Lines 95-96: The end of the sentence is a bit unclear due to the wording. I would change to "more than 33 km$^2$ of riparian zone has been revegetated until 2020" or "from 2007 to 2020". Have the trees been planted?

**Response:** *Yes, trees have been planted. We have changed this sentence to "More than 33 km$^2$ of riparian zone has been revegetated until 2020" (see P. 6, Lines 154-155).*

41. Comment:

Line 100: I would change to "from April to November 2019 and 2021".

**Response:** *We have changed it to "from April to November in 2019 and 2021" (see P. 6, Line 155).*

42. Comment:

Line 101: "were collected", not "was".

**Response:** *We have changed "was collected" to "were collected" (see P. 6, Line 162).*

43. Comment:

Line 104: I think it is "water level gauge".

**Response:** *We have changed "water gauge" to "water level gauge" (see P. 7, Line 167).*

44. Comment:

Line 105: "from April to November".

**Response:** *We have changed "during April to November" to "from April to November" (see P. 6, Lines 159).*

45. Comment:

Line 106: "with a total precipitation of".

**Response:** *We have changed "with total precipitation of" to "with a total precipitation of" (see P. 14, Line 382).*

46. Comment:

Line 107: I would correct and say "which was 1.8 times higher than for the drier year 2019 (445.6 mm)".

**Response:** *We have changed "which was 1.8 times of that in dry year of 2019 (445.6 mm)" to "which was 1.8 times higher than for the drier year 2019 (445.6 mm)" (see P. 14, Lines 383-384).*

47. Comment:

Line 108: I would also correct here: "fluctuated between X and X m" and "mean WTD across the three plots".

**Response:** *We have corrected "fluctuated at 27.9−28.9 m in 2019 and 27.3−29.7 m in 2021" as "fluctuated between 27.9 and 28.9 m in 2019 and between 27.7 and 29.3 m in 2021". The "mean WTD in three plots" has been changed to "mean WTD across the three plots" (see P. 15, Lines 393-396).*

48. Comment:

Line 109: Use "higher" not "larger" to compare values. Also, change to "higher than in X".

**Response:** *We have used "higher" to compare values. The "larger than that in 2021" has been changed to "deeper in 2019 (2.7 ± 0.3 m) than in 2021 (1.7 ± 0.5 m)" (see P. 15, Lines 194-195).*

49. Comment:

Lines 109-110: Is it not the opposite? The WTD is lower (shallower GW) closer to the stream (see Figures 1 and 3).

**Response:** *We have changed this sentence to "The WTD increased with increasing distances from the riverbank in both 2019 and 2021 (Fig. 3)." (see P. 15, Lines 395-396).*

50. Comment:

Lines 112-113: Please write the months in full.

**Response:** *We have rewritten the months in full. The corrected sentence is as follows: "Twelve sampling campaigns on May 5, June 14, July 26, August 15, September 26, November 5 in 2019 and April 24, May 25, June 26, July 15, September 1, November 5 in 2021 were conducted to collect groundwater, river water, soil, stem, and leaf samples." (see P. 7, Lines 174-176).*

51. Comment:

Line 132: I would say "extract water from xylem and soil samples" instead.

**Response:** *We have changed "extract water in stem and soil samples" to "extract water from xylem and soil samples" (see P. 8, Line 205).*

52. Comment:

Line 133: "above 99%".

**Response:** *We have changed "more than 99%" to "above 99%" (see P. 8, Line 207).*

53. Comment:

Line 137: I would use "xylem" rather than "stem" in this section.

**Response:** *We have used "xylem" rather than "stem" in this section. (see P. 7, Lines 193, and P.8, Lines 205, 212).*

54. Comment:

Line 138: Correct to "for both the IRIS and IRMS systems".

**Response:** *We have changed "between the IRIS and IRMS systems" to "for both the IRIS and IRMS systems" (see P. 8, Lines 213-215).*

55. Comment:

Lines 141-151: It is a clear section, but I am not familiar with $^{222}$Rn and don't really understand the sentence on lines 145-146 "to ensure… less than 80 Bq/m³". Can you clarify?

**Response:** *We have added more information about "to ensure… less than 80 Bq/m³". "The $C_{Air}$ in the $^{222}$Rn monitor was recorded at a 10-minute interval. The air inside the measurement set-up had maintained a certain $^{222}$Rn concentration right before the water sample injection ($C_{System}$, Bq/m³). It is generally assumed that the existing $C_{System}$ can be ignored accordingly when $C_{System}$ is around or lower than 80 Bq/m³. In this study, more than four intervals were conducted to ensure that the $C_{System}$ was less than 80 Bq/m³. The measurement range of $C_{Air}$ was 2–2,000,000 Bq/m³ with a measurement precision of 3%." (see P. 8, Line 220 to P. 9, Line -226)*

56. Comment:

Lines 153-156: I feel that these sentences do not belong here but in the introduction, for me the section starts on line 156 at "in this study…".

**Response:** *We have deleted these sentences in the M&M part and moved them to the introduction part (see P. 3, Lines 77-83).*

57. Comment:

Line 157: I would correct with "isotopes were integrated/used within/in the MixSIAR model and an iteration method was proposed to identify.". What do you mean by the "original"?

**Response:** *We have corrected with "In this study, stable water isotopes ($\delta^2H$ and $\delta^{18}O$) were integrated within the MixSIAR model and an iteration method was proposed to identify the contributions of the indirect river water that recharged riparian deep water to riparian S. babylonica trees (Figs. 4 and 5)" (see P. 9, Line 245 to P.10, Line 252). The "original" means that the total river water contribution to riparian deep water during the entire period of river losing flow to riparian deep zone since 2007. We have changed "original" to "total" and added more explanation about the total river water contributions in section 2.4.3. Here is the revised part: "It was worth noting that riparian deep soil water (80-170 cm) and groundwater can be recharged by river water continuously when the groundwater levels lied below the riverbeds (i.e., losing rivers). Therefore, the proportional contribution of the old river water (before t-1) to riparian deep water should not be ignored. The total RWC to riparian deep water should be quantified explicitly during the entire period of the river losing flow to riparian deep zone since 2007." (see P. 13, Lines 333-357).*

58. Comment:

Lines 158-159: I am not sure "merge" is the correct word, please correct the wording and grammar of the sentence.

**Response:** *We have changed the "merged" to "recharged". This sentence has been corrected as follows: "In this study, stable water isotopes ($\delta^2H$ and $\delta^{18}O$) were integrated within the MixSIAR model and an iteration method was proposed to identify the contributions of the indirect river water that recharged riparian deep water to riparian S. babylonica trees (Figs. 4 and 5)." (see P. 9, Line 245 to P.10, Line 252).*

59. Comment:

Lines 159-163: I would check the wording of these sentences, I found it difficult to understand (maybe follow the section titles you used for 2.4.1., 2.4.2. and 2.4.3.? – they are clear). For example, what do you mean by "root water uptake patterns"? The sources? "Without considering river water as a direct water source"? Also, using "figured out" connotes a lack of accuracy, so I would use

"determined" instead, for example.

**Response:** *We have changed the "root water uptake patterns" to "direct water source (including soil water at three different layers and groundwater) contributions to riparian trees". We have deleted "without river water as a direct water source" and changed "figured out" to "determined". The sentences have been corrected as follows: "Firstly, the direct water source (including soil water in different layers and groundwater) contributions to riparian trees were determined via $\delta^2H$ and $\delta^{18}O$ in different waters and the MixSIAR model. Secondly, the proportional contributions of river water to riparian deep water (i.e., riparian groundwater and deep soil water in the 80−170 cm layer) were determined by the MixSIAR model and water isotopes. Finally, the proposed iteration method was used to quantify the proportions of the indirect river water source taken up by riparian trees (Figs. 4 and 5)." (see P. 10, Lines 252-258).*

60. Comment:

Lines 165-168: I found these sentences difficult to understand, I would check the wording. What do you mean by "which was mixed proportionally", "relatively stable" in terms of seasonal variations in SWC, water isotopes and WTD"? I am also not sure these sentences are needed here.

**Response:** *We have carefully checked the wording of theses sentences and deleted "which was mixed proportionally with precipitation, old soil water, or even river water and groundwater" as well as "relatively stable". However, these sentences were needed to define the direct water sources for riparian trees in the MixSIAR model, which was critical to identify the contributions of indirect river water source to riparian trees. We have rewritten these sentences as follows:*

*"Soil water at different depths was taken up by riparian S. babylonica directly. We measured soil water isotopes at 11 depths in the three plots. In order to reduce errors in the analytical procedure, four soil layers (0−30 cm, 30−80 cm, 80−170 cm, and 170−300 cm) were used to identify the main root water uptake depth of riparian trees according to seasonal variations in the SWC, water isotopes and WTD. The average soil water isotopes values at depths of 0−5 cm, 5−10 cm, 10−20 cm, and 20−30 cm were calculated for the 0−30 cm soil layer, because the water isotopes went through strong evaporation and SWC varied significantly seasonally. The soil water isotope values at depths of 40−60 cm and 60−80 cm were averaged for the 30-80 cm soil layer, and those values at 90−110 cm and 150−170 cm depths were averaged for the 80−170 cm soil layer since the*

*water isotopes and SWC were relatively stable. The average isotopic values of soil water at deep depths (190−210 cm, 250−270 cm, and 280−300 cm) were calculated for the 170−300 cm soil layer, which varied with the fluctuations of groundwater levels. Groundwater could be regarded as a direct water source for phreatophyte riparian trees (Dawson and Ehleringer, 1991; Busch et al., 1992). As the isotopic composition of soil water in the 170−300 cm layer was similar to that of groundwater, they were considered to be one water source (groundwater)." (see P. 10, Line 270 to P. 11, Line 283).*

61. Comment:

Line 170: I don't understand why you refer to Figures 2, 3 and S1 here, I don't see the link with why you separated the soil in 4 layers. Correct "in the 170-230 cm layer".

**Response:** *We have deleted the Figures 2, 3, and S1 here and changed "in 170-300 cm layer" to "in the 170-300 cm layer". (see P. 11, Lines 284-287)*

62. Comment:

Line 172: I would change "determined" to "used as direct water sources".

**Response:** *We have revised it as suggested. (see P. 11, Line 294)*

63. Comment:

Lines 173-174: I would change "stem" to "xylem" since you measured isotopes of xylem water.

**Response:** *We have changed "stem water" to "xylem water" when we referred to "measured isotopes of xylem water" throughout the manuscript.*

64. Comment:

Lines 177-179: I don't think these sentences are needed here since you develop this point in section 2.4.2.

**Response:** *We have deleted this sentence in this paragraph. (see P. 11, Line 304 to P.12, Line 307)*

65. Comment:

Line 181: I would specify "deep soil water (80-170 cm) and groundwater", also check grammar of

the sentence.

**Response:** *We have specified "deep soil water" as "deep soil water (80-170 cm)" (see P. 13, Line 333-334). We have corrected the wording and grammar mistakes of the sentence and changed it to "It was worth noting that riparian deep soil water (80−170 cm) and groundwater can be recharged by river water continuously when the groundwater levels lied below the riverbeds (i.e., losing rivers)." (see P. 13, Lines 333-334).*

66. Comment:

Lines 181-184: There are some wording and grammatical mistakes I think. Change "could be" by "can be"? "were applied" by "were used"? "in the 80-170 cm layer".

**Response:** *We have corrected "could be" as "can be" (see P.13, Line 333) and changed "were applied" to "were used" (see P. 12, Line 312). We have modified all the "in ... cm layer" to "in the ... cm layer" throughout the manuscript.*

67. Comment:

Lines 183-190: I like this part, it's clear, and the Figure S2 is great! I just wonder if it would be better to have the figure in the main text rather than in the Supplement.

**Response:** *We have moved the Figure S2 to the main text (See Fig. 4).*

[Figure]

*Figure 4: Schematic diagram for potential water sources ofriparian deep soil water in the 80−170 cm layer (a) and groundwater (b).*

68. Comment:

Lines 191-192: I am not sure about the wording here: "recharged from the river to the underlying aquifer and/or riverbank"… or maybe I misunderstood.

**Response:** *We have changed "the residence time of groundwater recharged from the river to the underlying aquifer and/or riverbank" to "the average residence time ($T_{res}$, day) of recharged groundwater from river water" (see P. 9, Lines 234-236).*

69. Comment:

Line 199: I would not use "figured out" in a manuscript, I would use "determined" instead. Please correct the "figured out" throughout the manuscript.

**Response:** *We have changed "figured out" to "determined" throughout the manuscript (see P. 13, Line 332).*

70. Comment:

Line 200: I am not sure about the use of "merge" here.

**Response:** *We have changed "merged" to "recharged" (see P. 14, Line 363).*

71. Comment:

Line 202: What do you mean by "be consistent"? Can you change "proportions" to "contribution"?

**Response:** *We have changed "be consistent" to "be identical to" in this sentence (see P. 13, Line 339). And we have modified "proportions" to "contributions" (see P. 13, Line 340).*

72. Comment:

Line 222: I am not sure "recharge" is the best word, maybe say "we estimated the proportions of old and current river water in the riparian deep water", but should it be "in the riparian trees" instead (the aim of the section 2.4.3.)?

**Response:** *Yes, we have changed this sentence to "Using this proposed iteration method, we estimated the total proportions of old and current river water in the riparian trees." (see P. 14, Lines 365-367).*

73. Comment:

Lines 224-229: This section is a bit difficult to read, the first sentence it too long, I would try to separate it. Also, check the grammar and wording. The "regression analysis method" is unclear.

**Response:** *We have carefully checked the grammar and wording of this section. The revised section is as follows: "One-way analysis of variance (ANOVA) incorporating with the Kolmogorov-Smirnov, Levene's and post-hoc Tukey's tests ($p < 0.05$) were used to investigate the statistic differences of different variables. The variables included the WTD, SWC, $\delta^2H$ and $\delta^{18}O$ in different water sources and xylems, $^{222}Rn$ concentration of river water and groundwater, contributions of different water sources to riparian deep water or trees, and leaf $\delta^{13}C$ values in the three plots in 2019 and 2021. The linear regression model was fitted to the whole dataset in both years to get the general relationships between the WTD, leaf $\delta^{13}C$ values and the RWC to riparian trees. The statistical analysis was performed in the Excel (v2016) and SPSS (24.0, Inc., Chicago, IL, USA)." (see P. 14, Lines 369-379).*

**RESULTS**

74. Comment:

This section is well-organized and the results are presented concisely and quite clearly. There are some grammar and wording mistakes, especially when you present the isotopes results. You can't say "d$^2$H in precipitation was more depleted", but say "precipitation was more depleted in d2H" or "d2H in precipitation was higher/lower than…". Check throughout the section 3.1 (lines 232, 239, 240, 242-243, 244-245…). Also, you use a lot of "than these/that" (line 233) or "with that in" (line 236) to compare results between years, I would correct and use "than in" or "compared to" instead. Check also the use of "the".

**Response:** *Thanks for your helpful comments and suggestions. We have corrected the grammar and wording mistakes throughout the manuscript. The presentation of the isotopes results has been corrected as suggested throughout the manuscript (especially in the section 3.2, see P. 15, Line 403 to P. 16, Line 434). The incorrect words (including "than these/that", "with that in", and et al) were*

*changed to "than in" or "compared to" in order to compare results between years. We also corrected the use of "the" throughout the manuscript.*

75. Comment:

Line 232: I would not use "it was evident", just present the results clearly and briefly.

**Response:** *We have deleted "it was evident". (see P. 15, Line 403)*

76. Comment:

Line 234: Use "higher" rather than "larger" to compare values.

**Response:** *We have changed "larger" to "higher" in this sentence (see P. 15, Line 406) and throughout the manuscript.*

77. Comment:

Line 236: I would rather say "SWC of each soil layer" than say "of all four layer", at the first read I thought you combined all the soil layers together but you analyzed the difference between the plots for each soil layer separately from what I understood.

**Response:** *We have changed "SWC of all four soil layers" to "SWC of each soil layer". (see P. 15, Line 399)*

78. Comment:

Lines 246-253: This part reads better, it's clear and concise.

**Response:** *Thanks for your positive comments.*

79. Comment:

Line 250: I would correct to "decreased with increasing distance from the riverbank".

**Response:** *We have corrected to "decreased with increasing distance from the riverbank" (see P. 16, Line 430).*

80. Comment:

Line 253: What do you mean by "evidently"? Is it significant? and "plummeted"?

**Response:** *We have changed "evidently" to "significantly (p< 0.05)" and added the significance after plummeted "plummeted significantly (p< 0.05)" (see P. 16, Lines 432-434).*

81. Comment:

Lines 255-278: This section reads well and is well organized, check the mistakes I referred to previously. I would remove the first sentence and would go straight to the results.

**Response:** *We have corrected the wording, grammar, and other mistakes throughout the whole section 3.3 (see P. 16, Line 435 to P. 17, Line 464). We have also removed the first sentence of this section and went straight to the results (see P. 16, Lines 465-467).*

82. Comment:

Line 257: I am not sure about the use of the word "in-situ" to refer to the water that is already in the deep soil or groundwater compartment… But I don't really what word you could use instead so I'm not very helpful on this point.

**Response:** *Thanks for your comments. We have explained the meaning of "in-situ" word at the first use in the text (see P. 12, Lines 315-316). Here is the revised part: "The potential water sources of riparian deep soil water in the 80−170 cm layer at t included the in-situ (i.e., water that is already in the deep soil layer or groundwater) soil water in this layer at t-1, soil water in the 0−80 cm layer at t-1, river water between t-1 and t, precipitation between t-1 and t, and groundwater between t-1 and t (Fig. 4a)." (see P. 12, Lines 314-318)*

83. Comment:

Line 261: "15.7%".

**Response:** *We have changed "15.7" to "15.7%" (see P. 17, Line 443).*

84. Comment:

Line 262: "deep soil water", "the lowest", "and in June".

**Response:** *We have corrected them as suggested (see P. 17, Line 444).*

85. Comment:

Line 265: I would correct: "significant interannual and seasonal differences in the water sources…".

**Response:** *Thanks a lot for your suggestion. But we have deleted this sentence and went straight to the results according to the reviewer #1 comment #81.*

86. Comment:

Lines 268-269: Please correct the wording of this sentence.

**Response:** *We have changed this sentence to "The average contribution of the river water to riparian groundwater was 28.1% ± 12.1% during the observation period." (see P. 17, Lines 457-459).*

87. Comment:

Lines 271, 272: You should refer to Figure 3 here to help the reader understand since you present some of the WTD results.

**Response:** *We have referred to Figure 3 in this sentence (see P. 17, Line 457).*

88. Comment:

Lines 275-276: As I mentioned before, I still don't understand this sentence, maybe a wording mistake or me… Check the grammar as well.

**Response:** *We have changed the negative residence time "−0.09 ± 0.09 days" to "0" in this sentence. We also corrected the grammar mistake and revised it as follows: "The $T_{res}$ of recharged groundwater from river water increased from D05 (0 days) to D45 (0.15 ± 0.13 days)." (see P. 17 Lines 461-463, and Table 1)*

89. Comment:

Line 282: Please put the unit "%" after each result, "mean of X% ± X%", check and correct throughout the manuscript.

**Response:** *We have put the unit "%" after each result and corrected throughout the manuscript.*

90. Comment:

Lines 282, 284, 285, 286: Correct the grammar: "higher" not "more" or "larger", "lowest" not "least", "highest" not "most" to compare values. You can say "X was higher/lower than X" or "X was the highest/lowest in…".

**Response:** *We corrected this grammar mistake throughout the manuscript.*

91. Comment:

Lines 288-291: Use "between" not "among" to make comparisons. The first part of second sentence is perfect but the end is unclear "whereas….. in 2021" ("corresponding value"?, "along the distances"?), please correct the wording and grammar, or you could just say that there was no significant differences in 2021.

**Response:** *We have changed "among" to "between" to make comparisons throughout the manuscript. The end part of second sentence has been changed to "whereas there was no significant difference in 2021 (p > 0.05) (Fig. 10)." (see P. 18, Lines 488-490).*

92. Comment:

Line 292: I would slightly change the title to: "relationships between leaf d$^{13}$C, RWC to riparians trees and WTD".

**Response:** *We have corrected this presentation as suggested (see P. 18, Line 491).*

93. Comment:

Lines 293-298: Check the wording and grammar mistakes ("the" missing, "higher" not "larger", "significant" not "significantly"…) as commented above, write the months in full, use "significant" instead of "remarkably" (line 294).

**Response:** *We have corrected this paragraph as suggested. The revised version is as follows: "The leaf $\delta^{13}$C of riparian S. babylonica trees was significantly higher in 2019 ($-27.7‰ \pm 1.0 ‰$) than in 2021 ($-29.7‰ \pm 0.7 ‰$) (p < 0.05) (Table 1). There was a significant increase of the leaf $\delta^{13}$C from D05 ($-28.8‰$) to D45 ($-27.0‰$) in 2019 (p < 0.05), while no significant difference in the leaf $\delta^{13}$C was observed between different distances in 2021 (p > 0.05). The lowest leaf $\delta^{13}$C value of riparian trees occurred on August 15 in 2019 and July 14 in 2021, before when intense rainfall occurred in both years." (see P. 18, Line 491 to P.19, Line 498).*

94. Comment:

Lines 299-304: This section reads better. Correct "RWC to riparian trees". I would move the last sentence to the discussion section.

**Response:** *We have corrected "RWC to riparian trees" (see P. 19, Line 501). And we have moved the last sentence to the discussion section (see P. 26, Lines 706-710).*

DISCUSSION

95. Comment:

The flow of your thoughts is difficult to follow, I would try to be clearer in my explanations. There are wording and grammar mistakes that need to be corrected, also check the tense you use.

**Response:** *We have rewritten the whole discussion section and corrected the wording and grammar mistakes as well as tense mistakes in the revised version (see P. 21, Line 563 to P.22, Line 586).*

96. Comment:

Line 307-326: The section is well-organized but you don't need to repeat the results in this section (lines 307-310, 316-317, 324-325).

**Response:** *We have deleted the results and discussed the potential processes and implications in this section (see P. 21, Line 563 to P.22, Line 586). We also compare our results with previous work and provided more explanations for the reasons.*

97. Comment:

Line 311: I find it difficult to understand what do you mean by "contradictions", please develop your thoughts and the processes involved.

**Response:** *We have deleted the vague "contradictions" and rewritten this part. We also developed and made a complete story including my thoughts, potential processes, and implications. The revised part has been specified in the first paragraph of the section 4.2 (see P. 21, Line 563 to P.22, Line 586).*

98. Comment:

Line 311: I would be more precise and not use "interactions", we don't know if it is river-GW flow or GW-river flow, in your study you only looked at river-GW flow, say it.

**Response:** *In our study, there was only the process of river recharging the groundwater system. We have changed the confusing word "interactions" to "river water recharging riparian deep water", "river recharging riparian deep strata", or "recharged groundwater from river water" throughout the manuscript.*

99. Comment:

Line 321: Same comment for "exchange", it is not clear enough.

**Response:** *We have also changed the confusing word "exchange" to "river water recharging riparian deep water", "river recharging riparian deep strata", or "recharged groundwater from river water" throughout the manuscript.*

100. Comment:

Line 323: I don't think that "weakened" is the right word to use here.

**Response:** *We have deleted "weakened" and rewritten these sentences. The revised part is as follows: "Although there was no significant difference in the deep water (below the 80 cm layer) contributions to riparian trees between three plots, we observed the substantial effect of the declining groundwater level with increasing distance from the riverbank on the decreased indirect RWCs to riparian trees in dry 2019 (Fig. 10). Therefore, the interannual and spatial variabilities of the RWC to riparian S. babylonica trees were generally attributed to the various RWCs to riparian deep water rather than the water use patterns of riparian trees." (see P. 22, Line 604 to P.23, Line 608).*

101. Comment:

Line 326: "distance" from what?

**Response:** *We have specified "distance from the riverbank" throughout the manuscript.*

102. Comment:

Lines 329-335: The section here is also well-organized but I still find it difficult to follow your story.

**Response:** *We have rewritten this part to make a clearer story including my thoughts, potential explanations for small RWC to riparian trees, comparison with previous studies and corresponding explanations. The revision has been shown in the first two paragraphs of the section 4.2 (see P. 21, Line 560 to P. 22, Line 597).*

103. Comment:

Line 329: You can say "smaller than.." or say "small", please correct.

**Response:** *We have changed "smaller" to "small" (see P. 21, Line 563).*

104. Comment:

Lines 331-335: I like this part, just check the grammar and wording ("or that stored").

**Response:** *Thanks for your positive comment. We have combined this part with the potential explanations for small RWC to riparian trees in the first paragraph of section 4.2 to make the story clearer (see P. 21, Line 560 to P. 22, Line 597).*

105. Comment:

Line 346: Here and throughout the manuscript, don't forget: "RWC to riparian trees".

**Response:** *We have corrected "RWC to riparian trees" throughout the manuscript.*

106. Comment:

Line 347-349: These 2 sentences should be switched, I would first briefly remind the result and then say what it suggests, you did the opposite here.

**Response:** *We have switched these two sentences and rewritten these two sentences to make it clearer (see P. 22, Lines 590-594).*

107. Comment:

Line 349: I would reword "along the gradient of distance".

**Response:** *We have changed "along the gradient of distance" to "with increasing distance from the riverbank" (see P. 22, Line 606).*

108. Comment:

Line 351: The technical word "dimorphic" should be explained at first use to help the nonexpert reader to understand what you mean.

**Response:** *We have explained the meaning of "dimorphic" root systems at first use. The "dimorphic" root systems can help plant species to shift their main water sources between shallow and deep layers. Here is the revised part: "Our result is in contrast to previous study by Qian et al. (2017) who reported a significant increase of the RWC to G. biloba trees in response to the groundwater level decline. This discrepancy was ascribed to that riparian G. biloba had a dimorphic root system and shifted their main water sources from shallow soil layer to deeper soil layer. Nevertheless, the potential root growth rate of riparian phreatophyte S. babylonica trees can reach 1-13 mm/day, which allows the riparian S. babylonica trees to remain in contact with a rising/declining groundwater level and keep constant water uptake proportions from deep strata below the 80 cm depth (Naumburg et al., 2005)." (see P. 23, Lines 608-615)*

109. Comment:

Lines 351-355: Interesting difference between tree species, try to reword to clarify this point. The sentence on lines 354-355 is repeating the one on lines 351-353.

**Response:** *We have rewritten these sentences (see P. 22, Lines 598 to P23, Lines 614) to clarify the difference between tree species (see the response to Reviewer #1 Comment #108).*

110. Comment:

Line 357: I don't think "balance and coordination" are the right words here.

**Response:** *We have deleted this sentence and corrected to "Therefore, the relationships between the RWC to riparian trees, leaf-level physiological characteristics (e.g., leaf WUE) and hydro-meteorological conditions are critical to protect the revegetated riparian zones and maintain river runoff sustainability" (see P. 26, Line 718 to P. 27, Line 720).*

111. Comment:

Lines 361-369: I like the ideas here but I would improve the wording and check the grammar to improve the flow.

**Response:** *We have rewritten and improved the flow of this part (see the first paragraph of section 4.3; P. 24, Line 444 to P. 26, Line 695) to make a complete story including our thoughts, potential processes, and implications.*

112. Comment:

Line 362: "profligate" is not the right word here; it can't be used to describe water-use strategy.

**Response:** *We have modified "profligate water use strategy" to "consumptive river-water-use pattern" throughout the entire manuscript (see P26, Lines 707; P. 28, Line 753).*

113. Comment:

Line 366: Say "river water" not "river flow".

**Response:** *We have corrected it throughout the manuscript (see P. 25, Line 665).*

114. Comment:

Lines 371-374: This is results, not discussion.

**Response:** *We have deleted these results and rewritten this part (see P. 26, Lines 700-702).*

115. Comment:

Lines 375-380: I like the ideas here; grammar and wording need to be improved.

**Response:** *We have rewritten this part and improved the grammar and wording (see P. 26, Lines 696-710).*

116. Comment:

Line 375: Correct the grammar: "previous studies that showed an…".

**Response:** *We have corrected this sentence as "It was consistent with Horton and Clark (2001) who found an exponential increase of the leaf WUE of riparian Salix gooddingii with increasing WTD." (see P. 26, Lines 699-700).*

117. Comment:

Line 376: No need to add the equation.

**Response:** *We have deleted the equation as suggested (see P. 26, Lines 700-704).*

118. Comment:

Line 377: I think "coordinate" is not the right word here, maybe "optimize"?

**Response:** *We have changed delete "coordinate" and rewritten this part (see P. 26, Lines 706-710).*

119. Comment:

Line 379: I would reword "balancing the relationship" and "flow reservation".

**Response:** *We have deleted these words and rewritten these sentences. The revised part is as follows: "Therefore, the relationships between the RWC to riparian trees, leaf-level physiological characteristics (e.g., leaf WUE) and hydro-meteorological conditions are critical to protect the revegetated riparian zones and maintain river runoff sustainability." (see P. 26, Line 718 to P.27, Line 720).*

120. Comment:

Line 383: I would say "groundwater table" rather than "water table", it is clearer, please check throughout the manuscript

**Response:** *Thanks a lot for your comments. We have changed "water table" to "groundwater level" throughout the entire manuscript.*

121. Comment:

Lines 386-397: I find the flow of this section hard to follow, thoroughly check the wording and grammar, don't hesitate to ask an English native speaker.

**Response:** *We have double checked and corrected the writing in this section (see P. 25, Line 690 to P. 26, Line 699). The co-author Lixin Wang (Professor in Indiana University-Purdue University Indianapolis (IUPUI), USA) has further edited and polished the language of the revised manuscript.*

CONCLUSION

122. Comment:

Lines 399-412: Check and correct the wording and grammar.

**Response:** *We have corrected the wording and grammar in this part (see P. 27, Line 735 to P. 28, Line 757).*

123. Comment:

Lines 399-401: I like this part, reminding the objective of the study.

**Response:** *Thank you for the positive comments.*

FIGURES

124. Comment:

Figure 1: Great figure!! It is very clear, easy to read and show all the info needed. As I see dams along the river, I wonder if water was released during the study periods and how it could have affected river flow and the results?

**Response:** *Thanks for your positive comments. Due to continuous drought and groundwater overexploitation, the Chaobai River dried up from 1999 to 2007. The "ecological water" (including reclaimed water, reservoir water, and diverted water by the South-to-North Water Transfer Project) has been supplied via a systematic water release by dams to restore this dry river since 2007. A total of 51.1 million and 380 million cubic meters of ecological water sources were released to the Chaobai River in 2019 and 2021, respectively (see P. 6, Lines 152-154).*

*The dams along the Chaobai River are used to regulate the river water level especially in flood season, which has a great effect on the river runoff and thus the riparian groundwater level. Released water were significantly different between dry 2019 and wet 2021, which led to a remarkable discrepancy in river runoff and WTD between two years (see P. 6, Lines 152-154 and Fig. 3). Therefore, our study quantified the relationship between the RWC to riparian trees and WTD. We also found a significantly negative relationship between the RWC to riparian trees and the WTD ($R^2 = 0.57$; $p < 0.001$) (see P. 19, Lines 499-500 and Fig. 11a). This finding could help to control the river runoff and tree water requirement of revegetated riparian zones.*

125. Comment:

Figure 2: Check the reference to Figure 2 in the text, lines 106-107 should only refer to Figure 2a. I would change the caption to "Monthly average precipitation amount from 1961 to 2021 and monthly total precipitation amount for the observation years 2019 and 2021 (a), Daily total precipitation amount and precipitation isotopes during 2019 (b) and 2021 (c).

**Response:** *We have changed the reference to "Figure 2b and c" in the text to "Figure 2a" (see P. 14, Line 382). And we modified the caption to "Changes in monthly average precipitation amount from 1961 to 2021 and monthly total precipitation amount for the observation years 2019 and 2021 (a), Daily total precipitation amount and precipitation isotopes during 2019 (b) and 2021 (c)". (see Fig. 2)*

126. Comment:

Figure 3: The figure is clear but I would show groundwater level (on same scale as river water level) instead of water table depth so we can actually compare with river water level, but maybe I'm being too picky if the aim of the figure is only to show the seasonal variations.

**Response:** *This is a good comment. We have shown the groundwater level, water table depth, river water level in Fig. 3. We have also added the elevation of riverbed (26.0 m) as well as the riparian ground surface elevation (29.5 m) in the captions in order to indicate the groundwater level. The revised Fig. 3 is as follows (see Fig. 3):*

[Figure]

*Figure 3: Seasonal variations of the river water level and water table depth (WTD)/groundwater level (GWL) at distances of 5 m, 20 m, and 45 m away from the riverbank during the observation period in 2019 (a) and 2021 (b). The red arrow indicates the riparian ground surface level (29.5 m). The riverbed level is 26 m.*

127. Comment:

Figure 4: Nice flowchart, the reference is missing in the text.

**Response:** *Thanks for your positive comment. We have added the reference of the flowchart (Figure 5) in the revised version (see Fig. 4).*

128. Comment:

Figure 5: This is again a very nice figure. I would check and correct the wording of the caption (3 first sentences). Maybe try to increase the front size? I would use xylem instead of stem, as suggested before.

**Response:** *Thanks for your positive comment. We have checked and corrected the wording of the three first sentences in the caption. We also increased the front size and changed "stem water" to "xylem water" (see Fig. 6).*

[Figure]

*Figure 6: Dual-isotope ($\delta^2H$ and $\delta^{18}O$) biplots of different water bodies in the three plots (D05, D20, and D45) for the observation years 2019 and 2021. The local meteoric water line (LMWL) was fitted by precipitation isotopes for each year. The soil water line (SWL) was fitted by the soil water isotopes in the four layers across three plots (D05, D20, and D45) for each year. D05, D20, and D45 are the plots at distance of 5 m, 20 m, and 45 m away from the riverbank, respectively. The error bars indicate standard deviations.*

129. Comment:

Figure 6: Very nice figure, I would however correct the caption with "Seasonal variations in the (proportional) contributions of soil water and groundwater to riparian trees in the three plots.". Try to increase the front size.

**Response:** *Thanks for your positive comment. We have corrected the caption with "Seasonal variations in the proportional contributions of soil water and groundwater to riparian trees in the three plots (D05, D20, and D45) for the observation years 2019 (a−c) and 2021 (d−f).". And we also increased the front size. (see Fig. 7)*

[Figure]

*Figure 7: Seasonal variations in the proportional contributions of soil water and groundwater to riparian trees in the three plots (D05, D20, and D45) for the observation years 2019 (a−c) and 2021 (d−f). D05, D20, and D45 are the plots at distance of 5 m, 20 m, and 45 m away from the riverbank, respectively. The error bars indicate standard deviations.*

130. Comment:

Figure 7: Nice figure, correct the "in-situ" in the legend at the top of the figure and the y axis "contributions of water sources to riparian soil water in the 80-170 cm layer in the three plots", check the grammar in the caption. I would increase the front size as well.

**Response:** *We have corrected the "in-situ" in the legend at the top of the figure and the y axis "contributions of water sources to riparian soil water in the 80-170 cm layer in the three plots". We also modified the grammar in the caption and increased the front size (see Fig. 8).*

[Figure]

*Figure 8: Seasonal variations in the different water source contributions to riparian deep soil water in the 80−170 cm layer in the three plots (D05, D20, and D45) for the observation years 2019 (a−c) and 2021 (d−f). D05, D20, and D45 are the plots at distance of 5 m, 20 m, and 45 m away from the riverbank, respectively. The error bars indicate standard deviations.*

131. Comment:

Figure 8: Nice figure as well, just correct the "in-situ" in the legend, check the caption and try to increase the front size.

**Response:** *We have modified the "in-situ" in the legend as well as the grammar in the caption. We also increased the front size (see Fig. 9).*

[Figure]

*Figure 9: Seasonal variations in the different water source contributions to riparian groundwater in the three plots (D05, D20, and D45) for the observation years 2019 (a−c) and 2021 (d−f). D05, D20, and D45 are the plots at distance of 5 m, 20 m, and 45 m away from the riverbank, respectively. The error bars indicate standard deviations.*

132. Comment:

Figure 9: Nice and clear figure, correct the y axis "contributions of river water to riparian trees", I would try to better highlight the yearly average, maybe using bold? I did not notice it at first look. Check and correct the caption: "different letters show a significant difference in the RWC to riparian trees between two plots", maybe say "RWC to riparian trees in the three plots for each sampling campaign" rather than "seasonal variation" since you only show the statistical results of the differences between plots for a same campaign. The stats results for September 2021 look weird, D45 looks different than D05 and D20 not different from D05 and D45, can you check that?

**Response:** *We have corrected the y axis "contributions of river water to riparian trees" to "Proportions of river water for riparian trees" (see Fig. 10). The caption "The different letters represent significant differences in the river water contributions to riparian S. babylonica trees in*

*the three plots" has been corrected to "Different letters show a significant difference in the RWC to riparian trees between three plots for each sampling campaign". We highlighted the yearly average using bold fond (see Fig. 10). Additionally, we have modified the stats results of September 2021 as suggested (see Fig. 10).*

[Figure]

*Figure 10: River water contribution (RWC) to riparian trees in the three plots (D05, D20, and D45) for each sampling campaign for the observation years 2019 (a) and 2021 (b). Different letters show a significant difference in the RWC to riparian trees between three plots (p < 0.05). D05, D20, and D45 are the plots at distance of 5 m, 20 m, and 45 m away from the riverbank, respectively.*

133. Comment:

Figure 10: Very nice figure as well, maybe just try to increase the front size. Check the grammar in the caption.

**Response:** *Thanks for your positive comment. We have increased the front size of the caption and modified the grammar in the caption (see Fig. 11).*

[Figure]

*Figure 11: Relationships between the proportions of river water contributions for riparian trees and the water table depth (a), between the leaf δ¹³C values and the water table depth (b), and between the leaf δ¹³C values and proportions of river water contributions for riparian trees (c).*

**Response to Referee #2 (Dr. Remy Schoppach):**

**General comments:**

1. Comment:

This manuscript from Li et al. aims at quantifying the contribution of river water to the transpiration flux of trees growing in the riparian area. This is an important topic, of interest for the community. Globally, the paper quality suffers from a clear lack of structure, elusive objectives and a poor discussion. Figures are of relatively good quality. Clearly the authors are not native speakers, but the paper remains easily readable from a language perspective, except for the discussion. The effort put in the language is appreciated but the structure and the reasoning need to be substantially improved.

**Response:** *Thank you for insightful suggestions. We have substantially improved the structure of this manuscript especially for the Introduction and Discussion sections. A straightforward flow of ideas has been made to clarify a scientific gap that this paper aims to fill. We thoroughly discussed the potential processes, reasons, and implications of our findings in the study. We further compared the results with previous work and provided corresponding explanations throughout the Discussion section.*

*In the Introduction section, we have reorganized the structure and given a straightforward flow of ideas to address clearer objectives and scientific knowledge gaps of this study. Firstly, we have introduced the significance of determining what water sources and how much rive water is taken up by riparian trees nearby losing rivers (see P.2, Lines 40-52). Secondly, we have introduced a debate on whether the river water is a potential water source of riparian trees or not and how it becomes available to riparian plants. Thus, we have highlighted the question that how to separate and quantify the contributions of indirect river water source (which recharges riparian deep soil water/groundwater) to riparian trees nearby losing rivers (see P.2, Lines 53 to P. 4, Line 89). Thirdly, we have introduced several methods (e.g., MixSIAR model, water tracers ($\delta^2H$, $\delta^{18}O$, and $^{222}Rn$), iteration method) to identify the water source contributions to riparian plants. We also addressed the need to combine these methods to give a more reliable quantification of the river water contribution (RWC) to riparian trees (see P.4, Lines 90-104). Fourthly, we have introduced the effects of the RWC to riparian trees on the leaf water use efficiency (WUE) and the critical role*

*of the water table depth (WTD) in plant water use. Then we have raised the question that what is the quantitative relationship between the RWC, WUE and WTD? (see P.4, Lines 105 to P. 5, Line 114). Finally, we have provided three objectives of this study (see P.5, Lines 136 to P. 6, Line 142). In this section, we have highlighted the questions this study raised rather than displayed or compared different results from the literature. Some sentences which belong to the Introduction rather than the M&M have been moved into the Introduction section. (see P.3, Lines 77-84).*

*In the Discussion, we have reorganized and completely rewritten the entire section. Firstly, we have discussed the advantages and limitations of the MixSIAR model and the iteration method (see P.19, Lines 508 to P. 21, Line 556). In this part, we put more effort into explaining the feasibility of the methods used in this study, especially with consideration of their reflection and implications. Secondly, we discussed the potential processes of the RWC to riparian trees (see P.21, Lines 560 to P. 22, Line 597), and explaining the influence of the distance from the river and dry/wet year on the RWC to riparian trees (see P.22, Lines 598 to P. 24, Line 641). Finally, we discussed the link between RWC/WUE/WTD and developed its implications on management of riparian forest and river runoff (see P.24, Lines 642 to P. 27, Line 720). We have compared the results with previous work and provided corresponding explanations throughout the revised Discussion section. The entire experiment and in particular the results are placed into sufficient context with existing literatures to make the discussion more substantial. We have put quite more effort on the Discussion section by means of discussing the potential processes, reasons, and implications of our findings.*

*We have double checked and corrected the writing throughout this manuscript (especially discussion section). The co-author Lixin Wang (Associate Professor in Indiana University-Purdue University Indianapolis (IUPUI), USA) has helped us to further edit and polish the language of the revised manuscript.*

[Figure]

*Figure S1. Daily mean temperature (°C) and daily mean vapor pressure deficit (VPD, kPa) in 2019 (a) and 2021 (b). Daily reference evapotranspiration (ET$_0$, mm/day) and daily mean net radiation (W/m$^2$) in 2019 (c) and 2021 (d).*

[Figure]

*Figure S2: Seasonal variations of soil water content (SWC) in the 0−30 cm, 30−80 cm, 80−170 cm, and 170−300 cm layers on sampling campaigns (a−h) in 2019 (a−d) and 2021 (e−h). Different letters (a, b, and c) show a significant difference in the SWC between three plots (p < 0.05). D05,*

*D20, and D45 are the plots at distance of 5 m, 20 m, and 45 m away from the riverbank, respectively.*

2.   Comment:

Introduction lack of reasoning on the scientific gap. In some parts, it really reads like a discussion where the authors compare contrasting results from the literature without highlighting the questions it raises. Some concepts are not even introduced (e.g., the use of radon as an indicator or the need of an iteration method). The introduction requires a straight flow of ideas leading to a scientific gap that this paper aims to fill. The lack of structure is also visible as some parts of the introduction are displayed in the M&M.

**Response:** *Thanks very much for the insightful suggestions. We have reorganized the Introduction section and improved the flow of ideas leading to a scientific gap that this paper aims to fill (see P.2, Lines 40-52). In this section, we have highlighted the questions this study raised rather than displayed or compared different results from the literature (see P.2, Lines 53 to P. 4, Line 89). Some sentences which belong to the Introduction rather than the M&M have been moved into the Introduction section. (see P.3, Lines 77-84). We also introduced the concepts of using radon as an indicator and the need of an iteration method (see P.4, Lines 100-103).*

**Specific comments:**

**Abstract**

3.   Comment:

Line 16: write active. We propose instead of "were proposed"

**Response:** *We have changed this sentence to "We proposed an iteration method in combination with the MixSIAR model to quantify the proportional river water contribution (RWC) to transpiration of riparian S. babylonica and its correlations with the water table depth (WTD) and leaf $\delta^{13}C$." (see P. 1, Lines 19-22).*

4.   Comment:

Line 19: contributed by

**Response:** *We have corrected it as suggested (see P. 1, Line 24).*

5.    Comment:

Line 20: why using "but" instead of "and". Is the increase in river water acquisition in contradiction with the decrease in leaf $\delta^{13}C$? If yes, you need to explain why.

**Response:** *We have changed "but" to "and". The increase in river water acquisition is not in contradiction with the decrease in leaf $\delta^{13}C$. (see P. 1, Line 25).*

6.    Comment:

There is no explanation of the decrease in leaf $\delta^{13}C$ the abstract?

**Response:** *We have explained the decrease in leaf $\delta^{13}C$ in the Abstract. Here is the revised part: "Significantly increasing river water uptake (by 7.0%) and decreasing leaf $\delta^{13}C$ (by −2.0‰) of riparian trees were observed as the water table depth changed from 2.7 m in dry 2019 to 1.7 m in wet 2021 (p < 0.05). The higher water availability probably promoted stomatal opening and thus increasing transpiration water loss, which led to the decreasing leaf $\delta^{13}C$ in wet year compared to dry year." (see P. 1, Lines 25-28).*

7.    Comment:

Line 24. How the rising water table would "stimulate" trees to maximize transpiration? I think this is simply the results of a higher water availability leading to a less negative water potential in the root-zone and subsequent lower stomata regulation. There is no clue for any "stimulation" of the plant?

Moreover, the reasoning is a bit ambiguous as you say that a rising water table increases the transpiration and the water extraction from the river (but not from the groundwater?). This is puzzling to me.

**Response:** *We have deleted the words "stimulate riparian trees to maximize transpiration water consumptions". Riparian trees can take up groundwater directly, while the river water seeped into the groundwater is an indirect water source for riparian trees.*

*We have clarified and changed this sentence to "The rising groundwater level would trigger riparian trees to increase the water extraction from groundwater/river and show a consumptive river-water-use pattern, which could not be recommended in order to protect both rivers and riparian vegetation." (see P. 2, Line 32-35).*

8. Comment:

Line 27: why using capital letters for Groundwater-Soil-Atmosphere Continuum?

**Response:** *We have corrected as "groundwater-soil-atmosphere continuum". (see P. 2, Lines 36-37).*

**Introduction**

9. Comment:

Line 30: I'm not aware of a consequence of groundwater overexploitation on the alteration of precipitation regime. Maybe consider re-writing.

**Response:** *We have deleted the "precipitation regime" (see P. 2, Line 40).*

10. Comment:

Line 35: contributed "by"

**Response:** *We have changed "contributed 40%" to "contributed by 40%". (see P. 2, Line 45).*

11. Comment:

Line 38: delete "deeply"

**Response:** *We have deleted "deeply". (see P. 2, Line 48).*

12. Comment:

Line 39: their responses to the variation in the water table (response of what, transpiration? Variation in what, level?) Please be specific.

**Response:** *We have changed this sentence to "Therefore, determining what water sources and how much river water is taken up by riparian trees and the responses of tree water use characteristics to groundwater level variations can help implement management strategies for maintaining river runoff and tree water need of revegetated riparian zones." (see P. 2, Lines 48-52).*

13. Comment:

Line 39: Also, I don't understand how a "deep understanding" could help "balancing the river flow

and the revegetated riparian species"? It could help implementing management strategies maybe? But the understanding will not balance anything.

**Response:** *Thanks for your insightful suggestions. We have deleted "balance" in this sentence. We changed this sentence to "Therefore, determining what water sources and how much river water is taken up by riparian trees and the responses of tree water use characteristics to groundwater level variations can help implement management strategies for maintaining river runoff and tree water need of revegetated riparian zones." (see P. 2, Lines 48-52).*

14. Comment:

Line 39: Is "revegetated species" an already used term? I'm not native speaker but I don't think a species can be revegetated. A riparian area could be, but not a species, right?

**Response:** *We have changed "revegetated riparian species" to "revegetated riparian zones" (see P. 2, Line 52).*

15. Comment:

Line 41 to 43: contribution to what? Transpiration flux I guess, but please write it. Please also insert the references within the sentence after each corresponding argument.

**Response:** *We have changed "the river water contribution (RWC) to riparian trees" to "the river water contribution (RWC) to transpiration" (see P. 2, Line 59). We also inserted the references within the sentence after each corresponding argument. The revised part is as follows: "The graphical inference and direct comparison of stable isotopic values between plant stem water and different water sources (Dawson and Ehleringer, 1991; Busch et al., 1992; Costelloe et al., 2008; Zhao et al., 2016), statistical two- or multi-source linear mixing models (Alstad et al., 1999; Zhou et al., 2017), and the MixSIAR Bayesian mixing model (Wang et al., 2019a; Wang et al., 2020; White and Smith, 2020; Li et al., 2021) integrated with stable isotopes ($\delta^2H$ and $\delta^{18}O$) have been widely used to identify the potential water sources taken up by riparian trees." (see P. 4, Lines 90-104).*

16. Comment:

Line 45: "a separate water source". Separate from what, other sources? If yes, maybe list them.

**Response:** *We have changed "a separate water source" to "a direct water source" (see P.3, Line*

*58).*

17. Comment:

Line 53: There was a debate or there is?

**Response:** *We have changed "There was a debate" to "There is a debate" (see P.3, Line 60).*

18. Comment:

Line 63: inaccurate estimation of what? Please consider re-writing the entire sentence

**Response:** *We have rewritten these sentences and clarified the knowledge gap as follows: "Growing evidence showed that riparian trees rarely took up river water directly at a certain distance away from the riverbank because their lateral roots could not reach the river (Mensforth et al., 1994; Thorburn and Walker, 1994); Nevertheless, riparian trees could indirectly utilize river water that recharges deep zone (e.g., deep soil water and groundwater) when their roots tap into the groundwater level (Mensforth et al., 1994; Wang et al., 2019a). If we take river water as a direct water source, the RWC to transpiration of riparian trees may be overestimated. How to separate and quantify the contributions of the indirect river water source to riparian trees near losing rivers is a great challenge." (see P.3, Line 77 to P.4, Line 89).*

19. Comment:

Line 65: You can't just state that "how to separate and quantify the contribution of … is a great challenge" without reasoning it. Why is it a great challenge? If you don't explain it, the reader gets confused and can only believe you. You need first to introduce the reasons making this a challenge.

**Response:** *We have added the explanation of why is "how to separate and quantify the contribution of …" is a great challenge (see P.3, Line 77 to P.4, Line 89).*

*There is a debate on whether river water is a potential water source of riparian trees and how it becomes available to plants. Most of previous studies considered river water as a direct water source to evaluate the river water contribution (RWC) to transpiration of riparian trees (Alstad et al., 1999; Zhou et al., 2017; White and Smith, 2020). Nevertheless, some studies argued that river water was not a potential water source and rarely contributed to riparian trees (Dawson and Ehleringer, 1991; Bowling et al., 2017; Wang et al., 2019a). Growing evidence showed that riparian*

*trees rarely took up river water directly at a certain distance away from the riverbank because their lateral roots could not reach the river (Mensforth et al., 1994; Thorburn and Walker, 1994); Nevertheless, riparian trees could indirectly utilize river water that recharges deep zone (e.g., deep soil water and groundwater) when their roots tap into the groundwater level (Mensforth et al., 1994; Wang et al., 2019a). If we take river water as a direct water source, the RWC to transpiration of riparian trees may be overestimated. How to separate and quantify the contributions of the indirect river water source to riparian trees near losing rivers is a great challenge. (see P.3, Line 77 to P.4, Line 89).*

20. Comment:

Line 70 to 80 reads like a discussion more than an introduction.

**Response:** *We have deleted most of the sentences from line 83 to 88 and focused on the scientific gap as well as its corresponding reasons. This paragraph has been reorganized as follows:*

*"The RWC to riparian trees could substantially affect the leaf-level water use efficiency (WUE) and growth of riparian trees. Tree WUE is a key characteristic of plant water use, which can be defined as the ratio of photosynthetic rate to transpiration rate. Since leaf $\delta^{13}C$ values are positively related to tree WUE, leaf $\delta^{13}C$ has been widely used as an indicator of tree WUE for $C_3$ photosynthesis plants (Farquhar et al., 1989). For example, Thorburn and Walker (1994) found that the riparian Eucalyptus camaldulensis beside the ephemeral stream had higher tree WUE with more frequent access to river water based on the leaf $\delta^{13}C$ measurements. Moreover, the fluctuation of the water table depth (WTD) in the riparian zone resulting from changing river water levels plays a critical role in the RWC to riparian trees and tree WUE (Horton and Clark, 2001; Liu et al., 2017; Xia et al., 2018). However, little attention has been paid to quantifying the relationships between the RWC to riparian trees and tree WUE or WTD near a losing river." (see P.4, Line105 to P.5, Line 114).*

21. Comment:

Line 83: the first objective is the propose an iteration method. This comes from nowhere as no part of the introduction introduce the issue related to iteration methods? Also, what is an iteration method together with water stable isotopes?

**Response:** *This is a good comment. We have added the introduction of the iteration method in this part as follows: "The RWC to riparian trees can be estimated by quantifying both the direct water source contributions to riparian trees and the RWC to riparian deep water. A multi-iteration method (Marek et al., 1990; Zaid, 2010) is key to calculate the proportional contributions of total (old and current) river water to riparian deep water, which improves the estimation accuracy of the RWC to riparian trees." (see P.4, Lines 96 to 100).*

*We have changed "an iteration method together with water stable isotopes" to "an iteration method together with the MixSIAR model and water stable isotopes" (see P.5, Line 138).*

**M&M**

22. Comment:

Line 94: has been seriously degraded instead of degraded seriously.

**Response:** *We have corrected it as suggested (see P.6, Line 150).*

23. Comment:

Line 95: What is ecological water? How is that "ecological water" supplied? This part of context is worth being developed a bit more. Is it via a systematic water release by dams?

**Response:** *We have added the explanation of "ecological water" and how "ecological water" is supplied. The modified part is as follows: "Due to continuous drought and groundwater overexploitation, the Chaobai River dried up from 1999 to 2007 and the riparian ecosystem seriously degraded. The "ecological water" (including reclaimed water, reservoir water, and diverted water by the South-to-North Water Transfer Project) has been supplied via a systematic water release by dams to restore this dry river since 2007. A total of 51.1 million and 380 million cubic meters of ecological water sources were released to the Chaobai River in 2019 and 2021, respectively." (see P.6, Lines 149-154).*

24. Comment:

Line 103: via a pressure stage gauge

**Response:** *We have corrected it as suggested (see P.6, Line 165).*

25. Comment:

Line 108: Fluctuated between

**Response:** *We have corrected it as suggested (see P.15, Line 393).*

26. Comment:

Line 109: The mean WTD in the three plots was significantly (p<0.5) deeper in 2019 (value) than in 2021 (value).

**Response:** *We have corrected it as suggested (see P.15, Lines 394-395).*

27. Comment:

Line 105 to 110: Should this section displayed in the Results instead of M&M?

**Response:** *We have added a separate part "3.1 Hydro-meteorological conditions" in the Results section and Figure S1 to provide more information about the hydro-meteorological conditions. All the hydro-meteorological conditions in the M&M have been moved to section 3.1 (see P.14, Line 382 to P.15, Line 401).*

28. Comment:

Line 118: There is not mention of the $^{222}$Rn in the introduction. Therefore, the reader has no idea what $^{222}$Rn is and why you determine the its concentration?

**Response:** *This is a good comment. We have presented the significance of determining the $^{222}$Rn concentration in the Introduction section in this study. The revised part is as follows: "The radioactive isotope ($^{222}$Rn) has been widely used for tracing groundwater origins and corresponding pathways in the riparian zone (Close et al., 2014; Zhao et al., 2018). It is helpful to estimate the residence time of recharged groundwater from river water and its effects on the RWC to riparian trees." (see P.4, Lines 100 to 103).*

*References:*

*Close M., Matthews M., Burbery L., Abraham P. and Scott D.: Use of radon to characterise surface*

water recharge to groundwater. *Journal of Hydrology. 53(2): 113-127, 2014.*

Zhao D., Wang G., Liao F., Yang N., Jiang W., Guo L., Liu C. and Shi Z.: Groundwater-surface water interactions derived by hydrochemical and isotopic ($^{222}$Rn, deuterium, oxygen-18) tracers in the Nomhon area, Qaidam Basin, NW China. *Journal of Hydrology. 565, 650-661, 2018.*

29. Comment:

Line 120: Is it Three trees in each of the three plots or one three per plot? Please re-write to avoid confusion.

**Response:** *We have modified this sentence to "One riparian S. babylonica tree was chosen in each plot for $\delta^2H$ and $\delta^{18}O$ measurements in xylem water as well as $\delta^{13}C$ analysis in plant leaves. The mean diameter at breast height of three sampled trees was 28.6 ± 4.4 cm." (see P.7, Lines 186 to 188).*

30. Comment:

Line 133: Did you measure the efficiency of the extraction and how? By weighting your fresh and dry samples? Please explain it.

**Response:** *We have added more information about how to calculate the efficiency of the extraction: "We weighed all the xylem and soil samples before and after extraction. Subsequently, the efficiency of water extraction was calculated in order to ensure the water extraction efficiency above 99% to avoid isotopic fractionation during water extraction." (see P.8, Lines 205 to 208).*

31. Comment:

Line 141: Is there something missing in this sentence. What is exactly $C_{Air}$?

**Response:** *We have rewritten this paragraph to make it clear. "The radon ($^{222}$Rn) concentration in the groundwater and river water samples ($C_{Water}$, Bq/l) was determined based on the air $^{222}$Rn concentration values ($C_{Air}$, Bq/m$^3$) measured by a $^{222}$Rn monitor (Alpha GUARD PQ2000 PRO, Bertin Instruments, Germany). 100 ml of the water sample was slowly poured into the air-tight glass bottles and then purged with air in a closed gas cycling system. The $C_{Air}$ in the $^{222}$Rn monitor was recorded at a 10-minute interval. The air inside the measurement set-up had maintained a certain*

$^{222}$Rn concentration right before the water sample injection ($C_{System}$, Bq/m$^3$). It is generally assumed that the existing $C_{System}$ can be ignored accordingly when $C_{System}$ is around or lower than 80 Bq/m$^3$. In this study, more than four intervals were conducted to ensure that the $C_{System}$ was less than 80 Bq/m$^3$. The measurement range of $C_{Air}$ was 2–2,000,000 Bq/m$^3$ with a measurement precision of 3%." (see P.8, Lines 220 to P.9, Lines 229).

32. Comment:

Line 153 to 156: This is introduction, not M&M.

**Response:** *These sentences have been moved into the Introduction part (see P.3, Lines 77-84).*

33. Comment:

Line 160: Based on what did you decided to not consider river water as a direct source for tree water uptake? You have to justify this choice one way or another. Or at least explain it.

**Response:** *We have explained why we did not consider river water as a direct water source for tree water uptake in the section 2.4.1 (see P.11, Lines 288-294).*

*"Mensforth et al. (1994) and Thorburn and Walker (1994) characterized the projected edge of canopy as the extension range of lateral roots, which could indicate whether riparian trees take up river water directly or not. In this study, the projected edge of canopy was less than 5 m for riparian S. babylonica tree closest to the river (5 m away from the riverbank). It indicated that the lateral roots of S. babylonica trees could not tap into the river. Therefore, river water was not considered as a direct potential water source for tree water uptake, while groundwater and soil water in the 0−30, 30−80, and 80−170 cm layers were considered as direct water sources for riparian S. babylonica." (see P.11, Lines 288-294).*

34. Comment:

Line 173: What is the correction proposed by Li et al. (2021). Please explain briefly how this works.

**Response:** *We have explained on how the correction method –PWL correction method works in the section 2.4.1. Here is the revised part: "In this study, the $\delta^2H$ offsets between the xylem water in riparian trees and its corresponding potential source waters were observed, which could result from $\delta^2H$ fractionation in the plant water use processes (Li et al., 2021). These $\delta^2H$ offsets could lead to*

*large errors in estimating the water source contributions using the MixSAIR model. In order to eliminate the $\delta^2H$ offsets of xylem water from its potential water sources, the measured xylem water $\delta^2H$ values were corrected via the potential water source line (PWL) proposed by Li et al. (2021). The PW-excess (PW-excess = $\delta^2H - a_p\delta^{18}O - b_p$; $a_p$ and $b_p$ were slope and intercept of the PWL, respectively) was calculated to indicate the $\delta^2H$ deviation from the PWL, which was subsequently subtracted from the measured xylem water $\delta^2H$ values. The corrected $\delta^2H$ and raw $\delta^{18}O$ in xylem water were set as the mixture data in the MixSIAR model to quantify the contributions of direct water sources to riparian S. babylonica." (see P.11, Line 295 to P.12, Line 307).*

35. Comment:

Line 183: Try to avoid saying "As shown in figS2a". Just write your sentence and refer to the figure at the end.

**Response:** *We have corrected this mistake throughout the manuscript.*

36. Comment:

Line 194: Please add a reference for Eq. 2 and for the coefficient values.

**Response:** *We have added a reference (Hoehn and Von Gunten, 1989) for the equation and the coefficient (see P9, Lines 231-234).*

*Reference:*

*Hoehn, E. and Von Gunten, H. R.: Radon in groundwater: A tool to assess infiltration from surface waters to aquifers. Water Resources Research. 25(8), 1795-1803, 1989.*

37. Comment:

Line 195: this is the first time you indicate that $^{222}$Rn is Radon. Should come earlier in my opinion.

**Response:** *We have indicated that $^{222}$Rn is Radon in the Introduction section (see P.4, Lines 100-103).*

**Results**

38. Comment:

Line 262: was the lowest

Response: *We have corrected it as suggested (see P17, Line 444).*

39. Comment:

Line 276: that is recharged

Response: *We have changed it to "recharged groundwater from river water" (see P17, Lines 461-463).*

40. Comment:

Line 282: were significantly higher

Response: *We have corrected it as suggested (see P17, Line 468).*

41. Comment:

Line 285-286-287: Please consider re-writing

Response: *We have written these sentences as follows:*

*"The proportional contributions of river water to riparian S. babylonica trees were significantly higher in 2021 (mean of 23.8% ± 7.8%) than in 2019 (mean of 16.8% ± 4.7%) (p < 0.05). Specifically, the most significantly monthly difference in the RWC to riparian S. babylonica trees between dry 2019 and wet 2021 was up to 19.8% (p < 0.001). The monthly maximum RWC to S. babylonica trees was significantly higher in wet 2021 (35.2% ± 7.0%) compared to dry 2019 (24.2% ± 3.0%) (p < 0.05).*

*The riparian S. babylonica took up the most river water in July (35.2 ± 7.0%) in 2021, whereas the highest RWC to riparian trees occurred in June (24.2% ± 1.6%) in 2019. The minimum river water uptake for riparian S. babylonica in 2021 was in September (17.7% ± 2.7%), while trees took up the least river water in August 2019 (13.2% ± 1.9%). Although the precipitation amount in rainy season was much higher than in drought season (p < 0.001), no significant difference in the RWC to riparian S. babylonica trees was observed between the rainy and drought seasons in a same year (p > 0.05) (Figs. 2 and 9). The difference values of the RWC to riparian trees between the rainy and dry seasons were not significantly different (p > 0.05) in both 2019 (−4.0%) and 2021 (−4.4%) (Fig.*

*9). These showed that there were no significant seasonal variations in the RWC to riparian trees within a year (p > 0.05)." (see P. 17, Line 466 to P. 18, Line 485)*

42. Comment:

Line 293: reference the corresponding figure

**Response:** *We have corrected as suggested (see P.18, Lines 486-490).*

43. Comment:

Line 294: -27.7 is not remarkably larger than -29.7

**Response:** *The leaf $\delta^{13}C$ value of C3 plants generally ranged from -25.0‰ to -31.0‰. The statistical analysis showed that the leaf $\delta^{13}C$ in 2019 ($-27.7 \pm 1.0$ ‰) was significantly higher than in 2021 ($-29.7 \pm 0.7$ ‰) (p < 0.05). (see P.18, Lines 492-494).*

44. Comment:

Line 295: a significant increase

**Response:** *We have corrected it as suggested (see P18, Lines 494-495).*

45. Comment:

Line 297: before when (chose one)

**Response:** *We have deleted "when" (see P18, Line 497).*

46. Comment:

Line 303: This statement is dropped without any explanation. I suggest to move it to the discussion and to actually discuss it.

**Response:** *We have moved the last sentence to the Discussion section. Here is the revised sentence: "These indicated that optimal WTD for plant species was related to the highest leaf WUE, under that condition plant species could consume less water for transpiration to maximize $CO_2$ assimilation (Antunes et al., 2018; Xia et al., 2018). The break point of WTD was not observed in this study (Fig. 11a and b). Further investigations need to be conducted under deeper groundwater levels (WTD > 4 m) to optimize the WTD and riparian plant-water relations." (see P26, Lines 706-*

*710).*

Discussion

47. Comment:

Line 311: what are the interactions? Do you mean exchange of water?

**Response:** *In this study, there was only one process that the river water recharges the groundwater system along a losing river. We have deleted the confusing words "interactions" and "exchange" and modified as "river water recharging riparian deep water", "river recharging riparian deep strata", or "recharged groundwater from river water" throughout the manuscript.*

48. Comment:

Line 311: "These contradictions". What contradictions?

**Response:** *We have deleted the "contradictions" and rewritten this part to make the story clearer. We have further discussed the potential explanations for the small RWC to riparian trees in the 4.2 section (see P.21, Line 560 to P.24, Line 641).*

49. Comment:

Line 312: might be due to that the. Please re-write

**Response:** *We have rewritten this sentence to make it clearer. Here is the revised sentence: "This probably indicated that river water recharged mobile groundwater quickly but could not completely replace water held tightly in the soil pores (Brooks et al., 2010; Evaristo et al., 2015; Allen et al., 2019)." (see P.21, Lines 573-575).*

50. Comment:

Line 314: previous studies "showing". There is a word missing here

**Response:** *We have rewritten this sentence. The revised sentence is as follows: "It was consistent with Sprenger et al. (2019) who found that the lateral seepage of river water or rising groundwater level could briefly saturate riparian soils but not entirely replace/flush immobile waters or isotopically homogenize different water pools." (see P.21, Lines 575-577).*

51. Comment:

Line 314: You mention previous studies, but cite only one.

**Response:** *We have deleted "previous studies" and changed this sentence to: "It was consistent with Sprenger et al. (2019) who found that the lateral seepage of river water or rising groundwater level could briefly saturate riparian soils but not entirely replace/flush immobile waters or isotopically homogenize different water pools." (see P.21, Lines 575-577).*

52. Comment:

Line 319: The rising water table stimulated exchanges. Stimulated isn't an appropriate word here. Maybe triggered?

**Response:** *We have changed "stimulated" to "triggered" (see P.26, Line 716).*

53. Comment:

Line 319: I don't understand this part of the discussion. It sounds obvious to me that if the water table reach 1.7 m below the surface it will exchange water with the soil layer standing at that same depth and in a larger proportion compared to a situation where the groundwater level is 1m deeper.

**Response:** *We have deleted this part of the Discussion, and discussed the potential processes that river water recharges riparian deep soil water and groundwater (see P.22, Line 589 to P.23, Line 635).*

54. Comment:

Line 326: Again, you mentioned previous studies and cite only one.

**Response:** *We have deleted this sentence to make the story clearer.*

55. Comment:

Line 329: a smaller proportion than what?

**Response:** *We have changed "smaller" to "small" (see P.21, Line 563).*

56. Comment:

Line 331: "is similar" and "Nearby perennial streams"

**Response:** *We have deleted this sentence to make the story clearer.*

57. Comment:

Line 334: what is "that" in the sentence?

**Response:** *We have deleted this sentence and rewritten this section to make the story clearer.*

58. Comment:

Line 340: The authors actually did not measure the isotopic signature of the bound water in fine pore and neither the exchange between this water and the river water, so you can't state that it rarely exchanges. What you can do is to speculate and use this argument as an explanation.

**Response:** *This is a good comment. We have deleted the misleading statement (i.e., "it rarely exchanges") in the revised manuscript and just use the ecohydrology separation as an explanation. We also provided more evidence on speculating the ecohydrology separation between the plant accessible bound water pools and the fast-moving water pools in the revised sections 3.4 and 4.2 (see P.18, Lines 480-485; P22, Lines 581-586).*

59. Comment:

Line 348: significantly more river water. Actually no, you don't know. A higher proportion maybe, but I would bet that it is the opposite for the amounts.

**Response:** *We have deleted "significantly more river water". We quantified the proportional (rather than amounts) contributions of river water to riparian trees using the newly proposed iteration method combined with the MixSIAR model and water isotopes, and a higher proportion of the RWC to riparian trees was quantified in wet 2021. These methods are reliable to determine the proportions of RWC to riparian trees (see P.19, Line 510 to P.20, 525). In addition, we have added the hydro-meteorological data (including evaporative demand indicated by VPD and net radiation) to analyze changing trends of leaf-level characteristics (e.g., transpiration rate and photosynthetic rate) (see P.14, Line 381 to P.15, line 401). This analysis also indicated a higher proportion of the RWC to riparian trees in wet 2021 than in dry 2019. The detailed discussions have been added in the section 4.2 (see P.24, Line 644 to P. 26, Line 695).*

60. Comment:

Line 356: The was a balance and coordination…. because obvious differences were found. What you did not find these differences? This entire sentence isn't clear and I don't see the point of it.

**Response:** *We have deleted this sentence and rewritten the whole section to make the story clearer.*

61. Comment:

Line 359: If you say the tree grew more reliance on… you need to compare it with something. Otherwise, you should not use "more". More than what? More than who, another species?

**Response:** *We have deleted "more" or other comparison when we don't compare it with other thing throughout the manuscript.*

62. Comment:

Line 360 to 370: I don't understand how the authors can actually infer a profligate water use strategy whereas they have no ideas on the water fluxes. They have no transpiration measurements, no evaporative demand measurements. What about the radiation amount? This is a key variable driving water-use efficiency. If the radiation over 2021 growing season was twice less than the radiation in 2019, it is easily imaginable that the photosynthetic rate was substantially impacted and subsequently the WUE.

**Response:** *This is a good comment. We have added more information about the meteorological data (e.g., temperature, VPD, $ET_0$, and net radiation) in the Results section 3.1 and Figure S1 (see P.14, Line 381 to P.15, Line 401; Fig S1). We also extensively discussed the impacts of these water fluxes on the river-water-use pattern for riparian trees in section 4.3 (see P.24, Line 644 to P. 26, Line 695).*

*The average daily VPD during the observation period was significantly higher in dry 2019 (1.1 KPa) than in wet 2021 (0.9 KPa) (p < 0.05) (see Fig. S1a and b). Nevertheless, no significant difference in the average daily mean net radiation during the observation period was found between dry 2019 and wet 2021 (p > 0.05) (see Fig. S1 c and d) (see P.15, Lines 390-392). Dry 2019 was characterized as higher water demand (indicated by higher VPD) and lower water availability compared to wet 2021, but the energy resource (indicated by net radiation) for riparian trees was similar between two years (Figs. S1 and S2). We argued that water limitation rather than energy*

*limitation regulated the leaf-level stomatal conductance of riparian S. babylonica trees. The high water demands but low river water availability in dry year probably resulted in stomatal closure of riparian trees to minimize water loss, which could eventually lead to a decrease of transpiration rate and even photosynthetic rate (Fabiani et al., 2021; Behzad et al., 2022). Aguilos et al. (2018) further found that water stress would enhance radiation-normalized WUE because the lack of water availability induced stronger reduction in transpiration than photosynthesis. With no difference in the average net radiation between dry and wet years, the lower river water availability in the dry year probably resulted in an increase of leaf WUE. It can be inferred that riparian S. babylonica trees took up more river water and probably showed a consumptive river-water-use pattern in the wet year compared to the dry year. This agreed well with previous studies that the woody plants showed lower leaf WUE and consumptive water-use patterns in rainy season, while they showed higher leaf WUE and conservative water-use patterns with lower soil water availability in dry season (Horton and Clark, 2001; Cao et al., 2020; Behzad et al., 2022). However, consumptive river water taken up by riparian trees could result in a great loss of river water, which should be avoided in the riparian zone of a losing river restored by "ecological water". (see P.24, Line 644 to P.26, Line 695).*

*Overall, we quantified the relative contribution of river water to the transpiration of riparian trees using the newly proposed iteration method together with the MixSIAR model and stable water isotopes. We could infer that riparian S. babylonica trees took up more proportions of river water and probably showed a consumptive river-water-use pattern in wet year compared with dry year based on the plant water uptake patterns, leaf WUE characteristics, and hydro-meteorological data (e.g., evaporative demand, net radiation, soil water content, and water table depth). The measurements of transpiration flux could help to validate the water use strategy of riparian trees and we will investigate it in future studies.*

63. Comment:

Line 378: In my opinion the argument developed here is misleading. The water table is low, likely because of the high evapotranspiration, which indicates a high transpiration rate, and very likely an even higher photosynthetic rate, leading to a high WUE. Under the same radiation conditions but with a much higher WTD, the WUE would be as high and probably even higher, and not the opposite

as suggested by the authors.

**Response:** *This is a good comment. The reviewer suggested that the high transpiration rate and thus the high photosynthetic rate of riparian trees in dry 2019 could also lead to the groundwater level decline and increasing leaf WUE. However, this viewpoint may not work in this study. There are three non-exclusive reasons:*

*Firstly, in this study, the dams along the Chaobai River are used to regulate the river water level especially in flood season, which has a great effect on the river runoff and thus the riparian groundwater level. A total of 51.1 million and 380 million cubic meters of ecological water sources were released to the Chaobai River in 2019 and 2021, respectively. Therefore, significantly lower groundwater level in dry 2019 compared to wet 2021 was mainly attributed to the significantly lower amount of ecological water release in 2019 than in 2021 (Fig. 3) (see P.6, Lines 152-155).*

*Secondly, the hydro-meteorological conditions (including evaporative demand and net radiation) suggested that dry 2019 was characterized as higher water demand (indicated by higher VPD) and lower water availability compared to wet 2021, but the energy resource (indicated by net radiation) for riparian trees was similar between two years (Figs. S1 and S2). We argued that water limitation rather than energy limitation regulated the leaf-level stomatal conductance of riparian S. babylonica trees. Therefore, the lower water availability (resulting from lower rainfall and deeper groundwater levels) in dry 2019 could not induce riparian trees to increase the transpiration rate. In contrast, the high water demands but low river water availability in dry year probably resulted in stomatal closure of riparian trees to minimize water loss, which could eventually lead to a decrease of transpiration rate and even photosynthetic rate (Fabiani et al., 2021; Behzad et al., 2022). (see P.24, Lines 648-657).*

*Thirdly, Aguilos et al. (2018) further found that water stress would enhance radiation-normalized WUE because the lack of water availability induced stronger reduction in transpiration than photosynthesis. With no difference in the average net radiation between dry and wet years, the lower river water availability in the dry year probably resulted in an increase of leaf WUE. (see P.24, Line 657 to P. 25, Line 668).*

64. Comment:

Line 411: That is wrong, the authors did not measure the transpiration so they don't know if a WTD

of 4m minimize the plant transpiration.

**Response:** *Thanks for your insightful suggestions. We have deleted "the WTD of 4 m minimize the plant transpiration." In this study, we deeply discussed the relationship between the WTD and the RWC to riparian trees and its corresponding potential explanations. The revised discussion is as follows:*

*"The WTD played a critical role in the river water uptake of riparian trees near a losing river (Mensforth et al., 1994; Horton and Clark, 2001; Qian et al., 2017; Zhou et al., 2017). We observed that the proportional contributions of the river water source to riparian trees decreased linearly in response to groundwater level decline, leading to a proportional increase in leaf WUE (Fig. 11a and b). It was consistent with Horton and Clark (2001) who found an exponential increase of the leaf WUE of riparian Salix gooddingii with increasing WTD. As mentioned above, we emphasized the key role of reduced water availability on decreasing transpiration rate thus enhancing leaf WUE in this study. Nevertheless, there were some controversial views that leaf WUE of plant species increased firstly and then decreased with increasing WTD (Antunes et al., 2018; Xia et al., 2018). This could be due to the fact that riparian trees could tolerate reduced water availability only within a species-specific threshold, beyond which xylem cavitation and even crown mortality occurs (Naumburg et al., 2005). These indicated that optimal WTD for plant species was related to the highest leaf WUE, under that condition plant species could consume less water for transpiration to maximize $CO_2$ assimilation (Antunes et al., 2018; Xia et al., 2018). The break point of WTD was not observed in this study (Fig. 11a and b). Further investigations need to be conducted under deeper groundwater levels (WTD > 4 m) to optimize the WTD and riparian plant-water relations. (see P.26, Lines 696-710).*

*Our results have important implications for untangling the trade-offs between riparian tree water use and river runoff management. The proportion of the RWC to riparian trees has been compared between dry and wet years to investigate the effects of river water availability on the water use characteristics of riparian trees. The riparian S. babylonica trees showed the highest WUE and the lowest river water uptake proportion under the lowest groundwater level condition (with the WTD of 4 m). The rising groundwater level would trigger riparian trees to show a consumptive river-water-use pattern, which should not be recommended in the revegetated riparian zone beside an ecological-water-recharged losing river. Therefore, the relationships between the*

*RWC to riparian trees, leaf-level physiological characteristics (e.g., leaf WUE) and hydro-meteorological conditions are critical to protect the revegetated riparian zones and maintain river runoff sustainability." (see P.26, Line 711 to P.27, Lines 720).*

**Figure**

65. Comment:

Fig 1.: I like the figure; it describes the sampling site very well. Please indicate what dotted line represents in right panel (I guess water table level). Also indicate if it represents a measured level or just a schematic representation.

**Response:** *Thanks for your positive comment. We have added the indication of dotted line on the right panel. The dotted line is a schematic representation of the groundwater level (see Fig. 1).*

66. Comment:

Fig 3.: The figure indicates an average river water level of 29m. What does it mean exactly? What is the reference, the zero level? Could it be the distance from the river bed? Is it such a deep river? I think most large European rivers displayed a depth of 2 to 5 m. I've never been to this place but 29m deep sounds more like a lake than a river. BTW, how a 29m deep river can lose flow?

**Response:** *The figure indicates the elevation of river water level. The reference zero level is the sea water level. We have added the elevation of riverbed (26.0 m) as well as riparian ground surface elevation (29.5 m) in the captions (see Figure 3). The river water depth is ranging from 1.9 m to 2.9 m in dry 2019, whereas it fluctuated between 1.7 m and 3.3 m in wet 2021 (see Fig. 3).*

67. Comment:

Fig 4.: The Flowchart and its symbols must be explained in the legend.

**Response:** *We have added the explanation of flowchart and its symbols in the legend (see Fig. 5).*

[Figure]

*Figure 5: Flowchart for quantifying the proportional contributions of river water to riparian trees. The $P_s$ and $P_g$ represent the contributions of riparian deep soil water in the 80−170 cm layer as well as groundwater to riparian trees, respectively. The $s_r^{t-1}$ and $g_r^{t-1}$ represent the proportional contributions of the old river water (before t-1) to riparian deep soil water in the 80−170 cm layer and groundwater, respectively. The $s_s^{t-1}$, $s_r^t$, and $s_g^t$ represent the proportional contributions of in-situ soil water in the 80−170 cm layer at t-1, river water during t-1 to t, and groundwater during t-1 to t for riparian deep soil water in the 80−170 cm layer at t, respectively. The $g_g^{t-1}$ and $g_r^t$ represent the proportional contributions of in-situ groundwater at t-1 and river water from t-1 to t for riparian groundwater at t, respectively.*

68. Comment:

Table 2: units of residence time

**Response:** *We have added the units of residence time in the captions.*

*Here is the revised part: "Table 2: The $^{222}$Rn values in river water, background groundwater and riparian groundwater in three plots (D05, D20, and D45), and the average residence time of*

*groundwater ($T_{res}$, day) in 2021. The background groundwater represents groundwater in aquifers more than 100 m away from the riverbank." (see Table 2).*

---

## Referee Report (RR1)

**Review HESS**

**General comments:**

The authors have generally well addressed the comments made on the first version of the manuscript. Although the wording and grammar can still be improved (I understand writing in English in difficult for non-native English speakers), the science is understandable. After reading the response to reviewers the authors provided, I have some minor comments to add.

I would reorganize the discussion section; I would start by discussing the RWC to riparian trees (section 4.2 in the current version) which is the main objective of this study. Then, I would discuss the link between RWC, WUE and WTD (section 4.3) and end the discussion with the strength/limitations of the method you used (section 4.1). I would also try to improve the link/flow between the points discussed, sometimes it is difficult to understand why you move from one point to another.

**Technical comments:**

*The number before each comment below refers to the number given to each comment on the response to reviewers.*

14. Line 175: Can you clarify; the mean or the median of the water source contributions?

15. Lines 185-187 + 60. Lines 165-168: I would rewrite the section "The average soil WI values at depths of 0-5 cm, …. For the 0-30 cm soil for the 170-300 cm soil layer", the sentences are not clear. For example, you could say: "The average soil WI value for the 0-30 cm soil layer was determined as the average of the soil WI values of the 0-5 cm, 5-10 cm and 20-30 cm soil layers.", and "Similarly, we determined the average soil WI values for the 30-80 cm (average of 40-60 cm and 60-80 cm soil layers), 80-170 cm (average of 90-110 cm and 150-170 cm soil layers) and 170-300 cm (average of …) soil layers".

23. Lines 299-304: I would give the p values as p < XX instead of p = 0.000.

31. Lines 38-40: I would divide the last sentence ("Therefore, understanding […] revegetated riparian zones") in two, for clarity and readability.

73. Lines 224-229: I would rewrite the sentence "ANOVA […] different variables", "incorporating" does not seem to be the right word. I would clarify why each of these tests were performed, also clarify "to investigate the statistic differences of different variables". For example, you could say: "For each variable, we tested the homogeneity of variance between the 2 studied years and between the 3 plots using the Levene's test." …, be more precise about why you performed each test.

Figure 6: I would check the wording, grammar and correct the sentences 2 and 3 of the caption. Maybe something like: "The LMWL was determined for each year from the precipitation samples taken over each year", and "The SWL was determined for each year and plot from the soil water samples taken each year"?

Figure 9: Correct "contributions of water sources to ripaRIAN groundwater" on the y axis.

Figure 11: I would use "contribution of river water to riparian trees" instead or "proportion" on the y axis for consistency. I would also give the p values as p < XX instead of p = 0.000.

Figure S1: Correct the references to the panels in the caption (for example: air temperature and VPD are shown on panels a and b, not c and d).

---

## Author Response (AR2)

Dear Dr. Markus Hrachowitz, editor of Hydrology and Earth System Sciences:

Thanks very much for giving us the opportunity to revise this paper. Upon your request, we have carefully revised our manuscript (hess-2022-327) entitled "Quantifying river water contributions to riparian trees along a losing river: Lessons from stable isotopes and iteration method" after considering all the comments made by you and the other three anonymous reviewers. The comments have helped us greatly improve the overall quality of the manuscript. The following is the point-to-point response to all the comments. The page and line numbers in the following response refer to the revised manuscript with changes marked.

**Response to the editor (Dr. Markus Hrachowitz):**

Dear authors,

Thank you very much for your revisions. The reviewers generally appreciate your efforts and I large share this opinion. However, the reviewers also flag a few more points that could benefit from some more attention.

Please address the remaining reviewer suggestions in detail in a second round of revisions. Please make also sure that the manuscript is attentively proofread by native English speaker, as it still contains multiple language mistakes.

I am looking forward to receiving a revised version of your manuscript as soon as possible.

**Response:** *We are grateful to you and the three reviewers for your constructive and valuable comments on our manuscript. We have substantially revised the manuscript according to the comments (see the replies to Anonymous Referee #1, Referee #3, and Referee #4). We have improved the wording and grammar of this manuscript. We have also asked the EditSprings Company for English language editing of this manuscript.*

**Response to Anonymous Referee #1:**

**General comments:**

1.  Comment:

The authors have generally well addressed the comments made on the first version of the manuscript. Although the wording and grammar can still be improved (I understand writing in English in

difficult for non-native English speakers), the science is understandable. After reading the response to reviewers the authors provided, I have some minor comments to add.

I would reorganize the discussion section; I would start by discussing the RWC to riparian trees (section 4.2 in the current version) which is the main objective of this study. Then, I would discuss the link between RWC, WUE and WTD (section 4.3) and end the discussion with the strength/limitations of the method you used (section 4.1). I would also try to improve the link/flow between the points discussed, sometimes it is difficult to understand why you move from one point to another.

**Response:** *Thank you for the positive comments and insightful suggestions. We have asked the EditSprings Company for English language editing of this manuscript. We reorganized the discussion section and started by discussing the RWC to the transpiration and effects of the distance from the river on RWC which is the main objective of this study. Then we discussed the Link between RWC/WUE/WTD and the implications and ended the discussion with the advantages and limitations of the MixSIAR model and the iteration method (see P.17, Line 457 to P.24, Line 639). We also have improved the link/flow between the points discussed in the discussion section to make the story clearer (see P.17, Line 457 to P.24, Line 639).*

**Technical comments:**

The number before each comment below refers to the number given to each comment on the response to reviewers.

2. Comment:

14. Line 175: Can you clarify; the mean or the median of the water source contributions?

**Response:** *Thank you for the helpful suggestions. We have clarified that the mean and standard deviation (SD) values of different water source contributions could be estimated with the MixSIAR model (see P.9, Lines 234-241).*

3. Comment:

15. Lines 185-187 + 60. Lines 165-168: I would rewrite the section "The average soil WI values at depths of 0-5 cm, …. For the 0-30 cm soil for the 170-300 cm soil layer", the sentences are not clear.

For example, you could say: "The average soil WI value for the 0-30 cm soil layer was determined as the average of the soil WI values of the 0-5 cm, 5-10 cm and 20-30 cm soil layers.", and "Similarly, we determined the average soil WI values for the 30-80 cm (average of 40-60 cm and 60-80 cm soil layers), 80-170 cm (average of 90-110 cm and 150-170 cm soil layers) and 170-300 cm (average of …) soil layers".

**Response:** *We have rewritten this section to make these sentences clearer: "The average soil water isotope values for the 0−30 cm soil layer were determined as the average of the soil water isotope values of 0−5 cm, 5−10 cm, 10−20 cm, and 20−30 cm soil layers because the water isotopes underwent strong evaporation and SWC changed considerably seasonally. We determined the average soil water isotope values for the 30-80 cm (average of 40-60 cm and 60-80 cm soil layers) and 80-170 cm (average of 90-110 cm and 150-170 cm soil layers) soil layers because the water isotopes and SWC were almost stable. The average soil water isotope values for the 170-300 cm soil layer were determined as the average of the soil water isotope values of 190−210 cm, 250−270 cm, and 280−300 cm soil layers, which varied with the fluctuations of groundwater levels."(see P.9, Line 248 to P.10, Line 256).*

4.  Comment:

23. Lines 299-304: I would give the p values as p < XX instead of p = 0.000.

**Response:** *We have changed "p = 0.000" to "p < XX" throughout the whole manuscript  (see P.2, Line 30; P.17, Lines 451 to 453; P.24, Line 654; Fig.11).*

5.  Comment:

31. Lines 38-40: I would divide the last sentence ("Therefore, understanding […] revegetated riparian zones") in two, for clarity and readability.

**Response:** *We have divided this sentence into two "Therefore, it is critical to determine what water sources and how much river water are taken up by riparian trees and the responses of tree water use characteristics to groundwater level variations. This can help us to regulate river runoff and tree's water needs in the revegetated riparian zones." (see P.2, Lines 45-48).*

6.  Comment:

73. Lines 224-229: I would rewrite the sentence "ANOVA […] different variables", "incorporating"

does not seem to be the right word. I would clarify why each of these tests were performed, also clarify "to investigate the statistic differences of different variables". For example, you could say: "For each variable, we tested the homogeneity of variance between the 2 studied years and between the 3 plots using the Levene's test." …, be more precise about why you performed each test.

**Response:** *We have rewritten this sentence to be more precise about why we performed each test: "For each variable, we tested the homogeneity of variance between the two studied years and between the three plots using Levene's test. The one-way analysis of variance (ANOVA) was applied to examine differences in each variable among three plots in 2019 and 2021 ($p < 0.05$)." (see P.13, Lines 332-334).*

7. Comment:

Figure 6: I would check the wording, grammar and correct the sentences 2 and 3 of the caption. Maybe something like: "The LMWL was determined for each year from the precipitation samples taken over each year", and "The SWL was determined for each year and plot from the soil water samples taken each year"?

**Response:** *Thank you for the helpful suggestions. We have corrected the wording and grammar of sentences 2 and 3 of the caption "The local meteoric water line (LMWL) was determined for each year from the precipitation samples taken over each year. The soil water line (SWL) was determined for each year and each plot using the soil water samples taken over each year." (see P.36, Lines 861-865).*

8. Comment:

Figure 9: Correct "contributions of water sources to ripaRIAN groundwater" on the y axis.

**Response:** *We have corrected "Contributions of water sources to riparian groundwater" on the y-axis.*

9. Comment:

Figure 11: I would use "contribution of river water to riparian trees" instead or "proportion" on the y axis for consistency. I would also give the p values as $p < XX$ instead of $p = 0.000$.

**Response:** *Thank you for the helpful suggestion. We have used "Contributions of river water to the transpiration of riparian trees" instead of "Proportions of river water for riparian trees" on the y-*

*axis for consistency. We also changed "p = 0.000" to "p < 0.01".*

10. Comment:

Figure S1: Correct the references to the panels in the caption (for example: air temperature and VPD are shown on panels a and b, not c and d).

**Response:** *We have corrected the references to the panels in the caption "Daily mean temperature (°C) and daily mean vapor pressure deficit (VPD) (kPa) are shown in panels (a) and (b). Daily reference evapotranspiration ($ET_0$) (mm/day) and daily mean net radiation ($W/m^2$) are shown in panels (c) and (d)."*

**Response to Anonymous Referee #3:**

1. Comment:

In the introduction section, the statements need to be more focused and correspond to the purpose of your research. There is currently a lack of progress on water sources for plants at different distances away from the riverbank. In addition, the progress of radioactive isotope (222Rn) in related fields needs to be described.

**Response:** *Thank you for the helpful suggestions. We have closely linked the introduction section and the corresponding purpose of our research (see P.2, Line 35 to P5, Line 111). Moreover, we have supplemented a sentence to link with the second and third paragraphs as well as with the third and fourth paragraphs (see P.3, Lines 73-74; P.4, Lines 92-94).*

*In the second paragraph, we elucidated the effects of different distances away from the riverbank on the river water contributions to the riparian trees: "Growing evidence suggested that riparian trees rarely took up river water directly at a certain distance from the riverbank because their lateral roots could not reach the river (Mensforth et al., 1994; Thorburn and Walker, 1994). Nevertheless, riparian trees can indirectly utilize river water that recharges deep zone (e.g., deep soil water and groundwater) when their roots tap into the groundwater level (Mensforth et al., 1994; Wang et al., 2019a). The RWC to the transpiration of riparian trees may be overestimated if the river water is considered a direct water source." (see P.3, Lines 65 to 71). In this study, the three plots at different*

*distances from the riverbank were selected to provide a gradient of WTD. We have added the critical effect of the different WTDs at different distances from the riverbank on river water contributions to the transpiration of riparian trees (see P.4, Lines 102-108).*

*We have supplemented the process of radioactive Radon ($^{222}Rn$) in related fields in the introduction section as follows: "The radioactive Radon ($^{222}Rn$) has been broadly utilized for tracing groundwater origins and corresponding pathways in riparian zones (Close et al., 2014; Zhao et al., 2018). Based on $^{222}Rn$ concentration, Stellato et al. (2013) estimated the river infiltration velocities into the riparian groundwater system in the Petrignano d'Assisi plain in central Italy, which varied from 1 to 39 m/day. It is helpful to estimate the residence time of recharged groundwater from river water and its effects on the RWC to the transpiration of riparian trees." (see P.4, Lines 85-90).*

2. Comment:

Lines 35: What is the meaning of running dry? Giving some statements is necessary.

**Response:** *We have changed "running dry" to "drying up" to indicate the river has dried up (see P.2, Line 39).*

3. Comment:

The representativeness of Salix babylonica in the study area needs to be highlighted, otherwise any tree species can be selected

**Response:** *Thank you for the helpful suggestion. We have highlighted the representativeness of Salix babylonica "The deep-rooted riparian weeping willow (Salix babylonica L.) was one of the most widely planted species alongside the Chaobai River because the S. babylonica trees could adapt well to dramatic fluctuations in the WTD. Hence, this research selected S. babylonica trees as representative of riparian species. Three plots at different distances of 5 m (D05), 20 m (D20), and 45 m (D45) from the riverbank (one plot per distance) were also selected for field measurements and sample collection (Fig. 1)." (see P5, Lines 132-137).*

4. Comment:

How was the proposed iteration method used to quantify the proportions of the indirect river water source taken up by riparian trees? Please give more details steps.

**Response:** *Thanks for your suggestions. The detailed steps of the proposed iteration method have been highlighted: "In this study, water stable isotopes ($\delta^2H$ and $\delta^{18}O$) were integrated within the MixSIAR model and an iteration method was proposed to identify the contributions of the indirect river water that recharged riparian deep water to the transpiration of riparian S. babylonica trees (Figs. 4-5). First, the direct water source (including soil water in different layers and groundwater) contributions to the transpiration of riparian trees were determined via $\delta^2H$ and $\delta^{18}O$ values of different waters and the MixSIAR model. Second, the proportional contributions of river water to riparian deep water (i.e., riparian groundwater and deep soil water in the 80−170 cm layer) were determined by the MixSIAR model and water stable isotopes. Finally, the proposed iteration method was applied to quantify the proportions of the indirect river water source taken up by riparian trees (Figs. 4-5)." (See P.9, Lines 224-231).*

*The total RWC (including the old river water before t-1 and current river water between t-1 and t) to riparian S. babylonica trees near the losing rivers, which was described as Equation (4) (see P.12, Line 310-315).*

$RWC = P_s * S_r + P_g * G_r$

$= P_s*(s_r^t + s_r^{t-1}) + P_g*(g_r^t + g_r^{t-1})$

$= P_s*( s_r^t + s_r^t * s_s^{t-1} + s_r^t *( s_s^{t-1} )^2 + s_g^t * g_r^t + s_g* g_r^t * g_g^{t-1} + s_g^t * g_r^t *( g_g^{t-1} )^2) + P_g*(g_r^t + g_r^t *g_g^{t-1} + g_r^t *(g_g^{t-1})^2)$

$= (P_s*s_r^t + P_g*g_r^t + P_s*s_g^t *g_r^t) + (P_s*s_r^t *s_s^{t-1} + P_g*g_r^t *g_g^{t-1} + P_s*g_r^t *s_g^t *g_g^{t-1}) + (P_s*s_r^t *(s_s^{t-1})^2 + P_g*g_r^t *(g_g^{t-1})^2 + P_s*s_g^t *g_r^t *(g_g^{t-1})^2)$           (4)

*"The expression of "$P_s*s_r^t + P_g*g_r^t + P_s*s_g^t *g_r^t$" in Equation (4) was proposed to determine the current river water (between t-1 and t) contributions to the transpiration of riparian trees. The second iteration ($P_s*s_r^t *s_s^{t-1}+ P_g*g_r^t *g_g^{t-1} + P_s*g_r^t *s_g^t *g_g^{t-1}$) and the third iteration ($P_s*s_r^t *(s_s^{t-1})^2 + P_g*g_r^t *(g_g^{t-1})^2 + P_s*s_g^t *g_r^t *(g_g^{t-1})^2$) were applied to quantify the proportional contributions of old river water that recharged riparian in-situ deep water to trees (Fig. 5). We only applied three iterations because the differences between the RWCs in the third iteration and the next iteration were smaller than 0.1%. Using this proposed iteration method, we accurately estimated the total proportions of old and current river waters to the transpiration of riparian trees." (see P12, Lines 324-330).*

5. Comment:

Line 210: When combining different soil layers to 170−300 cm, 170-190 cm of soil layer is missing. "It indicated that the lateral roots of S. babylonica trees could not tap into the river", however, the roots of S. babylonica trees were not investigated.

**Response:** *Thanks for your suggestions. We have collected soil samples only at three depths of 190−210 cm, 250−270 cm, and 280−300 cm in the 170-300 cm soil layer. The average of the soil water isotope values at these three depths represented the soil water isotope values in the 170-300 cm soil layer in this study.*

*The roots of S. babylonica trees were not investigated in this study. However, the approximate lateral root extent could be inferred basing on the projected edge of the canopy of S. babylonica. "Mensforth et al. (1994) and Thorburn and Walker (1994) characterized the projected edge of the canopy as the extension range of lateral roots. In this way, it is possible to determine whether or not riparian trees take up river water directly. The projected edge of the canopy in our study was less than 5 m for the riparian S. babylonica trees which were closest to the river (5 m away from the riverbank). This indicated that the lateral roots of S. babylonica trees could not tap into the river." (see P10, Lines 264-269).*

*We have illuminated that the lateral roots of S. babylonica trees should be directly investigated to confirm our inference in further research in the 4.3 section of "Advantages and limitations of the MixSIAR model and the iteration method" in this study: "Third, we inferred the approximate lateral root extent based on the projected edge of the canopy of S. babylonica, which indicated that S. babylonica trees could not tap into the river or take up river water directly. However, the lateral roots of S. babylonica trees should be directly investigated in further research to confirm our inference." (See P23, Lines 630-633)*

6. Comment:

2.4.2 section, soil water in the 0−80 cm layer at t-1 contributes to deep soil moisture, but some isotopic changes from t-1 to t are not considered, such as fractionation during this period. Similar issues need to be considered when calculating groundwater sources.

**Response:** *Thank you for the insightful suggestions. We have assumed that "the isotopic changes from t-1 to t (such as fractionation during this period) were negligible when calculating the*

*contribution of upper soil water (i.e., in the 0-80 cm or 0-170 cm layers) at t-1 to deep moisture (i.e., soil water in the 80-170 cm layer or groundwater)" in this study (see P11. Lines 292-294). The main reason was that upper soil water at t-1 generally contributed a small proportion to deep moisture. For example, the contributions of soil water in the 0-80 cm layer at t-1 to deep soil moisture were $16.0 \pm 4.7\%$, and the soil water in the 0-170 cm layer at t-1 contributed $16.3 \pm 7.1\%$ to groundwater (Figs. 8 and 9). This indicated that the fractionation-induced isotopic change between t-1 and t was little during upper soil water infiltration into deep layers.*

*Nevertheless, the isotopic changes between t-1 and t might be varied with intense rainfall and strong evaporation on the soil surface, which will be further investigated and considered in the calculation of deep soil water/groundwater sources. We have supplemented the statement in further research in the 4.3 section of "Advantages and limitations of the MixSIAR model and the iteration method" in this study: "First, the riparian deep-water sources were identified using the water isotopic data collected in campaigns taking place at an interval of about one month. The riparian soil water movement was complex, and the water stable isotopes might not be uniform between the two campaigns along the losing river. Nevertheless, the isotopic changes from t-1 to t (such as fractionation during this period) were negligible when calculating the contribution of upper soil water (i.e., in the 0-80 cm or 0-170 cm layers) at t-1 to deep moisture (i.e., soil water in the 80-170 cm layer or groundwater). Assuming the isotopic uniformity over such a time interval may cause uncertainties in estimating the RWC to the transpiration of riparian deep water." (see P23. Lines 618-625).*

7.  Comment:

The result and discussion about the average residence time (Tres, day) of recharged groundwater from river water is rare. It will be more meaningful to properly supplement the statements on the effect of residence time and plant water sources.

**Response:** *Thank you for the insightful suggestions. In this study, we elucidated the extremely short residence time of recharged groundwater from river water at different distances (5 m, 20 m, and 45 m) from the riverbank in the result section. "There was a significant increase of $^{222}$Rn activity in groundwater from D05 ($494.5 \pm 107.5$ Bq/m$^3$) to D45 ($787.4 \pm 153.2$ Bq/m$^3$) (p < 0.05) (Table 1). The Tres of recharged groundwater from river water increased from D05 (0 days) to D45 (0.15 ±*

*0.13 days) (Table 1). This also indicated that the river recharged riparian deep strata rapidly and frequently, particularly more significant in the plots closer to the riverbank." (see P16. Lines 416-420).*

*While in the discussion section, we speculated some possible processes of recharging riparian groundwater from river water in detail based on the groundwater residence time and river water contributions (RWC) to riparian groundwater. These detailed processes of recharging groundwater from river water can help us understand why river water contributed a small proportion to riparian trees. "The ecohydrological separation (Brooks et al., 2010; Evaristo et al., 2015; Allen et al., 2019; Sprenger et al., 2019) possibly resulted in large isotopic discrepancies between fast-moving water flow and immobile water for plant water uptake. Although the residence time of recharged groundwater from river water was extremely short (less than 0.28 days) (Table 1), only one-third of riparian groundwater was replaced by the lateral seepage of river water (Fig. 9). Our finding probably indicates that river water recharged mobile groundwater quickly but could not completely replace water held tightly in the soil pores (Brooks et al., 2010; Evaristo et al., 2015; Allen et al., 2019). This was consistent with Sprenger et al. (2019) who found that the lateral seepage of river water or rising groundwater level could briefly saturate riparian soils but could not entirely replace/flush immobile waters or homogenize different water pools isotopically." (see P18. Line 490 to P19. Line 499). In addition, we also supplemented the indications of the residence time of recharged groundwater from river water in the 4.1 section: "The declining water table and increasing residence time of recharged groundwater from D05 to D45 could consequently lead to the decreasing RWC to riparian deep water along the distance away from the riverbank. Thus, the interannual and spatial variabilities of the RWC to the transpiration of riparian S. babylonica trees were generally attributed to the various RWCs to riparian deep water rather than the water uptake patterns of riparian trees." (see P20, Lines 539-543).*

8.  Comment:

Figure 4 is confusing and need to be revised

**Response:** *We have revised Figure 4 and illustrated it in the caption.*

[Figure]

*Figure 4: Schematic diagram of potential water sources: riparian deep soil water in the 80−170 cm layer (a) and groundwater (b). The red box represents riparian deep soil water in the 80−170 cm layer in panel (a) and groundwater in panel (b), respectively. The dark blue arrow indicates different potential water sources of riparian deep water.*

**Response to Anonymous Referee #4:**

**General comments:**

1. Comment:

The comments of the reviewers have been largely addressed and the manuscript was significantly improved, very nice work! Yet, some open questions remain before the manuscript can be accepted for publication.

The sample size of the experiment seems very small, the authors sampled only 1 tree per plot and there are only 3 plots (1 per distance, see Fig.1), i.e., all results are just based on three trees? It would be important to clarify and properly discuss this point in the manuscript. Also, I did not find any information on how many leaves were sampled.

If the sample size was so small, please elaborate more in the text on: Why did you not sample more trees? Why is the chosen small sample size enough to support your conclusions? What implications does the small sample size have for your results?

I added some new remarks and comments, mainly on the MM section.

There are still many language mistakes, especially in the new text that need to be addressed. Please consider having a native speaker for proof-reading (I am also not a native speaker).

**Response:** *Thank you for your positive comments and insightful suggestions. We have carefully addressed the reasons for the sample size of the experiment and clarified this point in the discussion section in the revised version (see P23, Lines 633 to P24, 639). The MM section has been revised detailly according to the remarks and comments (see the responses to Comments 2-7). We have corrected the language mistakes and have the EditSprings Company for proof-reading of the revised manuscript.*

*More than 50 mature leave samples were collected for each tree at each sampling campaign. We have added the number of leave samples as follows: "Meanwhile, more than 50 mature leaves without petioles were sampled from the collected stems using pruning shears and mixed into one leaf sample for $\delta^{13}C$ analysis." (see P7, Lines 167-169).*

*We think that the sample size of our experiment can support our conclusions, which can be explained by three un-exclusive reasons. First, we sampled three replicates of stem samples for each riparian S. babylonica tree at different distances of 5 m, 20 m, and 45 m from the riverbank. In order to reduce errors in random sampling, the average isotopic composition of three replicates was used to represent the actual isotopic value of each riparian S. babylonica tree (see P6, Line 165 to P7, Line 167). Secondly, we collected 108 plant stem samples in total and obtained 36 sets of different isotopic composition data for riparian S. babylonica trees along a gradient of depth to the water table (WTD) (ranging from 0.3 m to 4.0 m) in dry and wet years. In addition, the leaf $\delta^{13}C$ in these 36 sets of riparian S. babylonica trees varied a lot (between −26.5‰ and −30.4‰). The spatial (three plots at different distances from the riverbank) and temporal (two years) disparities in the tree water use characteristics (root water uptake patterns and leaf WUE) and water availability conditions (WTD and soil water content) can support us to investigate the effects of river water on the water use of riparian trees along a gradient of WTD. Thirdly, we have already investigated the overall water table conditions in the riparian zone along the Chaobai River in both 2019 and 2021 (these data have not been published yet). The riparian WTD along the reach of Chaobai River shown in Fig. 1 ranged from 0.2 m to 4.3 m. The chosen site in this study is the most representative site since there is a significant water table variation (range from 0.3 m to 4.0 m) in both spatial (at distances of 5 m, 20 m, and 45 m from the riverbank) and temporal (dry 2019 and wet 2021) scales. Therefore, the representative site coupled with three different plots is enough to support out conclusions.*

*We have elaborated more in the text to illuminate what implications the small sample size has for our results. "Fourth, the riparian WTD along the studied reach of Chaobai River (from Dam 5 to Dam 4) ranged from 0.2 m to 4.3 m in two studied years (these data have not been published yet). The selected site in this study was the most representative site since there was a significant water table variation (ranging from 0.3 m to 4.0 m) in the two studied years. However, the implications of quantifying the effects of river water on the water use of riparian trees in this study are only applicable to relatively shallow water table conditions (with the WTD less than 4 m). Further investigations should be conducted at deep-WTD sites to better understand and regulate river runoff and tree's water needs." (see P23, Lines 633 to P24, 639).*

Some general mistakes/points throughout the whole document:

2. Comment:

often "the" or "a" are missing.

**Response:** *Thank you for the helpful suggestions. We have checked the whole manuscript and added "a" or "the" in the text.*

3. Comment:

"at distance of 5m, 20 m, and 45 m away from" does not sound correct? maybe: at a distance of 5 m, 20 m, and 45 m from the shore, or: "at 5, 20, 45 m distance from the shore", "at distances of…"

**Response:** *Thank you for the helpful suggestions. We have changed "at distance of 5m, 20 m, and 45 m away from" to "at a distance of 5 m, 20 m, and 45 m from …" or "at distances of…" throughout the whole manuscript.*

4. Comment:

in "a" wet and "a" dry year, in "the" dry year?

**Response:** *We have changed "in dry/wet year" to "in a dry/wet year" throughout the whole manuscript.*

5. Comment:

it must be "water stable isotopes" NOT "stable water isotopes"

**Response:** *We have changed "stable water isotopes" to "water stable isotopes" throughout the*

*whole manuscript.*

6. Comment:

the use of "that" is often wrong as in "unclear that how" or "because that"

**Response:** *We deleted "that" in "unclear that how" as well as "because that". We have corrected "that" in the entire document.*

7. Comment:

"RWC to riparian trees" does not make sense as pointed out by both reviewers, it is the RWC to the transpiration/water uptake... , check the entire document for that expression. This also includes figures such as figure 10.

**Response:** *We have changed "RWC to riparian trees" to "RWC to the transpiration of riparian trees" in both the text and figure sections in Fig. 10.*

8. Comment:

please write the full name of the species when mentioned first (G. biloba)

**Response:** *Thank you for the helpful suggestions. We have written the full name of the species when mentioned first in this manuscript (see P4, Line 107).*

9. Comment:

I would also add "values" after δ2H and δ18O

**Response:** *Thank you for the helpful suggestions. We have added "values" after $\delta^2H$ and $\delta^{18}O$ throughout the whole manuscript.*

10. Comment:

in-situ NOT in-suit; check spelling of this word in entire document including figures.

**Response:** *We have changed "in-suit" to "in-situ" in the entire document including figures.*

**Specific comments:**

More comments can be found in the pdf (note: the comments on language mistakes are not complete).

Also, I noticed that the text in the authors' replies in the open discussion differs from the newly

submitted manuscript text. Line-by-line comments (line numbers refer to the tracked changes version):

11. Comment:

l14: brackets confusing, include into sentence

**Response:** *We have changed "rivers losing flow into underlying groundwater" to "rivers flow into underlying groundwater" in the bracket (see P1, Line 14).*

12. Comment:

l16: water stable isotopes

**Response:** *We have changed "stable water isotopes" to "water stable isotopes" throughout the whole manuscript.*

13. Comment:

l22: "contributed by 20.3% of water to riparian tree", to what of the riparian tree?

**Response:** *We have corrected this unclear sentence to "contributed 20.3% of water to the transpiration of riparian trees" (see P1, Line 23).*

14. Comment:

l138: "in combination with"

**Response:** *We have changed "together with" to "in combination with" (see P5, Line 113; P24, Line 643).*

15. Comment:

l56-57: trees… tree species; it is neither to combine or use the same term

**Response:** *We have changed "Similar findings have also been found in riparian phreatophyte trees (Populus fremontii and Salix gooddingii) and riparian deep-rooted tree species" to "Similar findings have also been reported regarding riparian phreatophyte trees (Populus fremontii and Salix*

*gooddingii) and riparian deep-rooted trees (Busch et al., 1992; Bowling et al., 2017; Wang et al., 2019a)." (see P3, Lines 61-63).*

16. Comment:

l76: mention Radon

**Response:** *We have changed "radioactive isotope ($^{222}$Rn)" to "radioactive Radon ($^{222}$Rn)" (see P4, Line 85).*

17. Comment:

l85: higher than what?

**Response:** *We have corrected this sentence as follows: "riparian Eucalyptus camaldulensis with more frequent access to river water had a higher tree WUE compared to those far away from the riverbank." (see P4, Lines 99-102).*

18. Comment:

l100: "a temperate…"?

**Response:** *We have changed "The temperate continental sub-humid monsoon climate prevails in this area" to "a temperate continental sub-humid monsoon climate prevails in this area" (see P5, Lines 123-124).*

19. Comment:

l156: please state clearly that it is one plot per distance (as it is shown in Fig.1)?

**Response:** *We have corrected this unclear sentence as follows: "Three plots at different distances of 5 m (D05), 20 m (D20), and 45 m (D45) from the riverbank (one plot per distance) were also selected for field measurements and sample collection (Fig. 1)." (see P5, Lines 136-137).*

20. Comment:

l165: add "and in 2021"

**Response:** *We have changed "in 2019 and 2021" to "in 2019 and in 2021" (See P6, Line 145).*

21. Comment:

l186: please also add here that it is in total 3 trees?

**Response:** *We have added "in total 3 trees" in this sentence: "One riparian S. babylonica tree was selected in each plot (three trees in total) for $\delta^2H$ and $\delta^{18}O$ measurements in xylem water as well as $\delta^{13}C$ analysis in plant leaves." (see P6, Lines 161-162).*

22. Comment:

l192: how many leaf samples did you take?

**Response:** *More than 50 mature leave samples were collected for each tree at each sampling campaign. We have added the number of leave samples as follows: "Meanwhile, more than 50 mature leaves without petioles were sampled from the collected stems using pruning shears and mixed into one leaf sample for $\delta^{13}C$ analysis." (see P7, Lines 167-169).*

23. Comment:

l208: were extracted samples filtered to remove organics?

**Response:** *Yes. We have added "All the extracted water from the xylem and soil samples was filtered to remove impurities." in the text (see P7, Lines 181-182).*

24. Comment:

l213: how did you determine the accuracy?

**Response:** *The measurement accuracy for the IRIS and IRMS systems were determined by both the measured and true values of standard samples.*

25. Comment:

l214: did you only use one standard?

**Response:** *We used one standard (Vienna Standard Mean Ocean Water, VSMOW) to calibrate and*

*normalize the $\delta^2H$ and $\delta^{18}O$ measurements in different waters. We used another standard (Vienna Pee Dee Belemnite, V-PDB) to calibrating leaf $\delta^{13}C$ values.*

26. Comment:

l226: how did you determine the precision?

**Response:** *The precision of the $^{222}Rn$ monitor was the factory parameter, which was determined by the instrument manufacturer.*

27. Comment:

l272-273: this sentence is confusing, maybe: "To account for different soil layers…"?

**Response:** *This is a good comment. We have changed "Soil water at different depths was taken up by riparian S. babylonica directly." to "Soil water was an important direct water source for the transpiration of riparian S. babylonica trees." (see P9, Lines 244-245).*

28. Comment:

l286-287: would be nice to add numbers to show how similar

**Response:** *We have added numbers to show how similar as follows: "As the isotopic composition of soil water in the 170−300 cm layer (−57.6‰ ± 2.0‰ for $\delta^2H$ and −7.3‰ ± 0.1‰ for $\delta^{18}O$) was similar to that of groundwater (−57.7‰ ± 1.4‰ for $\delta^2H$ and −7.4‰ ± 0.1‰ for $\delta^{18}O$), they were considered to be one water source (groundwater)." (see P10, Lines 262-269).*

29. Comment:

l378: in Microsoft Excel

**Response:** *We have changed "in Excel" to "in Microsoft Excel" (see P13, Line 341).*

30. Comment:

l386: was the peak the same for both years?

**Response:** *No, the peak values of $ET_0$ as well as their corresponding time was different between the*

*2 studied years. The original sentence "The daily mean VPD and ET$_0$ increased during the observation period, reaching a peak in June and May, respectively (Fig. S1)." is confusing and this result doesn't help to explain the discussions. Therefore, we deleted this sentence.*

31. Comment:

l389: what did you test here, the remaining months? rather state the time frame.

**Response:** *We have changed "the remaining months" to "the rest of the months". And we also added the time frame of "the rest of the months" in this sentence: "There was a significant difference in the average daily ET0 from June to September between the dry year of 2019 (5.0 mm/day) and the wet year of 2021 (4.3 mm/day) (p < 0.05), but no significant difference was observed during the rest of observation period (i.e., April, May, October, and November) between the two years (p > 0.05) (Fig. S1c and d)." (see P13, Lines 351-355).*

32. Comment:

l390-391: please specify observation period here by months

**Response:** *Thank you for the helpful suggestions. We have specified "observation period" by months when the "observation period" first occurred in the result section as follows: "The observation period (from April to November) in 2021 was wet with total precipitation of 802.5 mm, which was 1.8 times greater than for the drier year 2019 (445.6 mm) (Fig. 2a)." (see P13, Lines 345-346).*

33. Comment:

l419: in both years

**Response:** *We have changed "in both two years" to "in both years" (see P15, Line 393).*

34. Comment:

l432: delete "it was found"

**Response:** *We have deleted "it was found" in this sentence.*

35. Comment:

l352: "most significant"

**Response:** *We have changed "most significantly" to "most significant" (see P16, Line 424).*

36. Comment:

l397: I am not sure if I understand correctly, to me it would make sense to state: "However, the riparian deep-water sources were identified using the water isotopic data collected in campaigns taking place in an interval of about one month."?

**Response:** *Thank you for the helpful suggestion. We have corrected this sentence as follows: "the riparian deep-water sources were identified using the water isotopic data collected in campaigns taking place at an interval of about one month." (see P23, Lines 618-620).*

37. Comment:

Figure 2 caption: I would not say "changes". Just start with "Monthly…" (it is the absolute values, right?)

**Response:** *Thank you for the helpful suggestion. Yes, the monthly average precipitation amount from 1961 to 2021 is absolute value. We have deleted "changes in" in this sentence.*

38. Comment:

Figure 11: State/clarify that leaf d13C etc. are monthly (mean?) data from all distances in the legend.

**Response:** *The WTD, leaf $\delta^{13}C$ values, and river water contributions to the transpiration of riparian S. babylonica are monthly data at each plot at a distance of 5 m, 20 m, and 45 m from the riverbank during the observation period in both years. We have clarified in the caption of Figure 11.*

39. Comment:

Table 2: do you only have one value for each time point, or is the value a mean per time point?

**Response:** *We sampled more than 50 mature leaves without petioles from the collected stems and mixed into one sample to measure leaf $\delta^{13}C$ values. Therefore, we only have one value for each time*

*point and each plot.*

Additional comments in the pdf (note: the comments on language mistakes are not complete). Also, I noticed that the text in the authors' replies in the open discussion differs from the newly submitted manuscript text. Line-by-line comments (line numbers refer to the tracked changes version):

**Response:** *"The response letter to reviewer" document used by anonymous reviewer 4 was not the newly version of the response letter to reviewer. Some comments in the anonymous reviewer 4-revised document have been corrected in the newly version of the response letter to reviewer. But there are still many helpful and insightful suggestions for improving the manuscript. Here we showed the additional comments in the pdf.*

40. Comment:

Both the trace plots and three diagnostic tests are used to check that the MixSAIR model has converged (Stock and Semmens, 2013). Were used?

**Response:** *We have changed "are used" to "were used".*

41. Comment:

"Because the riparian trees" is not a sentence.

**Response:** *We have changed "Because….." to "This is because ….." (see P19, Lines 506 and 517).*

42. Comment:

"because that" is wrong

**Response:** *We deleted "that" in "because that". We have corrected "that" in the entire document.*

43. Comment:

Would need

**Response:** *We have changed "need" to "would need" throughout the entire document (see P22, Line 590; P23, Line 629).*

44. Comment:

phrasing.., even..., ?

**Response:** *We have corrected this sentence as follows: "several recent studies showed that phreatophytic/deep-rooted trees predominantly extended roots into fine pores to take up immobile soil water." (see P19, Lines 499-501).*

45. Comment:

add information in brackets to the sentence

**Response:** *We have corrected this sentence as follows: "In our study, we separated and determined the contributions of indirect river water sources (i.e., the river-recharged deep soil water in the 80−170 cm layer and groundwater also contained river water) to the transpiration of riparian trees." (see P19, Lines 521-523).*

46. Comment:

Change "supposed" to "suppose"

**Response:** *We have changed "supposed" to "suppose" (see P21, Lines 561-562).*

47. Comment:

Change "in wet year" to "in the wet year"

**Response:** *We have added "the" in "in the wet year" or "in the dry year" throughout the whole manuscript.*

48. Comment:

"in consistent with" is wrong

**Response:** *We have changed "These relationships are in consistent with previous studies" to "These relationships are consistent with previous studies" (see P21, Line 555).*

49. Comment:

Change "are likely due to that" to "are likely because"

**Response:** *We have changed "are likely due to that" to "are likely because" throughout the entire manuscript (see P21, Line 557).*

50. Comment:

Change "it" to "our finding"

**Response:** *We have changed "It was consistent with…" to "Our finding was consistent with…." (see P21, Line 579).*

51. Comment:

Change "supposed" to "suppose"

**Response:** *We have changed "supposed" to "suppose" (see P12, Line 305).*

52. Comment:

Figure S1: change "during the observation period in 2019 (c) and 2021 (c)." to "during the observation period in 2019 (c) and 2021 (d)."

**Response:** *We have corrected this sentence to "Daily reference evapotranspiration ($ET_0$) (mm/day) and daily mean net radiation (W/$m^2$) are shown in panels (c) and (d).".*

53. Comment:

Figure S1: change "Temperature" to "Air temperature"; change "KPa" to "kPa"

**Response:** *We have changed "Temperature" to "Air temperature" and changed "KPa" to "kPa".*

54. Comment:

give amount per year as well e.g. in brackets. Delete "and…."

**Response:** *We have added the precipitation sample amount per year in this sentence: "A total of 135 precipitation samples were collected throughout the whole years of 2019 (53 samples) and 2021 (82 samples)." (see P6, Lines 155-156). We have deleted "and" in the "And a total of …." (see P6,*

*Line 155).*

55. Comment:

so 3 three per distance? please add this information to be more clear

**Response:** *We have corrected this sentence as follows: "One riparian S. babylonica tree was selected in each plot (three trees in total) for $\delta^2 H$ and $\delta^{18} O$ measurements in xylem water as well as $\delta^{13} C$ analysis in plant leaves. The mean breast-height diameter of three sampled trees at different distances of 5 m, 20 m, and 45 m from the riverbank was $28.6 \pm 4.4$ cm." (see P6, Lines 161-164).*

56. Comment:

you mean summarized to? averaged?

**Response:** *We have corrected this sentence as follows: "The average soil water isotope values for the $0-30$ cm soil layer were determined as the average of the soil water isotope values of $0-5$ cm, $5-10$ cm, $10-20$ cm, and $20-30$ cm soil layers" (see P9, Line 248 to P10, Line 250).*

57. Comment:

Change "isotopes values" to "isotope values"

**Response:** *We have change "isotopes values" to "isotope values" (see P10, Line 249).*

58. Comment:

Table S1 might be worth not to put in the supplemental but in the text

**Response:** *We have put Table S1 in the text (Table 3).*

59. Comment:

Table 1: did you add this to your method part, this would be important to mention that negative values were set 0

**Response:** *We have clarified that the negative values were set to 0 in our mention part.*

60. Comment:

you could still consider testing how the fit looks like within a year (additionally). Especially since you plot them nicely with different colours.

**Response:** *We have corrected the Figure 11 as follows:*

[Figure]

*Figure 11: Relationships between the contributions of river water to the transpiration of riparian trees and the water table depth (a), between the leaf $\delta^{13}C$ values and the water table depth (b), and between the leaf $\delta^{13}C$ values and proportions of river water contributions to riparian trees (c). The red line represents the linear relationship fitted by the monthly data in three plots in 2019, while the blue line represents the linear relationship fitted by the monthly data in three plots in 2021. The black line represents the linear relationship fitted by the monthly data in three plots in both years. The WTD, leaf $\delta^{13}C$ values, and river water contributions to the transpiration of riparian S. babylonica are monthly data at each plot at a distance of 5 m, 20 m, and 45 m from the riverbank during the observation period in both years.*

61. Comment:

Change "WUE" to "leaf WUE".

**Response:** *We have added "leaf" in the "WUE" throughout the entire manuscript.*

62. Comment:

Delete "that" in the "it remains unclear that how to separate…"

**Response:** *We have deleted "that" in the sentence of "it remains unclear that how to separate…"*

*in the entire document.*

63. Comment:

Change "These suggested" to "This suggests".

**Response:** *We have changed "These suggested" to "This suggests" (see P24, Line 655).*

64. Comment:

delete in linear functions, or write linearly correlated

**Response:** *We have deleted "in linear functions" and corrected "linearly correlated" throughout the whole manuscript (see P1, Line 29 to P2, Line 30; P17, Line 453; P24, Lines 653-654).*

65. Comment:

Change "would trigger" to "may trigger"

**Response:** *We have changed "would trigger" to "may encourage" throughout the entire document (see P2, Line 30; P22, Line 597; P24, Line 656).*

66. Comment:

Add reference here

**Response:** *We have added the reference behind the sentence "It is generally assumed that when $C_{System}$ is around or lower than 80 Bq/m³, the existing $C_{System}$ can be ignored accordingly (Saphymo, 2017)." (see P8, Lines 204-205).*

67. Comment:

soil layers were rather group to 4?

**Response:** *Thanks for your helpful suggestions. This sentence is confusing. The soil layers were indeed rather group to 4. We have corrected this sentence as follows: "To reduce errors in the analytical procedure, four soil layers (0−30 cm, 30−80 cm, 80−170 cm, and 170−300 cm) were determined to identify the main root water uptake depth of riparian trees according to seasonal*

*variations in the SWC, water isotopes, and WTD." (see P9, Lines 246-248).*

68. Comment:

Figure 4: replace "of" by ":"

**Response:** *Thanks for your helpful suggestions. We have replaced "of" by ":"*

69. Comment:

Change "WTDs" to "WTD", change "$\delta^2$H and $\delta^{18}$O in different water sources and xylems" to "$\delta^2$H and $\delta^{18}$O of different water sources and xylem water"

**Response:** *Thanks for your helpful suggestions. We have changed "WTDs" to "WTD" and changed "$\delta^2$H and $\delta^{18}$O in different water sources and xylems" to "$\delta^2$H and $\delta^{18}$O of different water sources and xylem water" throughout the whole manuscript.*

70. Comment:

"It was ascribed to that" is wrong

**Response:** *We have change "It was ascribed to that…" to "This discrepancy was ascribed to the fact that…" (see P20, Line 545).*

71. Comment:

Change "riparian S. babylonica tree" to "riparian S. babylonica trees" in the sentence of "the outer projected edge of canopy was less than 5 m for riparian S. babylonica tree closest to the river.

**Response:** *We have change "riparian S. babylonica tree" to "riparian S. babylonica trees" (see P10, Line 268).*

72. Comment:

Change "It indicated" to "This indicated" in the sentence of "It indicated that the lateral roots of S. babylonica trees".

**Response:** *We have change "It indicated" to "This indicated" (see P10, Line 268).*

73. Comment:

"…, which could not be recommended in order to both…" sounds strange

**Response:** *We have corrected this sentence as follows: "The rising groundwater level may encourage riparian trees to increase the water extraction from groundwater/river and to exhibit a consumptive river-water-use pattern, which can have an adverse impact on the protection of both rivers and riparian vegetation." (see P2, Lines 30-33).*

74. Comment:

you mean the contribution to the transpiration flux? or uptake? you do not really know how they "used" the water as you have no transpiration data

**Response:** *We have changed "water use patterns" in this sentence to "water uptake patterns". Nevertheless, we still have several expressions of "water use characteristics of riparian trees" to indicate "the water uptake patterns and leaf WUE of riparian trees" in this manuscript.*

75. Comment:

Change "in riparian zone" to "in riparian zones"

**Response:** *We have changed "in riparian zone" to "in riparian zones" (see P4, Lines 87 and 103; P22, Line 598).*

76. Comment:

please state directly how you calculated the efficiency

**Response:** *We have added the equation of water extraction efficiency in the text (see P7, Lines 187-190).*

$$E_{WE} = \frac{W_{BE}-W_{AE}}{W_{BE}-W_{OD}} \times 100\% \tag{1}$$

*Whereas $E_{WE}$ represents the efficiency of water extraction. $W_{BE}$ and $W_{AE}$ represent the weights of xylem/soil samples before and after extraction, respectively. $W_{OD}$ represents the weights of oven-dried xylem or soil samples.*

77. Comment:

put 2021 after month directly

**Response:** *We have corrected this sentence as follows: "The riparian S. babylonica took up the most river water in July 2021 (35.2 ± 7.0%), whereas the highest RWC to the transpiration of riparian trees occurred in June 2019 (24.2% ± 1.6%)." (see P16, Lines 428-430).*

78. Comment:

The response to the comment "-27.7 is not remarkably larger than -29.7" does not consider the reviewer's comment. please better state: differences were significant but small.

**Response:** *Yes, the differences in leaf $\delta^{13}C$ of riparian S. babylonica trees between two studied years were small (2.0‰) but significant (p < 0.05). We have corrected this sentence as follows: "The leaf $\delta^{13}C$ of riparian S. babylonica trees was significantly higher in 2019 (−27.7‰ ± 1.0 ‰) than in 2021 (−29.7‰ ± 0.7 ‰) (p < 0.05) (Table 2)." (see P17, Lines 444-445).*

79. Comment:

Change "characterized as" to "characterized by"

**Response:** *We have changed "characterized as" to "characterized by" (see P21, Line 558).*

80. Comment:

Change "more proportions" to "a higher proportion"

**Response:** *The expression of "more proportions" does not occur in the newly submitted manuscript text.*

81. Comment:

however, you did not normalize your WUE against radiation, maybe repeat that radiation was not different between years.

**Response:** *Yes, we have repeated that radiation was not different between two years in the 4.2 section: "Higher leaf WUE associated with lower RWC to the transpiration of riparian trees and*

*lower groundwater levels are likely because water stress restricts the stomatal conductance and further reduces the transpiration rate of riparian trees. Specifically, the dry year of 2019 was characterized by higher water demand (indicated by higher VPD) and lower water availability compared to the wet year of 2021, but the energy resource (indicated by net radiation) for riparian trees was similar between the two years (Figs. S1-S2). Hence, we suppose that water limitation rather than energy limitation regulates the leaf-level stomatal conductance of riparian S. babylonica trees. The high water demands but low river water availability in the dry year likely resulted in the stomatal closure of riparian trees to minimize water loss, which eventually led to a decrease in transpiration rate and even photosynthetic rate (Fabiani et al., 2021; Behzad et al., 2022). Aguilos et al. (2018) further found that water stress would enhance radiation-normalized WUE because the lack of water availability induced a stronger reduction in transpiration than photosynthesis. With no difference in the average net radiation between dry and wet years, the lower river water availability in a dry year probably increased leaf WUE. It can be inferred that riparian S. babylonica trees took up more river water and possibly exhibited a consumptive river-water-use pattern in the wet year compared to the dry year. This agreed well with previous investigations during which the woody plants showed lower leaf WUE and consumptive water use patterns in the rainy season, while they showed higher leaf WUE and conservative water use patterns with lower soil water availability in the dry season (Horton and Clark, 2001; Cao et al., 2020; Behzad et al., 2022). However, consumptive river water taken up by riparian trees could result in a great loss of river water, which should be avoided in the riparian zone of a losing river that is under restoration by "ecological water"." (see P21, Lines 556-575).*